# Spatially patterned hydrogen peroxide orchestrates stomatal development in Arabidopsis

Wen Shi [1,3], Lingyan Wang [1,3], Lianmei Yao[1], Wei Hao[1], Chao Han [1], Min Fan [1], Wenfei Wang[2] & Ming-Yi Bai [1] ✉

Stomatal pores allow gas exchange between plant and atmosphere. Stomatal development is regulated by multiple intrinsic developmental and environmental signals. Here, we show that spatially patterned hydrogen peroxide ($H_2O_2$) plays an essential role in stomatal development. $H_2O_2$ is remarkably enriched in meristemoids, which is established by spatial expression patterns of $H_2O_2$-scavenging enzyme *CAT2* and *APX1*. SPEECHLESS (SPCH), a master regulator of stomatal development, directly binds to the promoters of *CAT2* and *APX1* to repress their expression in meristemoid cells. Mutations in *CAT2* or *APX1* result in an increased stomatal index. Ectopic expression of *CAT2* driven by *SPCH* promoter significantly inhibits the stomatal development. Furthermore, $H_2O_2$ activates the energy sensor SnRK1 by inducing the nuclear localization of the catalytic α-subunit KIN10, which stabilizes SPCH to promote stomatal development. Overall, these results demonstrate that the spatial pattern of $H_2O_2$ in epidermal leaves is critical for the optimal stomatal development in Arabidopsis.

Stomata control gaseous fluxes between the internal leaf air spaces and the external atmosphere and play pivotal roles in regulating $CO_2$ uptake for photosynthesis as well as water vaporization through transpiration[1–3]. In Arabidopsis (*Arabidopsis thaliana*), stomata arise from a subset of undifferentiated meristemoid mother cells, which undergo an asymmetric cell division to generate a meristemoid and a stomatal lineage ground cell (SLGC) as its smaller and larger daughters, respectively. The meristemoid is the stomatal precursor, which may go through one to three asymmetric divisions to produce guard mother cells[1,2]. The guard mother cell undergoes a single symmetric division to eventually generate a pair of guard cells surrounding a microscopic pore[4]. The division and differentiation of stomatal lineage cells are tightly controlled by a suit of closely related and sequentially expressed basic helix-loop-helix (bHLH) transcription factors SPEECHLESS (SPCH), MUTE, and FAMA in Arabidopsis[4–6]. *SPCH* is mainly expressed in the meristemoid cells and regulates the initiation and proliferation of stomatal precursors[5]. Several endogenous and environmental signals finely regulate SPCH activity to control stomatal development in response to changing environment[7–11].

Hydrogen peroxide ($H_2O_2$), an endogenous oxidant present in all aerobic cells, was recently viewed as a key signaling molecule in plant stem cell regulation[12,13]. Several studies have reported that $H_2O_2$ spatial distribution is critical for the plant stem cell activities[14–17]. In the shoot apical meristem (SAM), $H_2O_2$ is remarkably enriched in the peripheral zone to promote stem cell differentiation[15]. The accumulated $H_2O_2$ induces the reversible protein phase separation of transcription factor TERMINATING FLOWER (TMF) in the SAM of tomato. $H_2O_2$-induced oxidative modification activates TMF to bind and inhibit the expression of the floral identity gene *ANANTHA*, thereby directing the stem cell fate for flowering transition[17]. In the root apical meristem (RAM), $H_2O_2$ mainly accumulates in the expanding cells of the elongation zone, whereas $O_2^{·-}$ mainly accumulates in the dividing cells of

[1]The Key Laboratory of Plant Development and Environmental Adaptation Biology, Ministry of Education, School of Life Sciences, Shandong University, Qingdao 266237, China. [2]College of Horticulture, College of Life Sciences, Hai xia Institute of Science and Technology, Fujian Agriculture and Forestry University, 350002 Fuzhou, China. [3]These authors contributed equally: Wen Shi, Lingyan Wang. ✉e-mail: baimingyi@sdu.edu.cn

meristem zone. The basic helix-loop-helix (bHLH) transcription factor UPBEAT1(UPB1) regulates the expression of a set of peroxidases that are involved in establishing the reactive oxygen species (ROS) gradient pattern in root tips[14]. The ROOT MERISTEM GROWTH FACTOR1 (RGF1) peptide also plays a critical role for the ROS gradient pattern in root tips. RGF1 binding to RGF1 Receptor (RGFR) activates the transcription factor RGF1 INDUCIBLE TRANSCRIPTION FACTOR1 (RITF1) to control the ROS distribution and enhance the stability of PLT1/2, thus determining the root meristem size[16]. In addition to SAM and RAM, there are other meristems in plants to regulate their own development; however, it remains unclear whether $H_2O_2$ presents a specific spatial distribution pattern in other meristems to control their activities.

The maintenance of cellular and organismal energy homeostasis is vital for all living organisms. Eukaryotes have evolved a very sophisticated system to modulate the metabolism based on nutrient availability[18]. A central component of this system in plants is SUCROSE NON-FERMENTING-1 (SNF1)-RELATED KINASE 1 (SnRK1), which is orthologous to the AMP-ACTIVATED KINASE (AMPK) and SNF1 in mammals and yeast, respectively[19–21]. SnRK1/AMPK/SNF1 function as cellular energy sensors to induce catabolic reactions and repress energy-consuming anabolic processes when energy supplies become limited[22,23]. SnRK1 in Arabidopsis is a heterotrimeric complex that includes a catalytic α-subunit (KIN10, KIN11, and KIN12), two regulatory subunits consisting of three isoforms of KINβ (KINβ1, KINβ2, and KINβ3), and an atypical KINβγ subunit[20,24]. Low-energy stress caused by darkness or hypoxia promoted KIN10 nuclear translocation to regulate downstream gene expression[25]. The myristoylated and membrane-associated SnRK1 β regulatory subunits interact with KIN10 and restrict its nuclear localization, thereby inhibiting its functions[25,26]. A recent study showed that KIN10 exhibits the ubiquitous expression pattern in all epidermal cells of Arabidopsis leaves, but is mainly localized to the nucleus of meristemoid cells to phosphorylate and stabilize SPCH, then promoting stomatal development[11]. However, the molecular mechanism of the specific nuclear localization of KIN10 in meristemoid cells remains unclear. Here, we demonstrated that $H_2O_2$ is specifically enriched in meristemoids to reduce the interaction between KIN10 and KINβ2, thereby promoting KIN10 nuclear localization and activating SPCH to promote stomatal development.

## Results

### $H_2O_2$ induces KIN10 nuclear localization in plants

Our previous study showed that when plants were grown in ½ MS liquid medium containing 1% sucrose under long-day condition, *KIN10* is expressed in all epidermal cells of Arabidopsis leaves, but the subcellular localization of KIN10 protein exhibited a specific spatial pattern and was enriched in the nucleus of the stomatal lineage cells and guard cells as well as in the cytoplasm of the pavement cells in the epidermal cells of Arabidopsis cotyledons[11]. To determine whether KIN10 has such a spatial subcellular localization pattern in plants that are grown under other environmental conditions, we analyzed the subcellular localization of KIN10 in cotyledon epidermal cells of plants grown in ½ MS solid medium with or without 1% sucrose under different light intensities or different photoperiod conditions. The results showed that the nuclear-localized KIN10 proteins were highly enriched in the stomatal lineage cells and guard cells under all examined conditions, and that sucrose enhanced this distribution pattern of KIN10 (Supplementary Fig. 1a–d). To determine whether this spatial subcellular localization pattern of KIN10 exists in other leaves, we analyzed the subcellular localization of KIN10 in the rosette leaves and found a similar distribution pattern as that in cotyledons (Supplementary Fig. 1e, f). These results suggest that the spatial subcellular localization pattern of KIN10 is widespread in plant leaves.

Metabolic stress caused by hypoxia or 3-(3,4-dichlorophenyl)−1,1-dimethylurea (DCMU) treatment has been reported to induce KIN10 nuclear translocation[21]. ROS are the byproducts of aerobic metabolism

and play critical roles in plant growth, development, and stress responses[12,13]. Therefore, we speculated that metabolic stress may generate ROS to induce KIN10 nuclear localization. To test this possibility, we measured the $H_2O_2$ content of wild-type plants in the presence or absence of DCMU and/or potassium iodide (KI), a quencher of $H_2O_2$, using the fluorescent dye 2′,7′- dichlorodihydrofluorescein diacetate ($H_2DCFDA$). The results showed that DCMU treatment significantly increased the $H_2O_2$ levels, whereas DCMU and KI cotreatment inhibited $H_2O_2$ accumulation (Supplementary Figs. 2a, b and 3a, b). Staining with 3,3′-diaminobenzidine (DAB) further confirmed that DCMU treatment resulted in $H_2O_2$ accumulation in the leaves, whereas KI cotreatment strongly counteracted the effects of DCMU (Supplementary Fig. 2c, d). To examine the effects of DCMU and KI on KIN10 subcellular localization, we analyzed the subcellular localization of KIN10 in the leaf epidermal cells of the *pKIN10::KIN10-YFP* transgenic plants that were grown on ½ MS solid medium containing or spraying with or without DCMU and/or KI. DCMU treatment significantly increased the nuclear-to-cytoplasmic ratio of KIN10-YFP in the stomatal lineage cells, pavement cells and guard cells, but such effects of DCMU were counteracted by the cotreatments with KI (Fig. 1b–d, Supplementary Figs. 3c, d, and 4a, b). These results indicate that $H_2O_2$ plays an important role for DCMU-induced KIN10 nuclear localization in plants.

To examine the effects of $H_2O_2$ on KIN10 subcellular localization, we analyzed the relative levels of nuclear and cytoplasmic signals of KIN10-YFP with or without $H_2O_2$ treatment. The results showed that $H_2O_2$ treatment significantly increased the nuclear-to-cytoplasmic ratio of KIN10-YFP in meristemoids, stomatal lineage cells, pavement cells, and guard cells (Fig. 1e–g, Supplementary Figs. 4c, d and 7a–f). To determine whether the $H_2O_2$-induced nuclear localization of KIN10 is due to the biosynthesis of KIN10 protein, we analyzed the $H_2O_2$-promoted nuclear enrichment of KIN10 in the presence of cycloheximide (CHX), a protein translation inhibitor. The results showed that CHX didn't affect the $H_2O_2$-mediated nuclear localization of KIN10 (Fig. 1e–g and Supplementary Fig. 4c, d). To determine whether other types of ROS regulate the nuclear localization of KIN10, we analyzed the subcellular localization of KIN10 in the presence of methyl viologen (MV) or N,N′-dimethylthiourea (DMTU), which are specific superoxide anion ($O_2^{·-}$)-generating reagent or the scavenger of free radicals, respectively[27,28]. The results showed that MV and DMTU treatment had no significant effects on the subcellular localization of KIN10 in the epidermal cells of Arabidopsis cotyledons (Supplementary Fig. 5a–d). These results suggest $H_2O_2$ specifically induces the nuclear localization of KIN10 in plants.

To corroborate these pharmacological data, we analyzed the effects of $H_2O_2$ on KIN10 subcellular localization in the catalase over-expressing plants (*p35S::CAT2-Myc*), which accumulate less $H_2O_2$ in plants. The results showed that the nuclear signals of KIN10-YFP in the stomatal lineage cells, pavement cells and guard cells were significantly reduced in the *p35S::CAT2-Myc* plants compared with those in the wild-type background, and such reduction effects were counteracted by $H_2O_2$ treatment (Fig. 1h–l and Supplementary Fig. 6a, b). These results suggest $H_2O_2$ plays an important role in the KIN10 nuclear localization in plants.

### $H_2O_2$ specifically accumulates in meristemoids

Considering the important role of $H_2O_2$ in KIN10 nuclear localization in plants, it is speculated that the meristemoid cells may contain high levels of $H_2O_2$. To test this hypothesis, we examined the $H_2O_2$ distribution pattern in cotyledon epidermal cells of wild-type plants that were grown on ½ MS solid medium containing 1% sucrose under long-day conditions using the $H_2DCFDA$ and BES-$H_2O_2$-Ac fluorescent dyes. The reduced non-fluorescent compound $H_2DCFDA$ can be oxidized and converted into fluorescent 2′, 7′-dichlorofluorescein (DCF) by intracellular $H_2O_2$. BES-$H_2O_2$-Ac is a highly selective fluorescent probe

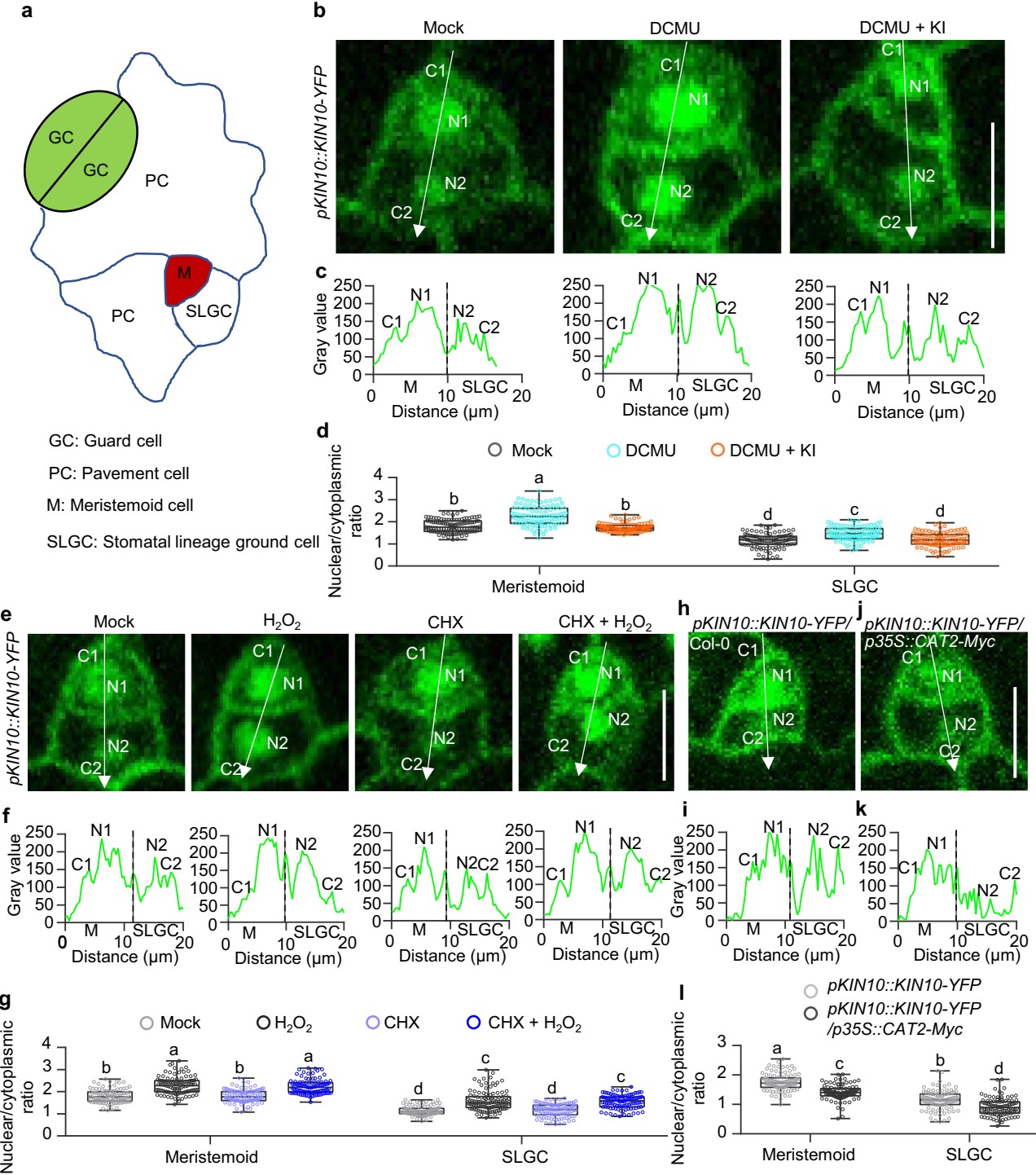

**Fig. 1 | H₂O₂ induces the nuclear localization of KIN10. a** Diagram of progressive stomatal lineage in the Arabidopsis early leaf epidermis. M meristemoid cell, SLGC stomatal lineage ground cell, PC pavement cell, GC guard cell. **b–d** KI prevents DCMU-induced nuclear localization of KIN10-YFP in M and SLGC. Seedlings of *pKIN10::KIN10-YFP* transgenic plants were treated with or without 50 μM DCMU and/or 1 mM KI for 12 h. *n* = 113 (Mock), *n* = 107 (DCMU) and *n* = 108 (DCMU + KI) meristemoid or SLGC cells in 10 cotyledons were analyzed by ImageJ in **d. e–g** H₂O₂ induces the nuclear localization of KIN10-YFP in plants. Seedlings of *pKIN10::KIN10-YFP* transgenic plants were treated with or without 2 mM H₂O₂ and/or 50 μM CHX for 2 h. *n* = 102 (Mock), *n* = 110 (H₂O₂), *n* = 103 (CHX) and *n* = 101 (CHX + H₂O₂) meristemoid or SLGC cells in 10 cotyledons were analyzed by ImageJ in **g. h–l** Overexpression of *CAT2* reduces the nuclear localization of KIN10-YFP. *n* = 102 (Col-0), *n* = 104 (*p35S::CAT2-Myc*) meristemoid or SLGC cells in 10 cotyledons were analyzed by ImageJ in **l**. Seedlings of *pKIN10::KIN10-YFP* (**b**–**g**) and *pKIN10::KIN10-YFP/p35S::CAT2-Myc* (**h**–**l**) transgenic plants were grown on ½ MS solid medium containing 1% sucrose under 16 h light/8 h dark photoperiod with 100 μMol/m²/s for 4 days. Serial Z-stack projection images were used for quantitative analysis. The white arrows inside the images show the areas used for line scan measurements that yielded plot profiles shown in the lower panels. C1, cytoplasmic signals of KIN10-YFP in meristemoid cells; N1, nuclear signals of KIN10-YFP in meristemoid cells; C2, cytoplasmic signals of KIN10-YFP in SLGC; N2, nuclear signals of KIN10-YFP in SLGC. Box plot shows maxima, first quartile, median, third quartile, minima. Different letters above the bars indicate statistically significant differences between the samples (Brown-Forsythe ANOVA analysis followed by Dunnett's T3 multiple comparisons test, *p* < 0.05 (**d**, **g**); Two-way ANOVA analysis followed by Tukey's multiple comparisons test, *p* < 0.05 (**l**)). Adjustments were made for multiple comparisons test. Scale bars in confocal images represent 10 μm.

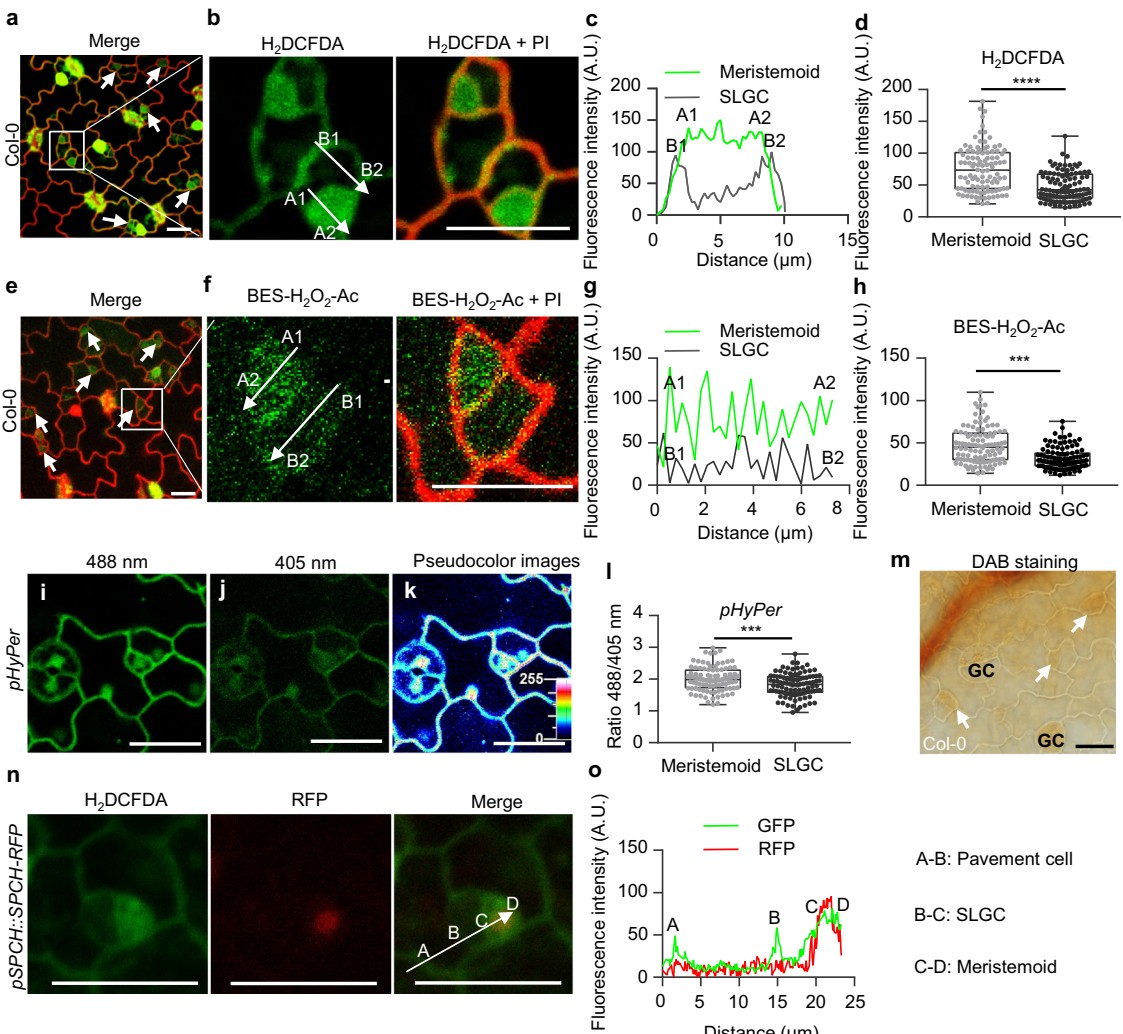

**Fig. 2 | H₂O₂ specifically accumulates in meristemoid cells.** Measurement of H₂O₂ in the epidermal cells of wild-type cotyledon using H₂DCFDA (**a**–**d**) and BES-H₂O₂-Ac (**e**–**h**). Magnifications of the meristemoid and SLGC are shown in **b** and **f**. The white arrows inside the images show the areas used for line scan measurements that yielded plot profiles shown in **c** and **g**. The arrow labeling with A1 and A2 represents the fluorescent signals in meristemoid cells, and the arrow labeling with B1 and B2 represents the fluorescent signals in SLGC. **i**–**l**, Quantification of *pHyPer* fluorescent signals in cotyledon epidermal cells. Ratio imaging of Arabidopsis epidermal cells expressing *HyPer* (**k**). Panels **i**, **j**, **k** represent the same region excited at 488 and 405 nm and the corresponding ratio image, respectively. Emission was collected at 530 nm with a bandpass of 30 nm. Note that the red color indicates a higher ROS concentration and blue color indicates a lower level. **m**, DAB staining in the epidermal cells of leaves. Arrow indicates the meristemoid cell. **n**, **o**, Co-localization of H₂DCFDA and RFP signals from *pSPCH::SPCH-RFP* in cotyledon epidermal cells. Seedlings of Col-0, *pHyPer* or *pSPCH::SPCH-RFP* transgenic plants were grown on ½ MS solid medium containing 1% sucrose under 16 h light/8 h dark photoperiod with 100 μMol/m²/s for 4 days (**a**–**m**) or 3 days (**n**, **o**), and then stained with H₂DCFDA, BES-H₂O₂-Ac or DAB. Fluorescent signals were taken using LSM700 microscope from Zeiss. *n* = 104, *n* = 108 and *n* = 103 meristemoid or SLGC cells from 10 cotyledons were examined in **d**, **h**, and **l**, respectively. The fluorescent signals of H₂DCFDA, BES-H₂O₂-Ac or RFP were determined along a line drawn on the confocal images using ImageJ software. Scale bars in confocal images represent 20 μm. The box plot shows maxima, first quartile, median, third quartile, minima. Asterisk between the bars indicated statistically significant differences between the samples (Two-tailed student's t test, ****$p < 0.0001$, ***$p < 0.001$).

for H₂O₂, and exhibits a higher H₂O₂ specificity but a lower fluorescent signal than H₂DCFDA[29]. The results showed that the fluorescent signals of H₂DCFDA and BES-H₂O₂-Ac were specifically enriched in the meristemoid cells and guard cells, but less distributed in pavement cells of Arabidopsis leaves on the abaxial side. (Fig. 2a–h and Supplementary Fig. 8a–d). HyPer is a genetically encoded probe, in which the H₂O₂-sensitive transcription factor OxyR is incorporated into circularly permuted yellow fluorescent protein[30]. It has been shown that Hyper is highly sensitive to H₂O₂ and not to other ROS[30]. The plants expressing the *HyPer* have been used to analyze the real-time production of H₂O₂ in plants. In the *pHyPer* transgenic plants, H₂O₂ content is measured by determining the ratio of fluorescent signals at an excitation wavelength of 488 nm relative to that at 405 nm[31]. In the abaxial epidermal cells of wild-type leaves, the ratio of fluorescent signals was higher in

the meristemoid cells and guard cells, but lower in the pavement cells (Fig. 2i-l and Supplementary Fig. 8e, f). DAB staining was also used to analyze the H₂O₂ distribution pattern in the leaves and found that H₂O₂ was more abundant in the abaxial meristemoid cells and guard cells than in the pavement cells (Fig. 2m). To further verify whether H₂O₂ is enriched in meristemoids, we examined the co-localization of the fluorescent signals of H₂DCFDA and *pSPCH::SPCH-RFP*. The results showed that the fluorescent signals of H₂DCFDA and SPCH-RFP were present simultaneously in the meristemoid cells (Fig. 2n, o). These results indicate that H₂O₂ is highly enriched in the abaxial meristemoids and guard cells in the epidermal cells of Arabidopsis leaves.

To verify the enrichment of H₂O₂ in meristemoids and guard cells, we also analyzed the H₂O₂ distribution pattern in *cat2 cat3, upb1, p35S::CAT2-Myc*, and *p35S::UPB1* plants[14,32]. CAT2 and CAT3 are key

catalase enzymes that scavenge $H_2O_2$ in plants[33,34]. UPB1 has been reported to repress a set of peroxidases to regulate the meristem activity in the plants[14,15]. The H2DCFDA fluorescent signals were stronger in the *cat2 cat3* and *p35S::UPB1* plants than those in the wild-type plants, whereas the distribution pattern of $H_2O_2$ was similar to that in wild-type plants. The H2DCFDA fluorescent signals were weaker in the *upb1* and *p35S::CAT2-Myc* plants than those in wild-type plants, and H2DCFDA fluorescent signals were observed only in the abaxial guard cells, but very weak in meristemoid and pavement cells (Supplementary Fig. 9a–d). These results indicate that catalase and UPB1 play important roles in the $H_2O_2$ distribution pattern in epidermal cells of leaves.

To determine whether the spatial pattern of $H_2O_2$ in the epidermal cells is widespread, we analyzed the $H_2O_2$ distribution pattern in the cotyledon epidermal cells of plants grown on ½ MS solid medium with or without 1% sucrose under different light intensities or different photoperiod conditions. The results showed that $H_2O_2$ specifically accumulated in the meristemoid cells under all conditions examined (Supplementary Fig. 10a–d). We also analyzed the $H_2O_2$ distribution pattern in the rosette leaves, and found that $H_2O_2$ was highly enriched in the meristemoid cells (Supplementary Figs. 10e, f and 11a–d). These results suggest that the spatial pattern of $H_2O_2$ is widespread in the epidermal cell of Arabidopsis leaves.

To determine whether other types of ROS display the specific distribution pattern in the epidermal cells of leaves, we analyzed the $O_2^{\cdot-}$ distribution pattern using the fluorescent dye dihydroethidium (DHE), which specifically detects $O_2^{\cdot-}$. The results showed that $O_2^{\cdot-}$ was evenly distributed in the abaxial epidermal cells of leaves and did not exhibit a specific distribution pattern (Supplementary Fig. 12a–f).

### SPCH directly represses the expression of *CAT2* and *APX1* in meristemoid cells

To elucidate the mechanism underlying $H_2O_2$-specific accumulation in meristemoid cells, we analyzed the expression patterns of ROS-related genes in the published cell-type-specific transcriptomic data of Arabidopsis leaves. The expression levels of several ROS-scavenging genes, such as *CAT2* and *ASCORBATE PEROXIDASE 1* (*APX1*), were lower in the meristemoid cells than in other types of cells[35]. To test the bioinformatic analysis results, the *pCAT2::GFP* and *pAPX1::GFP* transgenic plants were generated to examine the expression patterns of *CAT2* and *APX1* in the epidermal cells of leaves. The fluorescent signals of *pCAT2::GFP* and *pAPX1::GFP* were clearly observed in the pavement cells, but were difficult to detect in the smaller cells and guard cells (Fig. 3a–d and Supplementary Fig. 13a–c). Generally, the expression patterns of *CAT2* and *APX1* revealed a positive relationship between the intensity of GFP and the size of epidermal cells (Supplementary Fig. 14a–d). Co-localization analysis with *pCAT2::GFP* and *pSPCH::SPCH-RFP* revealed that *pCAT2::GFP* exhibited very weak signals in the meristemoid cells labeled with *pSPCH::SPCH-RFP* (Fig. 3e–g). These results indicate that *CAT2* and *APX1* have low expression levels in meristemoids.

The published ChIP-Seq data showed that SPCH binds to the promoters of *CAT2* and *APX1*[36]. To verify this result, we performed chromatin immunoprecipitation quantitative PCR (ChIP-qPCR) and DNA-protein pull-down assays. The results confirmed that SPCH directly binds to the promoters of *CAT2* and *APX1* (Fig. 3h, i and Supplementary Fig. 13d, e). Quantitative RT-PCR analysis showed that the expression levels of *CAT2* and *APX1* were increased in *spch-4* mutant but decreased in *p35S::SPCH-Myc* transgenic plants (Fig. 3j, and Supplementary Fig. 13f). These results indicate that SPCH directly binds to the promoters of the ROS-scavenging gene *CAT2* and *APX1* to repress their expression.

### $H_2O_2$ promotes stomatal development

To address the biological function of enriched $H_2O_2$ in meristemoids, we examined the stomatal phenotypes of wild-type plants, *cat2, cat3,* *cat2 cat3, cat1 cat2 cat3* mutants, and *p35S::CAT2-Myc* plants. The amount of meristemoid cells and GMC in the abaxial epidermis at 5 days after germination (DAG) was significantly increased in *cat2, cat3, cat2 cat3* and *cat1 cat2 cat3* mutants compared with that in the wild type, but was decreased in *p35S::CAT2-Myc* transgenic plants (Fig. 4a, b). As a result, at 8 DAG, the *cat2* and *cat3* single mutants exhibited a slightly increased stomatal index, which refers to the number of stomata relative to the total number of epidermal cells, and the *cat2 cat3* double mutants and *cat1 cat2 cat3* triple mutants showed a significantly increased stomatal index (Supplementary Fig. 15a). Overexpression of *CAT2* dramatically reduced the stomatal index, and $H_2O_2$ treatment partially suppressed the decreased stomatal index of *CAT2* over-expression plants (Supplementary Fig. 15a, b). To examine the role of spatial $H_2O_2$ distribution in stomatal development, we manipulated CAT2 activity in the stomatal lineage cells by expressing *CAT2* under *SPCH* promoter. The results showed that the expression of *CAT2* in stomatal lineage cells significantly inhibited stomatal initiation and reduced the stomatal index, and $H_2O_2$ treatment also partially suppressed the decreased stomatal index of *pSPCH::CAT2-RFP* transgenic plants (Fig. 4a, b and Supplementary Fig. 15a, b). Consistent with this, the *apx1* mutant and *p35S::UPB1* transgenic plants exhibited the significant increased stomatal index, whereas the *upb1* mutant exhibited the decreased stomatal index (Supplementary Fig. 15c). These results indicate that $H_2O_2$ plays an important role in stomatal development.

To further determine $H_2O_2$ effects on stomatal development, we analyzed the stomatal phenotypes of wild-type and *kin10* mutants that were grown on ½ MS solid medium containing various concentrations of $H_2O_2$. The results showed that $H_2O_2$ increased the stomatal index at low concentrations, but such effects decreased in the presence of high concentrations of $H_2O_2$ (Fig. 4c). The *kin10* mutant displayed a low stomatal index, and $H_2O_2$ treatment had no significant effects on stomatal development of *kin10* mutant (Fig. 4c). Consistent with this, $H_2O_2$ failed to increase the stomatal index of *pSPCH::SPCH^{S4A}-RFP/spch-4* transgenic plants, in which SPCH^{S4A} could not be phosphorylated by KIN10. $H_2O_2$ treatment increased SPCH protein levels, but had no significant effects on SPCH^{S4A} (Supplementary Fig. 16a–c). These results suggest that KIN10 contributes to $H_2O_2$-induced stomatal development.

To determine the effects of CAT-mediated $H_2O_2$ scavenging on the stomatal lineage cell fate transition, we compared the fluorescent signals of cell-type-specific marker lines *pSPCH::nucGFP* and *pSPCH::SPCH-GFP* in wild-type and *cat2 cat3* mutant background. The marker line *pSPCH::nucGFP* is mainly expressed in the stomatal lineage cells and the marker line *pSPCH::SPCH-GFP* is mainly expressed in the meristemoids and MMCs. We observed that mutation of *CAT2* and *CAT3* markedly increased the number of cells marked by *pSPCH::nucGFP* and *pSPCH::SPCH-GFP* compared with those in the wild-type background, whereas KI treatment substantially suppressed this increase (Fig. 4d–g). These results indicate that $H_2O_2$ increases the frequency of stomata-producing division in Arabidopsis leaves.

To further determine whether other types of ROS regulate stomatal development, we analyzed the stomatal phenotype of plants treated with DMTU, the scavenger of free radicals. DMTU treatment did not alter the expression patterns of *pSPCH::nucGFP* and *pSPCH::SPCH-GFP*, and had no significant effects on the stomatal index (Supplementary Fig. 17a–e). Moreover, the *ndufs4* mutant, in which accumulate less $O_2^{\cdot-}$ in plants by affecting the NADH dehydrogenase subunits[37,38], showed a similar stomatal index as wide type (Supplementary Fig. 17f). To further test the biological function of $O_2^{\cdot-}$ in stomatal development, we treated plants with Diethyldithiocarbamic acid (DDC) and MV, which increased the $O_2^{\cdot-}$ content in plants, and found that DDC and MV had no significant effects on stomatal development (Supplementary Fig. 17g, h). The published transcriptomic data indicated that the expression of $O_2^{\cdot-}$ biosynthesis and metabolism-related genes did not display cell-type-specific pattern[35].

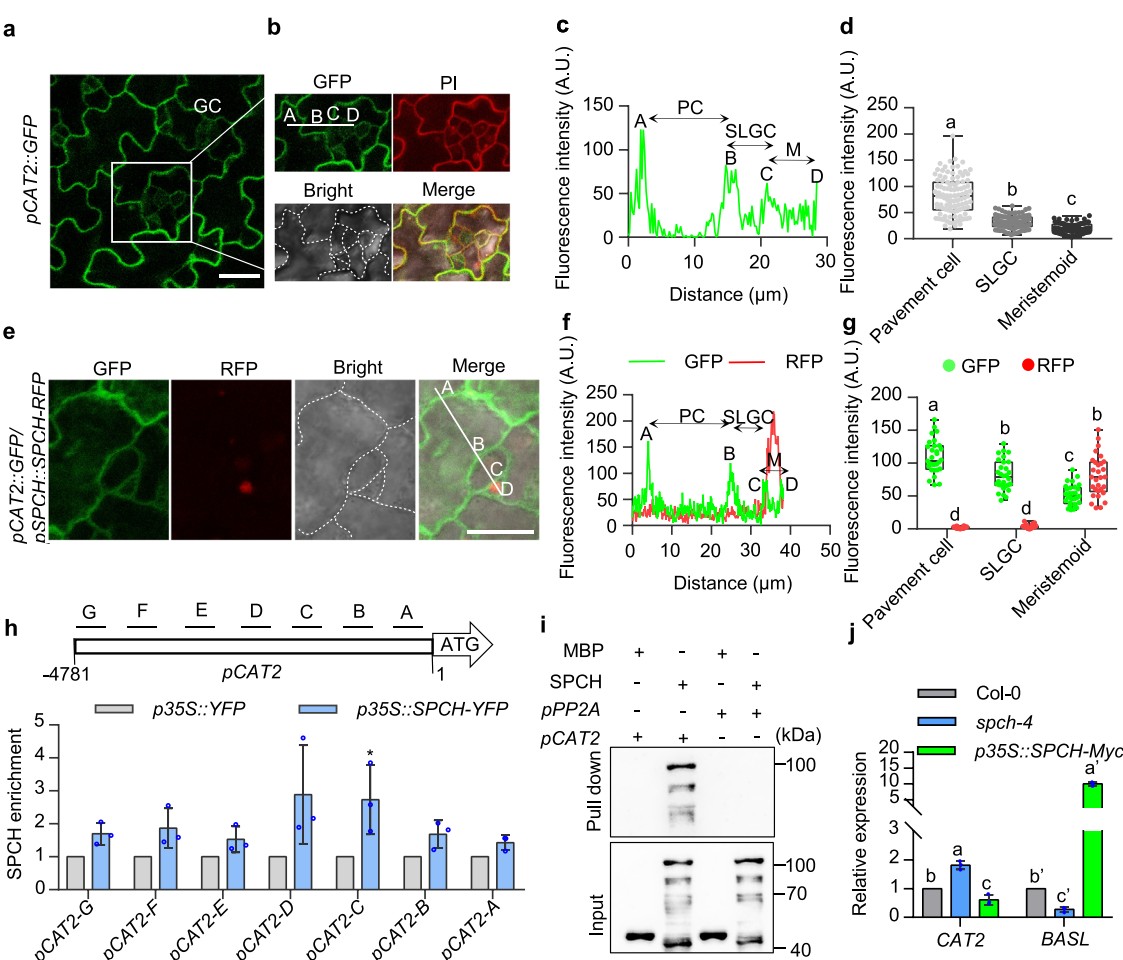

**Fig. 3 | SPCH directly represses the expression of *CAT2*. a–d** The expression pattern of *CAT2* in the cotyledon epidermal cells. Magnifications of the epidermal cells are shown in **b**. The *pCAT2::GFP* fluorescent signals were in green, PI-marked cell outlines were in red. Merged images showed the higher *CAT2* expression levels in the pavement cells and the lower levels in the smaller cells where cell division occurs. *n* = 110 (Pavement), *n* = 102 (SLGC) and *n* = 102 (meristemoid) cells from 10 cotyledons were examined in **d**. **e–g** Co-localization of *pCAT2::GFP* and *pSPCH::SPCH-RFP* in cotyledon epidermal cells. *n* = 30 (Pavement), *n* = 30 (SLGC), and *n* = 30 (meristemoid) cells were examined in **g**. **h** Quantitative ChIP-PCR showed that SPCH binds to *CAT2* promoter. Seedlings of *p35S::YFP* and *p35S::SPCH-YFP* were used to performed ChIP assays. The levels of SPCH binding were calculated as the ratio between *p35S::SPCH-YFP* and *p35S::YFP*, and then normalized to that of control gene *PP2A*. Error bars indicate standard deviation (S.D.) (n = 3 biologically independent samples). **i** SPCH directly binds to the promoter of *CAT2* in vitro. MBP or MBP-SPCH were incubated with biotinylated DNA fragments from the *PP2A* and *CAT2* promoters immobilized on streptavidin beads. The DNA-bound proteins were immunoblotted using anti-MBP antibody. **j** RT-qPCR analysis of the expression of *CAT2* and *BASL* in wild type, *spch-4* mutant and *p35S::SPCH-Myc* transgenic plants. *PP2A* gene was used as an internal control. Error bars indicate standard deviation (S.D.) (*n* = 3 biologically independent samples). Seedlings of *pCAT2::GFP* (**a–d**), *pCAT2::GFP/pSPCH::SPCH-RFP* (**e–g**), Col-0, *spch-4* and *p35S::SPCH-Myc* transgenic plants (**j**) were grown on ½ MS solid medium containing 1% sucrose under 16 h light/8 h dark photoperiod with 100 μMol/m²/s for 4 days (**a–d**) or 3 days (**e–g**) or 6 days (**j**).The fluorescent signals of GFP were determined along a line drawn on the confocal images using ImageJ software. Fluorescent signals were taken using LSM700 microscope from Zeiss. The box plot shows maxima, first quartile, median, third quartile, minima. Different letters above the bars indicated statistically significant differences between the samples (Brown-Forsythe ANOVA analysis followed by Dunnett's T3 multiple comparisons test, *p* < 0.05 (**d, g**); Adjustments were made for multiple comparisons test; Two-tailed student's t-test, *\*p* < 0.05 (**h**); One-way ANOVA analysis followed by Tukey's multiple comparisons test, *p* < 0.05 (**j**). Adjustments were made for multiple comparisons test.). Scale bars in confocal images represent 20 μm.

Collectively, these results suggest that $O_2^{\cdot-}$ is not involved in stomatal development in Arabidopsis.

## H₂O₂ reduces the interaction between KIN10 and SnRK1 β subunits

A recent study showed that the membrane-associated SnRK1 regulator β subunits interact with KIN10 to restrict KIN10 nuclear localization[25]. Our results indicated that $H_2O_2$ promotes the nuclear localization of KIN10 to induce stomatal development. Therefore, we speculate that $H_2O_2$ may regulate KIN10 subcellular localization by regulating the interaction between KIN10 and SnRK1 β subunits. To test this hypothesis, we analyzed $H_2O_2$ effects on the binding affinity of KIN10 to KINβ2, which plays major roles in the inhibition of KIN10 nuclear

localization[25]. The pull-down assays revealed that glutathione-S-transferase (GST)-KIN10 interacted with maltose-binding protein (MBP)-KINβ2, but not MBP alone, and $H_2O_2$ treatment markedly inhibited the interaction between KIN10 and KINβ2 in a dose-dependent manner (Fig. 5a, b and Supplementary Fig. 18). The co-immunoprecipitation assays confirmed that KIN10 interacted with KINβ2 in plants, but this interaction was significantly reduced by $H_2O_2$ treatment (Fig. 5c, d). To determine whether $H_2O_2$ regulates the nuclear localization of KIN10 through SnRK1 β subunits, we expressed KIN10 and KINβ2 in the epidermal cells of the tobacco leaves with or without $H_2O_2$ treatment. The results showed that KIN10-YFP distributed in the nucleus and cytoplasm, co-expression with KINβ2 resulted in the exclusion of KIN10-YFP from the nucleus, and $H_2O_2$

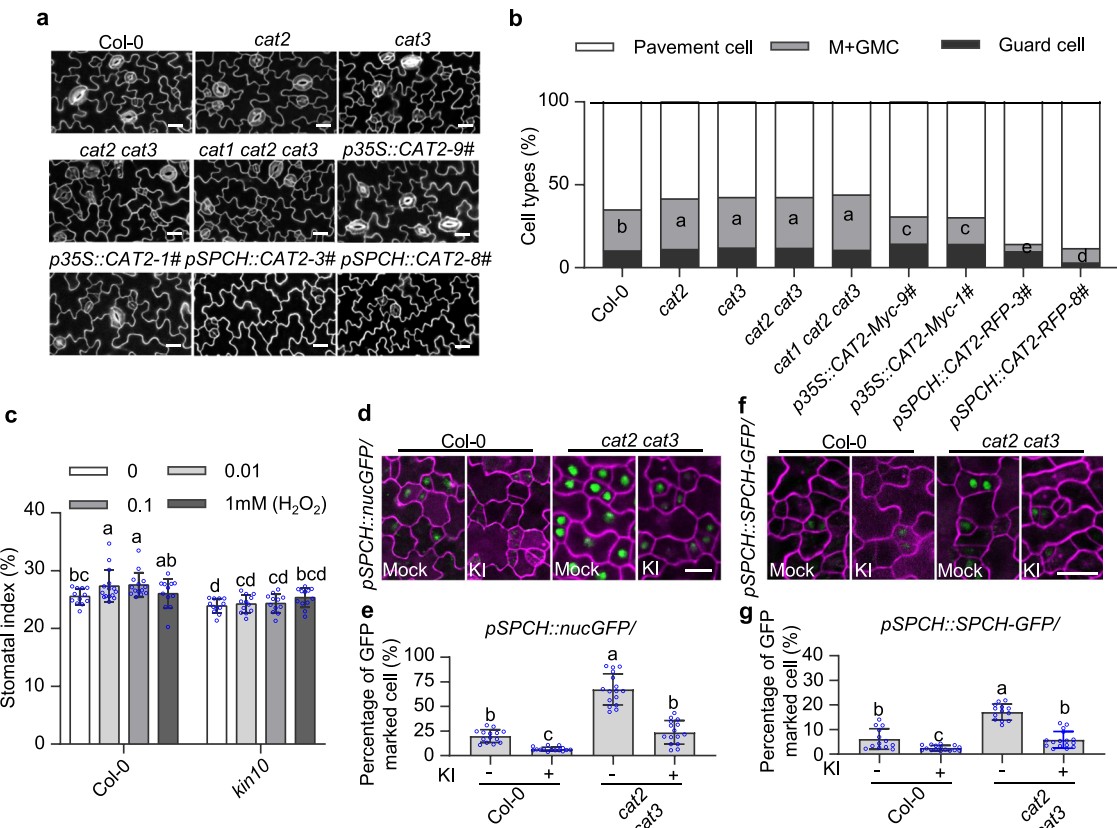

**Fig. 4 | H₂O₂ promotes stomatal development. a**, **b** Quantification of epidermal cell types of Col-0 and indicated plants, expressed as percentage of total cells. GMC, guard mother cell; M, meristemoid. $n = 13, 14, 14, 14, 14, 14, 15, 15, 12$ independent cotyledons were examined in **b**. Seedlings of wild-type Col-0 and indicated plants were grown on ½ MS solid medium containing 1% sucrose under 16 h light/8 h dark photoperiod with 100 μMol/m²/s for 5 days. **c**, Quantification of the effects of H₂O₂ on stomatal index. $n = 12, 15, 14, 12$ (Col-0), $n = 12$ (*kin10*) independent plants were examined in **c**. Seedlings of wild-type Col-0 and *kin10* mutant were grown on ½ MS solid medium containing 1% sucrose and different concentrations of H₂O₂ under 16 h light/8 h dark photoperiod with 100 μMol/m²/s for 8 days. **d**–**g** *CAT* mutations altered cell fate in Arabidopsis epidermis. $n = 14, 15, 16, 15$ independent cotyledons were examined in **e**. $n = 13, 16, 13, 16$ independent cotyledons

were examined in **g**. Seedlings of *pSPCH::nucGFP*, *pSPCH::nucGFP/cat2 cat3*, *pSPCH::SPCH-GFP*, and *pSPCH::SPCH-GFP/cat2 cat3* were grown on ½ MS solid medium containing 1% sucrose with or without 1 mM KI under 16 h light/8 h dark photoperiod with 100 μMol/m²/s for 3 days. Error bars indicate standard deviation (S.D.). Different letters above the bars indicated statistically significant differences between the samples (One-way ANOVA analysis (**b**), Two-way ANOVA analysis (**c**) followed by Uncorrected Fisher's LSD multiple comparisons test, $p < 0.05$; No adjustments were made for multiple comparisons test; Brown-Forsythe ANOVA analysis followed by Dunnett's T3 multiple comparisons test, $p < 0.05$ (**e**, **g**). Adjustments were made for multiple comparisons test). Scale bars in confocal images represent 20 μm.

treatment promoted the nuclear localization of KIN10 (Fig. 5e, f). Consistent with this, the ratio of nuclear-to-cytoplasmic KIN10 protein was significantly decreased in *p35S::KINβ2-RFP* transgenic plants, but increased in *kinβ2* mutant, and H₂O₂ treatment significantly increased the nuclear-to-cytoplasmic ratio of KIN10 in plants (Fig. 5g, h).

To understand the function of SnRK1 β subunits in the stomatal development, we analyzed the stomatal development phenotypes of wild type, SnRK1 β subunit loss-of-function and overexpression plants. The results showed that the *kinβ2* single loss-of-function mutants displayed the slightly increased stomatal index, but simultaneous mutation of *KINβ1* and *KINβ2* resulted in the significantly increased stomatal index (Fig. 5i, j). Overexpression of *KINβ2* exhibited the decreased stomatal index (Fig. 5i, j). These results indicate that SnRK1 β subunits negatively regulate the stomatal development.

## Discussion

As the major epidermal structure that facilitates gas exchange between plants and the atmosphere, the formation of stomata is controlled by multiple developmental and environmental signals. Here, we showed that the spatially patterned redox signal, H₂O₂, orchestrates stomatal development in Arabidopsis. SPCH directly binds to the promoters of *CAT2* and *APX1* to reduce their expression and increase H₂O₂ levels in

meristemoid cells. H₂O₂ reduces the interaction between KIN10 and KINβ2, thereby promoting the nuclear localization of KIN10. Nuclear-localized KIN10 phosphorylates and stabilizes SPCH, and subsequently promotes stomatal development. Our results provide a mechanistic framework for H₂O₂-mediated control of epidermal cell differentiation in Arabidopsis leaves (Fig. 5k).

The fine-controlled gradient distribution patterns of H₂O₂ and O₂·⁻ are critical for the balance between plant stem cell maintenance, proliferation, and differentiation[13]. H₂O₂ promotes plant stem cell differentiation, and O₂·⁻ is the key signaling molecule for plant stem cell maintenance. In the shoot apex, O₂·⁻ highly accumulates in the central zone to maintain stemness, and H₂O₂ accumulates in the peripheral zone to promote stem cell differentiation[15]. In the Arabidopsis root tip, H₂O₂ mainly accumulates in the expanding cells of the elongation zone, whereas O₂·⁻ mainly accumulates in the dividing cells of meristem zone[14]. However, in the present study, we only observed the specific accumulation of H₂O₂ in meristemoid cells, and no specific spatial distribution pattern of O₂·⁻ in the leaf epidermal cells. One possibility for the different distribution patterns of H₂O₂ and O₂·⁻ in the leaf epidermal cells is that there are different organizational structure and different regulatory mechanisms in the leaf epidermal stem cells compared with shoot and root apical meristems. The shoot or root

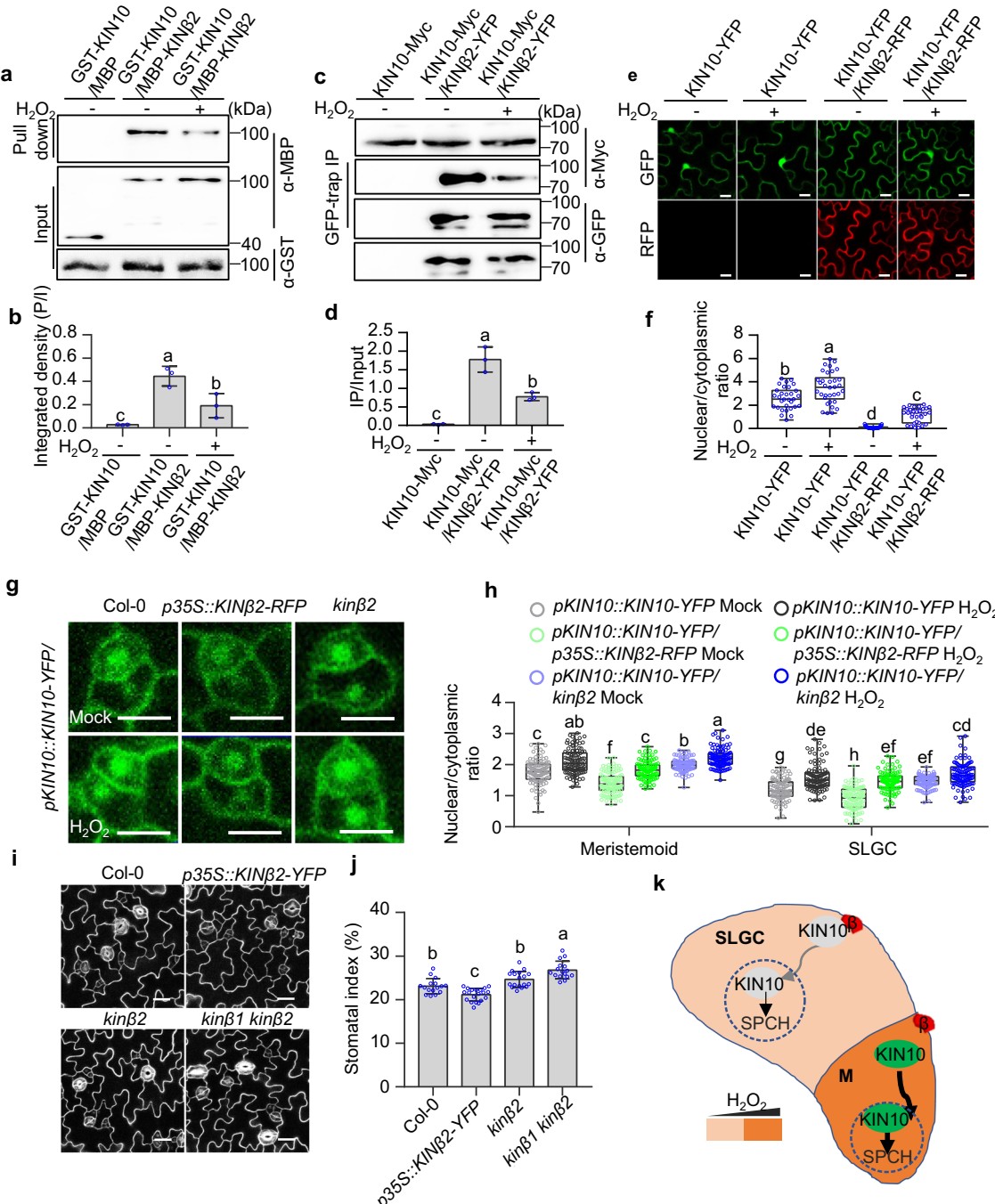

**Fig. 5 | H₂O₂ induces KIN10 nuclear translocation through reducing the interaction between KIN10 and KINβ2. a–d** H₂O₂ reduced the interaction between KIN10 and KINβ2 in vitro and in vivo. Error bars represent the SD of three independent experiments. **e–f** H₂O₂ reduces the effects of KINβ2 on the KIN10-cytoplasmic retention in tobacco leaves. The tobacco leaves were transformed with KIN10-GFP or co-transformed with KIN10-GFP and KINβ2-RFP. After 2 days, the leaves were treated with or without 2 mM H₂O₂ for 6 h. *n* = 31, 34, 33, 31 independent cells were examined in **f**. The fluorescent signals of GFP (KIN10) were determined using ImageJ software. **g, h** H₂O₂ reduces the effects of KINβ2 on the KIN10-cytoplasmic retention in Arabidopsis leaves. *n* = 108 (Col-0-Mock or H₂O₂), *n* = 110 (*p35S::KINβ2-RFP*-Mock or H₂O₂), and *n* = 108 (*kinβ2*-Mock or H₂O₂) meristemoid or SLGC cells in 10 cotyledons were examined in **h**. The box plot shows maxima, first quartile, median, third quartile, minima. Seedlings of *pKIN10::KIN10-YFP*, *pKIN10::KIN10-YFP/kinβ2* and *pKIN10::KIN10-YFP/p35S::KINβ2-RFP* were grown on ½ MS solid medium containing 1% sucrose under 16 h light/8 h dark photoperiod with 100 μMol/m²/s for 4 days, and then treated with H₂O₂ (Mock) or 2 mM H₂O₂ for 3 h. Serial Z-stack projection images were used for quantitative analysis.

**i, j** Overexpression of *KINβ2* inhibits stomatal development. *n* = 16, 20, 18, 16 independent cotyledons were examined in **j**. Seedlings of wild type, *kinβ2*, *kinβ1 kinβ2*, and *p35S::KINβ2-YFP* were grown on ½ MS solid medium containing 1% sucrose under 16 h light/8 h dark photoperiod with 100 μMol/m²/s for 5 days (**i**) or 8 days (**j**). Error bars indicate standard deviation (SD). **k** A working model for the function of hydrogen peroxide on stomatal development in Arabidopsis. SPCH directly binds the promoters of *CAT2* and *APX1* to reduce their expression, resulting in the high level of H₂O₂ in the meristemoids. The higher H₂O₂ in meristemoids induced the nuclear localization of KIN10 by reducing the interaction between KIN10 and KINβ2, thereby stabilized SPCH to promote stomatal development. Different letters above the bars indicated statistically significant differences between the samples (One-way ANOVA analysis followed by Tukey's multiple comparisons test, *p* < 0.05 (**b, d, j**); Brown-Forsythe ANOVA analysis followed by Dunnett's T3 multiple comparisons test, *p* < 0.05 (**f, h**)). Adjustments were made for multiple comparisons test. Scale bars in confocal images represent 10 μm (**g**) or 20 μm (**e, i**).

apical meristems are organized with the slowly dividing cell zones, such as the central zone at shoot and the quiescent center at root, and the differentiated amplifying cell zones such as the peripheral zone of shoot and the division zone of root. In the epidermal cells of leaves, there is no specific organ that functions similar to the central zone or quiescent center with slow cell division, resulting in no specific enrichment of $O_2^{\cdot-}$ in leaf epidermal cells.

The spatial distribution of $H_2O_2$ in meristems is essential for the stem cell differentiation in plants[14,15,39,40]. Here, we showed that $H_2O_2$ is abundantly enriched in meristemoid cells to promote stomatal development. The Arabidopsis seeds contain a lot of triacylglycerols, which are hydrolyzed during seed germination to provide carbon skeletons and energy for early seedling growth, whereas a large amount of $H_2O_2$ is generated as a by-product of fatty acid β-oxidation[41–44]. In addition, the plant shoot apical meristem is embedded in a low-oxygen niche and new leaves are produced under the hypoxic condition. When the new leaves emerge from the stem, they encounter an aerobic environment, thereby producing massive amounts of $H_2O_2$. Catalases and ascorbate peroxidase are key $H_2O_2$ scavenging enzymes in plants, and play important roles in plant antioxidative and detoxification processes that are closely correlated with ROS generation during plant growth and stress responses. Our results showed that $CAT2$ and $APX1$ are widely expressed in most cells of leaf epidermis except meristemoid cells. SPCH directly binds to the promoters of $CAT2$ and $APX1$ to repress their expression in meristemoid cells, resulting in the reduced $H_2O_2$ content in meristemoid cells. UPB1 was previously identified as a key transcription factor to repress a set of peroxidase expression in root. The accumulated $H_2O_2$ in meristemoids induces the nuclear localization of KIN10, which phosphorylates and stabilizes SPCH to promote stomatal development, thereby forming a positive feedback loop to control stomatal development.

$H_2O_2$ has been reported to promote stomatal closure by increasing the concentrations of cytosolic free calcium. The plant stress hormone abscisic acid (ABA) as well as diverse biotic and abiotic stress stimulated $H_2O_2$ accumulation, and subsequently induced stomatal closure, which suggests that $H_2O_2$ acts as a negative regulator for stomata to inhibit their function on gas exchange between plants and the atmosphere. In the present study, we showed that $H_2O_2$ positively regulates stomatal development. This opposite regulation on stomatal functions by $H_2O_2$ may be due to the different effects of different $H_2O_2$ contents on stomatal development and stomatal movement. The content of $H_2O_2$ in guard cells is approximately 10 times higher than that in meristemoid cells when plants were grown under normal conditions. ABA treatment significantly increase in the $H_2O_2$ content in the guard cells, suggesting that the concentrations of $H_2O_2$ in meristemoid cells that is required to promote stomatal development is at least 10 times lower than that in guard cells, which are responsible for promoting stomatal closure. Consistent with this, our results showed exogenous $H_2O_2$ treatment promoted stomatal development, but such promoting effects decreased as the increasing $H_2O_2$ concentrations. Furthermore, our previous study showed that $H_2O_2$ promoted stomatal opening in the intact leaves at the low concentrations[45]. The $rbohd\ rbohf$ mutant and $CATALASE2$ overexpression transgenic plants exhibited low $H_2O_2$ concentrations in guard cells and failed to open stomata normally[45]. These results indicate that $H_2O_2$ promotes the stomatal development and stomatal opening at low concentrations, thereby inducing efficient $CO_2$ uptake and the photosynthesis under the normal growth conditions; whereas $H_2O_2$ promotes stomatal closure at high concentrations, then reducing water loss and improving stress resistance of plant under stress conditions.

A recent study reported that plastoquinone oxidation inhibits stomatal development by negatively regulating $SPCH$ and $MUTE$ expression[46]. DCMU is a photosynthesis inhibitor, and can induce the oxidation of plastoquinone. DCMU treatment quickly induced the phosphorylation of MPK6 to activate MPK6. Activated MPK6 interacted and phosphorylated the epidermal specific expressed HD-ZIP transcription factor HDG2, and then inhibited the transcriptional activity of HDG2 and reduced the expression of $SPCH$ and $MUTE$, thereby inhibiting stomatal development. In the present study, we showed that short-term DCMU treatment induced the nuclear localization of KIN10, while the long-term DCMU treatment led to KIN10 protein degradation. DCMU treatment induced the accumulation of $H_2O_2$ that reduced the interaction between KIN10 and KINβ2 and promoted KIN10 nuclear localization. Cotreatment of KI and DCMU counteracted the promoting effects of DCMU on KIN10 nuclear localization, suggesting $H_2O_2$ is required for DCMU-induced nuclear localization of KIN10. However, DCMU also has been reported to have no significant effects on $H_2O_2$ accumulation[46–48]. One possibility for this inconsistency among different studies maybe the different inhibitory effects of DCMU on photosynthesis in different studies. Plastoquinone functions not only an essential component of photosynthesis to transport electrons, but also acts as a potent antioxidant to regulate the state transitions and gene expression[49]. Treatment with short-term and low-dose DCMU leads to plastoquinone oxidation, while long-term and high-dose DCMU treatment leads to plastoquinone oxidation and $H_2O_2$ generation, and much high-dose DCMU treatment leads to higher $H_2O_2$ accumulation and programmed cell death.

SnRK1 is a central metabolic regulator of energy homeostasis and plays important roles in regulating plant growth and development in response to energy status[19,20]. Our previous work showed that when plants were grown under mild energy starvation conditions, such as short-day photoperiod or liquid cultures, sucrose supply induced the protein accumulation of KIN10, which phosphorylated and stabilized SPCH to promote stomatal development[11]. Such promoting effects of sucrose and KIN10 on stomatal development were significantly reduced when plants were grown under high light irradiance or long-term photoperiod conditions compared to plants grown under low light irradiance or short-term photoperiod conditions, suggesting the low light irradiance or short-term photoperiod specifically activate KIN10 through an unknown mechanism[11]. In a recent study, we showed that sucrose dynamically regulates stomatal development through fine-tuning KIN10 activity. Sucrose induced the total KIN10 protein accumulation at all tested concentrations, but only induced the phosphorylated KIN10 protein accumulation at the low concentrations. Under high sucrose concentrations, sucrose supply triggers the accumulation of trehalose-6-phophate, which represses KIN10 activity by reducing the interaction between KIN10 and its upstream kinase SnAK1/2. Consistent with this, sucrose promoted stomatal development at low concentrations, but such promoting activity decreased with the increasing sucrose concentrations, suggesting phosphorylation is an important regulatory mechanism for KIN10-mediated stomatal development[50]. Here, in this study, we showed that $H_2O_2$ induced the nuclear localization of KIN10 by reducing the interaction between KIN10 and KINβ2. In the epidermal cells of Arabidopsis leaves, $H_2O_2$ specifically accumulated in the meristemoid cells, which induces the nuclear localization of KIN10. Disruption of the specific spatial distribution of $H_2O_2$ in the epidermal cells using $SPCH$ promoter to drive $CAT2$ expression significantly inhibits stomatal development, suggesting $H_2O_2$-mediated KIN10 nuclear localization plays critical roles for KIN10-mediated stomatal development. Taken together, sucrose-mediated KIN10 protein accumulation, $H_2O_2$-mediated KIN10 nuclear localization, and KIN10 phosphorylation, these three regulator mechanisms finely regulate KIN10 activity to optimize stomatal development in response to diverse environmental and developmental signals.

## Methods

### Plant materials and growth conditions

All wild-type, various mutants, and transgenic plants in this study are in Col-0 ecotype background. Mutants and transgenic plants used in this study include $p35S::KIN10\text{-}Myc$[11], $pKIN10::KIN10\text{-}Myc$[11], $pKIN10::KIN10\text{-}$

*YFP*[11], *kin10*[11], *pCAT2::GFP*, *pAPX1::GFP*, *cat2*, *cat2 cat3*, *cat1 cat2 cat3*[32], *p35S::CAT2-Myc*, *pSPCH::nuc-GFP*[5], *pSPCH::SPCH-GFP*[5], *pSPCH::SPCH-RFP*[11], *pSPCH::SPCH-RFP/spch-4 (SPCH-8#)*[11], *pSPCH::SPCH$^{S4A}$-RFP/spch-4 (4A-2#)*[11], *p35S::SPCH-Myc*[11], *p35S::SPCH-4A-Myc*[11], *pSPCH::CAT2-RFP*, *p35S::UPB1*, *pKINβ2::KINβ2-YFP*, and *p35S::KINβ2-YFP*. The T-DNA insertion mutants of SALK_076998 (*cat2*), SALK_092911 (*cat3*), SALK_000249 (*apx1*), SALK_115536 (*upb1*), SALK_008325 (*kinβ1-1*), and SALK_052521 (*kinβ2*) were ordered from the Arabidopsis Biological Resource Center. Plants were grown in a greenhouse with white light at 100 μMol/m²/s under a 16 h light/8 h dark photoperiod at 22 °C for general growth. For chemical treatment, the 4-day-old general growth seedlings of the indicated plants treated with or without DMSO (Mock), 50 μM DCMU and/or 1 mM KI for 12 h.

## Plasmid constructs and transgenic plants
Full-length coding region of *KINβ2* and *CAT2* without stop codon were amplified by PCR and cloned into pENTR™/SD/D-TOPO™ vectors (Thermo Fisher), and then recombined with destination vector pX-YFP (*p35S::X-YFP*), p1390-MH (*p35S::X-Myc-His*), and pAL-1296R (Native promoter::RFP), and pMAL2CGW (N-MBP). The promoters of *CAT2* and *APX1*, the promoter and genomic DNA of *KINβ2*, were amplified by PCR and cloned into pENTR™/SD/D-TOPO™ vector (Thermo Fisher), and then recombined with destination vector pEG-TW1 (Native promoter::YFP) to generate *pCAT2::GFP* and *pAPX1::GFP*, respectively. HyPer is a genetically encoded yellow fluorescent protein (YFP)-based $H_2O_2$ sensor, which is highly sensitive to $H_2O_2$ and not to other ROS in bacteria and animal cells[30]. To investigate $H_2O_2$ metabolism in plants, we amplified the HyPer from the pHyper-cyto plasmid (Evrogen, FP941) and cloned into pENTR™/SD/D-TOPO™ vectors, and then recombined with destination vector pEARLY 100 (*p35S::X*). This construct was introduced into *Agrobacterium tumefaciens* (strain GV3101), and transformed into Col-0 plants to generate *p35S::HyPer* transgenic plants. Fluorescence intensity of pHyPer transgenic plants was determined over regions of interest, at 530 nm after excitation at 488 nm or 405 nm. Emission was collected at 530 nm with a bandpass of 30 nm. Image analysis was performed with an ImageJ software. A fluorescence ratio was calculated as 488/405. Oligo primers used in this study are listed in Supplementary Data 1. All binary vector constructs were introduced into *Agrobacterium tumefaciens* (strain GV3101), and transformed into Col-0 plants by the floral dipping method[51].

## Fluorescence imaging of ROS
The H₂DCFDA staining assay was performed slightly modified as described previously[52]. Briefly, seedlings were incubated in 10 μM H₂DCFDA in 10 mM Tris-HCl (pH 7.2) for 20 min in the dark. For BES-$H_2O_2$-Ac staining, seedlings were infiltrated with BES-$H_2O_2$-Ac solution (50 μM BES-$H_2O_2$-Ac in ½ MS liquid medium) and incubated at room temperature for 1 h in the dark. Excess H₂DCFDA and BES-$H_2O_2$-Ac were removed by washing 5 times with 10 mM Tris-HCl (pH 7.2) and ½ MS liquid medium respectively. H₂DCFDA, BES-$H_2O_2$-Ac and Propidium Iodide (PI) staining images were captured on a Zeiss LSM700 laser scanning confocal microscope.

## DAB staining
2 mg/ml 3,3′-diaminobenzidine (DAB) (Sigma-Aldrich) dissolved in 10 mM disodium hydrogen phosphate (pH 6.5) and were incubated in the dark until a reddish-brown precipitate was observed. The staining was terminated by transferring the plants into the solution (ethanol: glycerin: acetic acid = 3:1:1). The material was then imaged under a light microscope. The midleaf region of the cotyledon was used to analyze the $H_2O_2$ content by Scion Image software.

## Quantitative analysis of fluorescent signals of KIN10-YFP
To capture the fluorescent signal, a series of Z-stack confocal images covering the entire epidermal cells (7 layers) were subjected to surface rendering in green channel to capture *pKIN10::KIN10-YFP*-expressing nuclei and cytoplasm in the three-dimensional space (for each time point). The Z-stack confocal images were taken using LSM880 microscope from Zeiss. A cut-off value of 0.82 was set for sphericity, which effectively removed objects with non-specific signals. Quantitative analyses of KIN10-YFP nuclear/cytoplasmic ratio were determined along a line drawn on the confocal images using ImageJ software and nuclear and cytoplasmic KIN10-YFP signals from more than 200 stomatal lineage cells in 10 cotyledons were analyzed. All graphs and all statistical analysis were generated using GraphPad Prism 9.0.

## Stomatal quantification
Arabidopsis seedlings for stomatal quantification were cleared by decolor solution (75% ethanol: 25% acetic acid) until the leaf were no longer green. Cotyledons were dried on absorbent paper, then submerged into Hoyer's solution until cotyledon turned transparent completely for microscope observation. Images were captured per cotyledon from central regions of abaxial leaves. Guard cells were counted for stoma index calculation.

## Protein-protein pull-down assays
KIN10 fused to GST were purified from bacteria using glutathione beads (GE Healthcare). KINβ2 fused to MBP were purified using amylose resin (New England Biolabs). Glutathione beads containing 1 μg of GST-KIN10 were incubated with 1 μg MBP or MBP-KINβ2 in pull-down buffer (20 mM Tris-HCl, pH 7.5; 100 mM NaCl; 1 mM EDTA), with or without 2 mM $H_2O_2$ at room temperature for 1 h and the beads were washed 4 times with wash buffer (20 mM Tris-HCl, pH 7.5; 300 mM NaCl; 1 mM EDTA; 0.5% Triton X-100). The eluted proteins were analyzed by immunoblot analysis with an anti-MBP antibody (TransGen Biotech, Cat: HT701-01,1:5,000 dilution) and anti-GST antibodies (TransGen Biotech, Cat: HT601-01,1:5,000 dilution).

## DNA-protein pull-down assays
The recombinant proteins MBP and MBP-SPCH were expressed and affinity purified from *Escherichia coli* using amylose resin(TransGen Biotech, Cat: HT701-01,1:5000 dilution). The fragments of *CAT2* and *APX1* promoter were amplified by PCR using the biotin-labeled primers (Supplementary Data 1). The DNA and proteins were incubated, and then DNA-binding proteins were pull down using streptavidin-agarose beads and immunoblotted with anti-MBP antibodies(TransGen Biotech, Cat: HT701-01,1:5000 dilution).

## Co-immunoprecipitation assays
Arabidopsis seedlings expressing *pKIN10::KIN10-Myc* only or co-expressing *pKIN10::KIN10-Myc* and *pKINβ2::KINβ2-YFP* were grown on ½ MS solid medium supplemented with 1% sucrose for 10 days and then treated with or without 2 mM $H_2O_2$ for 3 h. Plant materials were harvested and ground in liquid nitrogen and then extracted in lysis buffer containing 20 mM HEPES-KOH, pH 7.5; 40 mM KCl; 1 mM EDTA; 0.5% Triton X-100; 1 mM PMSF and 1× protease inhibitor cocktail (Sigma-Aldrich). After centrifugation at 4 °C, 12,000 × *g* for 10 min, the supernatant was incubated with GFP-Trap agarose beads (Chromotek) at 4 °C for 1 h, and the beads were washed 4 times using wash buffer (20 mM HEPES-KOH, pH 7.5; 40 mM KCl; 1 mM EDTA; 300 mM NaCl; and 1% Triton X-100). The proteins were eluted from the beads by boiling with 2× SDS sample buffer, analyzed by SDS-PAGE, transferred to nitrocellulose membrane and immunoblotted with anti-GFP (TransGen Biotech, Cat: N20610,1:5000 dilution) and anti-Myc (Sigma-Aldrich, Cat: M4439, 1:5000 dilution) antibodies.

## Chromatin immunoprecipitation (ChIP) Assay
Seedlings of *p35S::SPCH-YFP*, and *p35S::YFP* were grown on ½ MS solid medium supplemented with 1% Sucrose for 12 days under a long-day photoperiod, harvested, and cross-linked in 1% formaldehyde for

30 min under vacuum. Immunoprecipitation was performed as previously described[53], using GFP-Trap agarose beads (Chromotek). ChIP products were analyzed by qPCR, and the fold enrichment was calculated as the ratio between *p35S::SPCH-YFP*, and *p35S::YFP*, and then normalized by *PP2A* (*At1g13320*) gene, which was used as an internal control. The ChIP experiments were performed with three biological replicates, from which the means and standard deviations were calculated.

### RT-qPCR analysis

Seedlings for wild type and indicated plants were grown on ½ MS solid medium supplemented with 1% sucrose for 6 days under a long-day photoperiod, harvested to extract total RNA (TransGen Biotech). First-strand cDNAs were synthesized using RevertAid reverse transcriptase (Thermo Fisher Scientific) and used as qPCR templates. Quantitative PCR analysis was performed on a CFX connect real-time PCR detection system (Bio-Rad) using synergy brands (SYBR) green reagent (Roche) with gene-specific primers (Supplementary Data 1).

### Reporting summary

Further information on research design is available in the Nature Research Reporting Summary linked to this article.

## Data availability

All data in this study are available in the manuscript or the Supplementary materials. Source data are provided with this paper.

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

## Acknowledgements
We thank Prof. Dominique C. Bergmann and Prof. Jie Le for providing the seeds of *pSPCH::nucGFP, pSPCH::SPCH-GFP*, Prof. Changle Ma for providing the seeds of *cat1 cat2 cat3*, and Prof. Zhong Zhao for providing the seeds of *ndufs4*. We thank Haiyan Yu and Xiaomin Zhao from the Analysis and Testing Center of SKLMT (State Key Laboratory of Microbial Technology, Shandong University) for assistance with the laser scanning confocal microscopy. This work was funded by the National Natural Science Foundation of China (grant nos. 31870262, 31970306, and 32070210) and by the Shandong Province Natural Science Foundation (grant nos. ZR2019ZD16 and 2019LZGC-015).

## Author contributions
W.S. and M.B. together designed the experiments. W.S. and L.W. performed the ROS staining, microscopy analysis, stomatal index analysis, western blot, subcellular location analysis, pull down, CoIP assay. W.S., L.Y., W.H., C.H., and M.F. generated *pCAT2::GFP, pAPX1::GFP, pSPCH::CAT2-RFP, kinβ1 kinβ2, p35S::KINβ2-YFP*. C.H. and W.W provided the critical discussion on the work. W.S. performed all other experiments. W.S. and M.B. wrote the manuscript.

## Competing interests
The authors declare no competing interests.
