## [Peer Review File · Nature Communications]

Spatially patterned hydrogen peroxide orchestrates stomatal development in ArabidopsisREVIEWER COMMENTS

Reviewer #1 (Remarks to the Author):

Authors showed previously that KIN10, a catalytic α -subunit of energy homeostasis controlling kinase SnRK1, accumulates to nucleus in stomatal precursor cells and thereby promotes stomatal development (Han et al., 2020). This regulatory circuit was shown to operate only upon carbon starvation and in sucrose dependent fashion. In their current work Shi et al. suggest that cell type specific H₂O₂ levels regulate subcellular localization of KIN10 and thus control stomatal development. Blanco et al. (2019) showed previously that treatment with a herbicide DCMU is sufficient to change subcellular localization of KIN10 and promote its nuclear localization in the root. Shi et al. study effects of DCMU and notice that it promotes nuclear localization of KIN10 in cotyledon epidermis. DCMU specifically blocks electron transfer downstream of PSII, however, authors suggest that effect of DCMU on KIN10 localization may be caused by metabolic stress induced H₂O₂ production. Effect of DCMU on KIN10 localization can be reversed by Potassium Iodide treatment, quencher of H₂O₂, supporting this idea. Further, authors show that also H₂O₂ treatment promotes nuclear localization of KIN10 and further, leads to increased stomatal index (ratio of stomata to total epidermal cells) indicating that subset of asymmetric division types in the stomatal lineage might be affected. Authors show that stomatal precursor cells, meristemoids, show high H₂O₂ levels and that various mutants with high and low H₂O₂ levels show reduced altered H₂O₂ levels specifically in the meristemoids. Authors find that the genes encoding ROS- scavenging enzymes, CATALASE2 (CAT2) and ASCORBATE PEROXIDASE 1 (APX1) show low expression levels in meristemoids. Interestingly, the master regulator of stomatal lineage initiation, SPEECHLESS (SPCH), has been previously shown to bind promoter regions of CAT2 and APX1 (Lau et al., 2014). Authors confirm this interaction by quantitative ChIP-PCR and hypothesize that SPCH may directly regulate expression of CAT2 and APX1 in meristemoids, and this regulation may be repressive in nature. Loss of CAT or APX1 leads to elevated stomatal index and *cat2 cat3* mutants shows dramatically increased translational and transcriptional SPCH expression. Finally, authors show that KIN γ 2 and KIN10 interaction (which has been previously shown to reduce nuclear localization of KIN10 by Ramon et al. (2019)) is repressed by H₂O₂. This data shows that meristemoid specific high H₂O₂ levels further enhances nuclear localization of KIN10 and thus promote stomatal development.

This work contains novel and interesting findings. Authors show that ROS plays a role in stomatal development and refine the regulatory scheme they previously identified. Major weakness of this manuscript is variable description of experimental conditions. It would have been especially important since this work is continuation of the previous work (Han et al., 2020) where authors showed that in the nucleus, KIN10 phosphorylates SPCH and this interaction leads to stabilization of SPCH and promotion of stomatal development only upon carbon starvation. In the current version of the manuscript, information on used experimental conditions is limited and in addition, several methods are poorly described and therefore it is hard to follow what has been done. It is also problematic, that it is not discussed whether the proposed model operated only upon carbon starvation. I will specify these aspect and my other concerns below.

Major comments

1. Growth conditions and data interpretation

Authors showed previously that *kin10* mutant shows consistent response to sucrose only when grown in darkness or liquid growth media, however, response becomes inconsistent when grown on solid media or different light conditions (Han et al., 2020; Figure 2c and Supplementary Figure 5). Therefore, it is unfortunate that authors do not specify clearly in their current work on KIN10 did they use solid or liquid growth media and what were the other growth conditions. Indication of the use of sucrose seems random, which is unfortunate, since sucrose modifies molecular interaction in the studied system. In few cases it has been stated that solid media was used, however, it is not explained why authors choose to use in these particular experiments different conditions. It is difficult

to define what are the used standard conditions.

Specific comments to figures:

- Figure 1: What is "medium" – liquid or solid? Is some information missing or were these experiment done with different conditions? For example in Figure 1 b-d: 1% sucrose was used; Figure 1 e-g: sucrose is not mentioned – is this correct? Also, day length is mentioned only for Figure 1 h-l – is used day length same in all the experiments?
- Figure 4: No growth conditions are mentioned for Figure 4a & 4b, however, Figure 4c label indicates that ½ MS solid medium has been used but no sucrose mentioned. Is this correct information? If it is, what was then the agar concentration used? Please indicate why did you choose to change growth conditions only for this particular experiment where you show that kin10 does not respond to H₂O₂?
- Figure 5: c-d no sucrose mentioned - Is this correct information?
- Supplementary fig 1: no sucrose mentioned - correct?
- Supplementary fig 2 c-d: no sucrose mentioned - correct?
- Supplementary fig 5: no sucrose mentioned - correct?
- Supplementary fig 8: solid medium, no sucrose - correct?

Please describe all standard growth conditions carefully in the Materials and methods and if there are any exceptions, clearly indicate this case by case and explain why different growth conditions were used.

Authors state in the discussion:

"Our previous study showed that the KIN10, the α -catalytic subunit of SnRK1, specifically locates in the nucleus of meristemoids to phosphorylate and stabilize SPCH, and then promoting stomatal development under carbon starvation conditions. Here, we showed that H₂O₂ is abundantly enriched in meristemoid to promote the nuclear localization of KIN10 by reducing the interaction between KIN10 and its regulatory subunit KIN β 2."

Please discuss your findings in the context of the previously identified regulatory scheme and clearly state, whether or not the H₂O₂ mediated regulation operates only upon hypoxia causing conditions.

2. DCMU treatment

Recent paper by Zoulias et al. (Curr Biol, 2021) showed that DCMU treatment causes decrease in SPCH levels and reduces stomatal index. Here Shi et al., show that DCMU enhance nuclear localization of KIN10 and that this DCMU effect on KIN10 localization is caused mainly by increased accumulation of H₂O₂. However, plants treated with exogenous H₂O₂ or mutants with elevated H₂O₂ levels show increased stomatal index, which is opposite to DCMU caused phenotype described by Zoulias et al. In addition to the differences observed in the phenotype, Zoulias et al. also come up with completely different regulatory circuits upstream of SPCH. Interestingly, Zoulias et al., use milder DCMU treatment: single spray treatment with 10 μ M DCMU whereas Shi et al., used 12 hour treatment with 50 μ M DCMU. It is possible that this milder short-term treatment causes different effect at the molecular level and that other differences in the experimental set up may also play a role.

12h 50 μ M DCMU treatment seems strong – does it affect to the cell area or leaf size? Although DCMU data is not central to this manuscript, it would be interesting to see, whether milder treatment, such as single spray event with 10 μ M DCMU, will cause similar effect on KIN10 localization than 12 hour treatment with 50 μ M DCMU. Please also refer to the work by Zoulias et al. and discuss, why DCMU and H₂O₂ treatments in these two studies may cause different effects on stomatal index.

3. Effect of H₂O₂ on SPCH expression

Here authors show that DCMU enhanced nuclear localization of KIN10 and accumulation of H₂O₂. Mutants with elevated H₂O₂ levels show increased SI and more transcriptional and translational SPCH signal. Please explain how the effect of high H₂O₂ on SPCH transcription is mediated in their model?

4. KIN10- SPCH interaction upon H₂O₂ treatment

Authors have previously established variants of SPCH with altered KIN10 phosphorylation sites (Han et al., 2020). These variants would be excellent backgrounds to test the proposed model. Please test whether or not stomatal index and SPCH protein stability are affected in these lines and wild-type as a response to exogenous H₂O₂ treatment.

5. KIN10 localization in kin β 2

Row 373: "H₂O₂ reduces the interaction between KIN10 and KIN β 2, thereby promoting the nuclear localization of KIN10". Currently this is not shown. It would be informative to see KIN10 localization in the kin β 2 and p35S-kin β 2-YFP backgrounds to support this statement.

6. Discussion

Currently big part of the discussion (rows 399-413 and 429-434) is used to describe additional data. If this data is central to this manuscript, describe the data in the Results section and update the manuscript with all the appropriate information related to these experiments. If this data is not central, please remove the data.

Minor comments

7. Figure 1: It seem that DCMU + KI and CAT-OE repress strongly KIN10 expression – do you see this constantly in your experiments? What may be cause for this?

8. Please describe difference between H₂DCFDA and BES-H₂O₂-Ac dyes and what exactly do they indicate? Add also reference where they have been used before.

9. Please write the gene names open for all genes when you mention them first time and also refer to the original paper describing the gene for the first time (for example UPB1, CAT, APX1)

10. Row 222: Authors may mean positive relationship (larger the cell the more there is signal), not negative relationship. If this is correct, please modify to be in line with the data.

12. Row 256: "These results indicated that SPCH directly bound to the promoters of ROS scavenging genes CAT2 and APX1 to repress their expression and promote H₂O₂ accumulation in meristemoids." The data currently shown does not show that SPCH represses CAT2 and APX1. Please modify if additional support for the direct repression is not presented.

13. Figure 4e: 90% SPCH expressing cells seems incredible high number of cells for any plant and especially considering the mild phenotype of this particular mutant. Is this correct? Also consider whether the number 120% on Y-axis is meaningful.

14. Row 307: H₂O₂ may not alter epidermal fates but increases frequency of stomata producing divisions. Please modify.

15. Description of pharmacological treatments, staining methods and measurement methods

Please add and improve Materials and Methods so that all the used methods are included so that experiments can be easily repeated.

- Please describe DMCU and all other treatments in detail in the Materials & Methods. Describe also what were the mock treatments. Please add this information for all treatments.

- Please describe how DAB staining was done. Also, describe how did you measured and quantified DAB data in Supplementary Figure 1 c-d? (veins show very strong signal compared to the other parts of the leaf – which regions were used?)

- Please describe the source of HyPer line and imaging set up in Materials and Methods.

- Description of the nuclear/cytoplasmic ratio measurement is quite ok (Figure legend 1), however, it would be easier to find it from Materials and Methods. So please consider transferring this information there. Please also consider how this measuring method fits to the idea that KIN10 is localized to the

ER as described by Blanco et al., 2019.

- Please describe how cell area/fluorescent intensity was measured (For example presented in the Supplementary figure 2)

Reviewer #2 (Remarks to the Author):

The work performed by Shi and coauthors demonstrated that spatially patterned hydrogen peroxide (H₂O₂) plays an essential role in stomatal development. The study is interesting and addresses important aspects related to the involvement of redox-related signals in the development of stomata in Arabidopsis cotyledons. The experiments appear to have been well executed and the results are clear. The manuscript is very well written and the obtained results support the main conclusions of this paper. However, the authors should consider the following points:

- 1) The authors tested the spatial distribution of H₂O₂ and O₂ in epidermal cells and stomatal development. However, it is not clear from the manuscript whether other redox-related signals would play similar roles in stomatal development using the same mechanism.
- 2) The conclusions of this work are based on experiments carried out in cotyledons of Arabidopsis seedlings. The article would benefit from the results of experiments carried out on young leaves. It would be important to indicate whether the proposed mechanism is also valid for the final stomatal density observed in adult plants.
- 3) The manuscript does not present any discussion about the interactions that the proposed mechanism may have with environmental factors. Specifically, the authors could address the following questions: Would abiotic stress conditions lead to some change in stomata density through changes in the levels of redox-related compounds in epidermal cells via the proposed mechanism? What is the level of involvement of hormones associated with the development of stomata, such as ABA?
- 4) The authors did not explain the source/sites of production of H₂O₂ in the meristemoid cells and why they performed most of the analyses in 4-day-old seedlings. A previous study demonstrated that "Following germination, Arabidopsis seeds rely on storage oil breakdown to supply carbon skeletons and energy for early seedling growth, and massive amounts of H₂O₂ are generated within the peroxisome as a by-product of fatty acid β -oxidation" (Eastmond et al 2007). Accordingly, previous studies show that much of the generation of ROS is formed as byproducts of the fatty acid β -oxidation, albeit additional sources of ROS do exist during early stages of development, such as the NADPH oxidases. Data from Fulda et al. 2004, Bernhardt et al. 2012 and Feitosa-Araujo et al 2020, among others, provided evidence that β -oxidation is strongly required in 4-day-old seedlings. The authors should contemplate these points in the discussion.

References:

Fulda M, Schnurr J, Abbadi A, Heinz E, Browse J. Peroxisomal Acyl-CoA synthetase activity is essential for seedling development in Arabidopsis thaliana. Plant Cell. 2004 Feb;16(2):394-405. doi: 10.1105/tpc.019646. Epub 2004 Jan 23. PMID: 14742880; PMCID: PMC341912.

Eastmond PJ. MONODEHYDROASCORBATE REDUCTASE4 is required for seed storage oil hydrolysis and postgerminative growth in Arabidopsis. Plant Cell. 2007 Apr;19(4):1376-87. doi: 10.1105/tpc.106.043992. Epub 2007 Apr 20. PMID: 17449810; PMCID: PMC1913749.

Bernhardt K, Wilkinson S, Weber AP, Linka N. A peroxisomal carrier delivers NAD⁺ and contributes to optimal fatty acid degradation during storage oil mobilization. Plant J. 2012 Jan;69(1):1-13. doi: 10.1111/j.1365-313X.2011.04775.x. Epub 2011 Oct 25. PMID: 21895810.

Feitosa-Araujo E, da Fonseca-Pereira P, Pena MM, Medeiros DB, Perez de Souza L, Yoshida T, Weber APM, Araújo WL, Fernie AR, Schwarzländer M, Nunes-Nesi A. Changes in intracellular NAD status

affect stomatal development in an abscisic acid-dependent manner. *Plant J.* 2020 Dec;104(5):1149-1168. doi: 10.1111/tpj.15000. Epub 2020 Nov 24. PMID: 32996222.

5) Why did the authors not consider SnRK2? According to a recent study published at *Nature Plants* by Belda-Palazón et al 2020, SnRK2 kinases have a dual role in the regulation of SnRK1 and plant growth.

Reference: Belda-Palazón, B., Adamo, M., Valerio, C. et al. A dual function of SnRK2 kinases in the regulation of SnRK1 and plant growth. *Nat. Plants* 6, 1345–1353 (2020).

<https://doi.org/10.1038/s41477-020-00778-w>

6) The authors mentioned in the description of the results that “catalase and UPB1 play important roles in H₂O₂ distribution pattern in epidermal cells of leaves”. However, at no time UPB1 was mentioned in the discussion. Specifically, the authors could provide a deeper and more integrated mechanism to be included in the discussion and they also could explain how UPB would participate in the observed responses.

7) The authors cited reference 29 to say that “We found that the expression levels of several ROS-scavenging genes, such as CAT2 and APX1, were lower in the meristemoid cells than those in other types of cells”. Where are the protein products of CAT2 and APX1 located? Did the authors or reference 29 demonstrate a change in the expression of other isoforms of catalase and ascorbate peroxidase? Which other ROS-scavenging genes showed a lower expression in meristemoid cells? The authors could provide in a supplemental table the identification (including predicted subcellular location) of the main ROS-scavenging genes (in addition to CAT2 and APX1) whose expression are reduced in meristemoid cells (as provided by reference 29). They should further justify why did they specifically choose CAT2 and APX among these genes in the list to test through the generation of transgenic plants.

8) Lines 384-392 of the discussion repeats in part some of the information already mentioned in the previous paragraph of the discussion. The authors could rework these sentences to avoid repetition.

Reviewer #3 (Remarks to the Author):

Reviewer 's Remarks to Authors:

Shi, Wang and coworkers have studied the link between the distribution of H₂O₂ in leaves of seedlings and the stomatal development in *Arabidopsis*. The authors found that the concentration of H₂O₂ is higher in meristemoids cells, which are the precursor cells of stomatal guard cells, than in pavement cells. This finding was in line with reduced activity of H₂O₂-scavenging enzymes like CATALASE2 and ASCORBATE PEROXIDASE1. Moreover, through fluorescence microscopy techniques and COIP, they demonstrated that SPEECHLESS (SPCH), a transcription factor associated with stomatal development, is the regulator of these enzymes and, then, responsive to H₂O₂ concentration pattern. Finally, and as a follow-up of a previous publication of the team, SnRK1.1, aka KIN10; is indicated as a key player of this developmental programme by stabilizing SPCH. The authors conclude that H₂O₂ pattern in epidermal cells is central to orchestrate the development of stomatal cells. This study pursues to shed light on the stomatal development process by identifying 2 of its key players, KIN10 and SPCH and conferring a pivotal role to the levels of H₂O₂ in their coordinated function. The article presents interesting results; for instance, revealing SPCH is able to bind to the CAT2 and APX1 promoters or the effect of mutations in KIN β subunits on the Stomatal index. As mentioned above, the rationale behind this work is to delve into the mechanistic aspects of the link between the energy gauge mediated by SnRK1.1 and modulation of stomatal development in response to changing environments previously informed by Han et al in this Journal last year (*NATURE COMMUNICATIONS* | (2020) 11:4214 | <https://doi.org/10.1038/s41467-020-18048-w>).

This reviewer thinks that the article is correctly presented and the topic is highly relevant by its potential to understand adaptive mechanisms involving the tuning of gas exchange in plants to

external conditions. The deepening in the comprehension of KIN10 and SPCH interaction provides knowledge to prime the design of new approaches to re-adapt photosynthesis in a current adverse climate scenario.

At the same time, some concerns detailed below point to two caveats of the work:

first, inaccurate assumptions regarding the pharmacological approaches and, second, the chosen quantification approach to analyse the observed phenomena is not robust enough to draw the presented conclusions.

The first issue is regarding the identification of H₂O₂ as the secondary messenger, whose levels are related to the enrichment of KIN10 in the nucleus. The authors performed a 12 hrs-DCMU treatment in ½ MS medium on 4 days seedling, and determined that a) KIN10 presents a mostly nuclear localization in meristemoid cells and Stomatal lineage ground cells, while in pavement cells is mainly localized in the cytoplasm, b) DCMU triggers an increase in the nuclear fraction of KIN10 in all epidermal cell types (the stomatal lineage cells, pavement cells and guard cells) and c) DCMU increases H₂O₂ levels in roots (H₂DCFDA) and leaves (H₂DCFDA, DAB and BES-H₂O₂-AC staining). The identification of H₂O₂ as the trigger of nuclear localization of KIN10 upon DCMU usage might be biased by the temporal frame of the experiment (Mune Bosch 2021 November 05, 2021 DOI:<https://doi.org/10.1016/j.tplants.2021.10.005>).

DCMU is a herbicide, aka DIURON that binds to the QB site of PSII, avoiding electron transfer to Plastoquinone (Mackay and O'Malley ; Z Naturforsch C J Biosci . Mar-Apr 1993;48(3-4):291-8). Its main effect in PSII is the blockage of photochemical reactions downstream of QA and DCMU produces a concomitant increase in the lifetime of the primary radical pair (3P680+Pheo-) (Krieger-Liszkay J Exp. Bot. 56 (411), Jan 2005, 337–346). This scenario provokes a boost in the Singlet oxygen production in photosynthesis, one of the most studied signaling ROS- molecules (Flors et al. J. Exp. Bot., 57(8) May 2006, 1725–1734; Dogra and Kim Front. Plant Sci., 08 January 2020). The kinetics of DCMU in illuminated adult leaves, at concentrations similar to the treatment used in the draft, demonstrate that, as early as 30 minutes, there is an induction in the production of singlet oxygen (tracked by SOSG fluorescence, Flors et al 2006). After 90 min, the whole leaf area is producing singlet oxygen, which leads to i) a profound change in gene expression (multiple studies, e.g. op den Camp et al. (2003) Plant Cell 15, 2320–2332) and ii) a wide range of other (with longer lifespan) IROS species including H₂O₂ but also volatile signaling molecule β-cyclocitral (β-CC) and products of lipid peroxidation. Then, addressing H₂O₂ as the ROS molecule involved in relocation of KIN10 after 12hrs of treatment with DCMU is not completely correct. This is even more complex if we consider seedling of 4 days whose photosynthetic machinery is far from being fully assembled and are more prone to suffer photoinhibition at PSII and photobleaching. The opinion of this reviewer is that the chosen pharmacological approach to evaluate KIN10 intracellular redistribution (full blocking of photosynthesis for 12 h in seedlings in the middle of the transition to autotrophy) is closer to a harsh stress treatment than a developmental reprogramming signaling leading to an increased gas exchange by higher stomatal index. As suggested option, the usage of controlled or brief treatment with methyl viologen (inducing superoxide) in combination with mutants in catalase and/or peroxidase can be more appropriate to mimic an increase in H₂O₂ concentrations.

The second main remark is centred on two methodological aspects which are central in the construction of the working hypothesis of the draft: a) evaluation of KIN10 intracellular change and b) quantification of H₂O₂ concentration.

Regarding a) most of the published evidence relies on at least 2 techniques (e.g. fluorescent microscopy and fractionation/Western blot). It is fully understandable that the complexity of the system complicates any biochemical enrichment of stomatal lineage cells. At the same time, a better imaging via fluorescence microscopy is needed to confirm the conclusions proposed by the authors. Basically, the usage of LSCM leads to better resolution than normal fluorescence microscopy but requires a more detailed imaging along Z axis, and the usage of Bright field Images to support the conclusions obtained from fluorescence images (as a reference paper: for reference see. Putarjuna et al., 2019 NATURE PLANTS | VOL 5 | JULY 2019 | 742–754 | www.nature.com/natureplants742). Usually the focal plane of two-neighbour-cells' nuclei do not always coincide, being observed in 2 different positions in the Z axis. Then, if the analysis of their fluorescence is performed in one single image, an intrinsic underestimation of the signal of one of the cells takes place. In other words, the analysis compared one nucleus in focus with maximum fluorescence and one nucleus out of focus. This

situation is even more pronounced in comparisons between guard cells/meristemoid cells and pavement cells. In the case of the last ones, the vacuole occupancy of the majority of cell space “push” the nucleus to the abaxial position of the cell. At the same time, guard cells are the epidermal cells with the more adaxial position in Z axis. Then, it is really complicated with only one single image to determine that 2 cells have different KIN10-GFP distribution. For example, the “presumed/assumed” enriched nuclear-localization of KIN10 in meristemoid and stomatal lineage ground cells, to promote stomatal development, compared to mainly cytoplasmic localization in pavement cells (Fig. 1a-d and Supplemental Fig. 2a, b, also in Han, C. et al. Nat Commun 11, 4214 (2020) might be considered more carefully. Similar assumptions based on “one-focal plane image” can be also found In Han et al 2020 (for instance, Fig 4e show a central pavement cell imaged in a focal plane in Z axis in a more adaxial position with respect to the nucleus position). This limitation in the analysis based on “single fluorescence microscopy image” can be partially corrected by using BF images and/or an organelle marker in a second channel of fluorescence to localize unambiguously the position of the nucleus in each cell. Either a nuclear marker or an ER marker, with the associated perinuclear ring structure, allow to clearly identify nucleus position, and are options to conduct these experiments. Despite this approach, the recommended analysis should be performed using different ROIs for each cell over a set of Z stack images along the Z axis and FIJI associated-plugins (e.g. Costes et al., Biophys J. 2004 Jun;86(6):3993-4003) to ensure unbiased, highly stringent analysis of the fluorescence intensities during the quantification process. Regarding b) quantification of H₂O₂ concentration, the usage of different in situ ROS detection methods is adequate, though some aspects should be considered. Together with the above-mentioned heed regarding fluorescence microscopy imaging, the specificity of the detection assays is crucial to infer correct conclusions (Noctor et al., 2016 Plant, Cell and Environment (2016) 39, 1140–1160; Smirnoff and Arnaud,2019; New Phytologist (2019) 221: 1197–1214). Among the different chosen methods in the draft to detect H₂O₂ (DAB; H₂DCFDA₂, BES-H₂O₂-Ac and pHYPER), pHyper and BES-H₂O₂-Ac can be considered the most exact approach due to their high specificity to H₂O₂ (Smirnoff and Arnaud,2019; New Phytologist (2019) 221: 1197–1214, Maeda et al., 2008 Volume1130, Issue1 Fluorescence Methods and Applications: Spectroscopy, Imaging, and Probes.). On the other hand, Fluorescein probes like H₂DCFDA react with a wide range of radical-based reactive species, including peroxynitrite and reactive sulfur species . DAB is not suitable for live tissues and requires peroxidase activity in the tissue in study to provide fast detection assay. Then, Results obtained by using H₂DCFDA staining methods should be considered as an indicative of an increase in total ROS-levels (Supplemental Fig. 1c, d, Fig 2 a-d), which in part is due to H₂O₂. In this context, a scenario of oxidative stress in the whole leaf tissue is a plausible alternative scenario to the proposed specific induction of H₂O₂ in meristemoid cells. Unfortunately, figure 2 only indicates differences between SLGC and meristemoids cells. With this info, this reviewer thinks that the mechanistic link between H₂O₂ and KIN10 specific redistribution in meristemoids cells is not sufficiently demonstrated.

Reviewer #4 (Remarks to the Author):

This manuscript aims to show that SPEECHLESS, the transcription factor that drives stomata development, represses the expression of antioxidant enzyme genes (CAT2, APX), leading to the accumulation of ROS specifically in the cells where SPCH is expressed (meristemoids and SLGC). Increased ROS levels in turn lead to the nuclear translocation of the KIN10 kinase, which in previous work was shown to stabilize SPCH.

The subject of this manuscript is really interesting and the model presented intriguing. Unfortunately, the study lacks robust evidence to support many of the statements.

Main concerns:

- The authors show compelling evidence that ROS accumulates to higher levels in meristemoid cells.

However, all other aspects are extremely difficult to evaluate, as there is not sufficient information provided about the materials and conditions, and when it is provided it changes from one experiment to another. Some of the factors that seem to vary amongst experiments are: presence/absence of sucrose in the media, use of solid vs. liquid media, age of seedlings (see more details below)

- I believe the statistics used in most of the analyses should be revisited, as there is no correction for multiple comparisons.
- The authors try to cover an enormous amount of aspects (some of them only described in the Discussion) but, because of that, most of them remain very superficial and lack robustness.
- In many instances authors use a single transgenic line or KO mutant to address a specific aspect rather than the two independent lines that is usually desired to confidently associate a phenotype with a particular modification. In addition, a basic characterization of these lines is missing (to show e.g. that they are indeed overexpressors or knockout mutants, etc).
- Regarding the impact of DCMU/H₂O₂ on KIN10 localization I have the concern that this is rather an effect on protein accumulation (either specific to KIN10 or more general one)
- Regarding the interaction of KIN10 with the subunits, the activity of KIN10 should be shown to demonstrate that the peroxide treatment is not simply denaturing the proteins and that is why the interaction is lost. What is the activity of KIN10 when treated or not with this H₂O₂ concentration (alone and in combination with the beta subunit)?

In more detail:

Figure 1

- Is the effect of DCMU and H₂O₂ really an effect on KIN10 localization or an effect on KIN10 accumulation in general? I think these treatments may be triggering KIN10 accumulation in other tissues/cell types. Also, is the effect shown for KIN10 a general one or specific to KIN10? Could this treatment be inhibitory to the proteasome and in that way reflect ALSO on KIN10? Authors could use as a control an NLS-GFP or RFP line to show that this is indeed KIN10-specific
- How do the 35S:CAT2-MYC plants look like? These plants were generated in this study but a basic characterization of these is totally lacking. Do the authors have a second independent line that behaves similarly? Can KIN10 accumulation in the nucleus indeed be rescued in these plants by applying H₂O₂ exogenously?
- In the legend of this and other figures it is said: error bars indicate standard deviation, but these seem to be box plots and in that case the bars would represent quartiles
- What was the material used in Suppl Fig. S1 to check that DCMU induces H₂O₂ accumulation and the counteracting effect of KI? The leaves shown in the picture look like leaves of soil-grown plants, and not of 5d-old seedlings. The effect of DCMU and DCMU+KI (Fig. S1) should be shown with the same material and under the same conditions that are used for assaying KIN10 localization and stomata development (Fig. 1)
- Indeed, many assays are done in MS medium + 1% sucrose. But in others, including those of Suppl Fig. 1 (also Fig. 1e-g, Fig. 4a-c, Fig. 5c-d, Suppl. Fig. 2c-d, Suppl. Fig. 8), no sucrose was used. Why? This is really important, as the previous work of the authors showed that KIN10 promotes stomata development only in seedlings grown in 1% sucrose. Also, the presence of sugar is likely to influence ROS levels and the energy conditions have been shown to change the localization of SnRK1 (Ramon et al. 2019).

Figure 2:

- Please describe what BES-H₂O₂-Ac and H₂DCFDA are and what are their differences
- Also, what are pHyPer plants?
- Legend of Fig. 2: "H₂O₂ specifically accumulated in meristemoid cells". The authors say themselves in the text that it is also in guard cells (lines 165-167), so please correct
- Information on growth conditions used for these assays is totally missing

Figure 3

- There is no information nor basic characterization about the pCAT2:GFP or pAPX:GFP lines. How was the promoter of these lines determined? How can we know that indeed these reporters reflect the

expression of the corresponding genes?

- In the chromatin-IP experiment of Fig. 3g (and Fig. S5) it is hard to conclude much without having controls. A TF will bind DNA as compared to a non-TF protein like YFP. The key here is to show that this binding is specific for that particular sequence. For example, showing that SPCH binds to the APX promoter as opposed to another promoter or as opposed to the coding region of APX.
- Related to the same figures, which regions of the promoters were used? Are these the regions that were shown to be bound by SPCH in the reference #30? Also, it is not clear from the supplementary table and methods description whether the same regions were amplified also for the DNA-protein pulldowns

Figure 4

- Even though I understand the point of the authors in introducing new loss-of-function cat mutants to the study, it would be important to have here the 35S::CAT2 line used in Fig. 1. Since that line was used for assessing the effect of ROS on KIN10 localization it would be important to know how it behaves regarding stomata development. Do these plants have reduced stomata numbers?
- The conditions are also different in this figure, as no sucrose seems to have been used
- The authors say in the text (lines 261-263) "The amount of meristemoid cells and GMC in the abaxial epidermis at 5days after germination (DAG) were significantly increased in cat2, cat3, cat2 cat3, cat1 cat2 cat3 mutants compared to that in the wild type (Fig. 4a, b)". However, there are no statistical analysis being performed to back up that statement of significance
- In Fig. 4c they use solid medium and the plants were grown for 8d as opposed to the 4-5 days of other experiments. Do these H2O2 treatments lead to different ROS concentrations that can indeed be correlated with the observed stomata indexes? Also, is H2O2 stable in plates for so many days in the light? Showing ROS levels would support that H2O2 is stable and is having the desired effect
- In Fig. 4d-g the seedlings are 3d old. It is very difficult to compare results from 3d old seedlings with those from 8d old seedlings, in particular when this involves the transition from heterotrophy to autotrophy
- Related to Fig. 4d-g the text claims in lines 304-308 "We observed that mutation of CAT2 and CAT3 dramatically increased the number of cells marked by pSPCH::nucGFP and pSPCH::SPCH-GFP compared to those in a wild type background (Fig. 4d-g). These results indicated H2O2 alters the epidermal cell fates in the Arabidopsis leaves". This is quite an overstatement. Lack of CAT2/CAT3 can have numerous consequences beyond increased ROS levels and those could be causing the changes in SPCH. To have such a strong statement the authors should demonstrate that indeed this is due to increased ROS and could perhaps use a ROS scavenger like KI to revert the effect of the CAT mutations

Figure 5

- The authors employ in these assays a high concentration of peroxide that may denature the proteins and as consequence of that the interaction is lost. Is KIN10 still active after such treatment? Are other protein interactions not broken under such conditions?
- Does the H2O2 treatment lead to nuclear localization of KIN10 when it is expressed alone as KIN10-GFP? What was the H2O2 concentration used in this assay?
- To the best of my knowledge these KINbeta lines have not been previously described so a proper characterization is needed to show that these are bonafide loss-of-function mutants.

OTHERS

- The paper cited in line 86 has no data regarding nuclear translocation of KIN10
- Line 508. Correct chromosome to chromatin
- English language should in general be revised
- The manuscript could be better structured, as it starts with KIN10, in the middle shows the effects of ROS on stomata, and then goes back to KIN10 through the interaction with the beta subunits

Responses to comments by reviewers

We wish to express our deep appreciation for the constructive comments on our manuscript by the reviewers. In response to these comments, we have conducted additional experiments and modified the text extensively to improve our manuscript. Specifically, we have added the following results:

1. We have reanalyzed the fluorescent signals of KIN10-YFP in the epidermal cells of leaves using the z-stack projection images, which capture *pKIN10::KIN10-YFP* expressing nuclei in the three-dimensional space (Fig. 1b-i, Fig. 5g-h, Supplementary Fig. 1a-f, Supplementary Fig. 4a-d, Supplementary Fig. 5a-b, and Supplementary Fig. 6a-b). Consistent with our previous data, these results revealed that KIN10-YFP is mainly localized in the nucleus of meristemoid cells under all examined conditions.
2. We have analyzed the H₂O₂ distribution patterns in the leaf epidermal cells of plants grown under normal growth conditions or carbon starvation caused by low light intensities or short photoperiods, and found that H₂O₂ is specifically enriched in the meristemoid cells of plants grown under all examined conditions (Supplementary Fig. 10a-d).
3. We have analyzed the KIN10 subcellular localization patterns in the epidermal cells of plants grown under normal growth conditions or carbon starvation caused by low light intensities or short photoperiods (Supplementary Fig. 1a-d).
4. we have analyzed the H₂O₂ distribution patterns and KIN10 subcellular localization patterns in the epidermal cells of rosette leaves (Supplementary Fig. 1e-f, Supplementary Fig. 10e-f, and Supplementary Fig. 11a-d).
5. We have analyzed the superoxide anion distribution pattern in the leaf epidermal cells using the fluorescent dye dihydroethidium (DHE), and found superoxide anion was evenly distributed in the epidermal cells of leaves (Supplementary Fig. 12a-f).
6. We have analyzed the superoxide anion effects on the subcellular localization

of KIN10 and stomatal development, and found that superoxide anion had no significant effects on KIN10 subcellular localization and stomatal development (Supplemental Fig.5a-d and Supplementary Fig. 17a-h).

7. We have re-analyzed the effects of *CAT2* ectopic expression on KIN10 subcellular localization and stomatal development using two independent transgenic lines for each construct (Fig. 4a-b, Supplementary Fig. 6a-b, and Supplementary Fig. 15a-b).
8. We have analyzed the subcellular localization of KIN10 in wild type, *kinβ2* mutant, and *p35S::KINβ2-RFP* transgenic plants with or without H₂O₂ treatment (Fig. 5g-h).
9. We have added the experimental information in the figure legends and Material&Methods section.
10. We have extensively edited the manuscript to correct grammatical errors and to improve clarity.

Sincerely,

Mingyi

REVIEWER COMMENTS

Reviewer #1 (Remarks to the Author):

Authors showed previously that KIN10, a catalytic α -subunit of energy homeostasis controlling kinase SnRK1, accumulates to nucleus in stomatal precursor cells and thereby promotes stomatal development (Han et al., 2020). This regulatory circuit was shown to operate only upon carbon starvation and in sucrose dependent fashion. In their current work Shi et al. suggest that cell type specific H₂O₂ levels regulate subcellular localization of KIN10 and thus control stomatal development. Blanco et al. (2019) showed previously that treatment with a herbicide DCMU is sufficient to change subcellular localization of KIN10 and promote its nuclear localization in the root. Shi et al. study effects of DCMU and notice that it promotes nuclear localization of KIN10 in cotyledon epidermis. DCMU specifically blocks electron transfer downstream of PSII, however, authors suggest that effect of DCMU on KIN10 localization may be caused by metabolic stress induced H₂O₂ production. Effect of DCMU on KIN10 localization can be reversed by Potassium Iodide treatment, quencher of H₂O₂, supporting this idea. Further, authors show that also H₂O₂ treatment promotes nuclear localization of KIN10 and further, leads to increased stomatal index (ratio of stomata to total epidermal cells) indicating that subset of asymmetric division types in the stomatal lineage might be affected. Authors show that stomatal precursor cells, meristemoids, show high H₂O₂ levels and that various mutants with high and low H₂O₂ levels show reduced altered H₂O₂ levels specifically in the meristemoids. Authors find that the genes encoding ROS- scavenging enzymes, CATALASE2 (CAT2) and ASCORBATE PEROXIDASE 1 (APX1) show low expression levels in meristemoids. Interestingly, the master regulator of stomatal lineage initiation, SPEECHLESS (SPCH), has been previously shown to bind promoter regions of CAT2 and APX1 (Lau et al., 2014). Authors confirm this interaction by quantitative ChIP-PCR and hypothesize that SPCH may directly regulate expression of CAT2 and APX1 in meristemoids, and this regulation may be repressive in nature. Loss of CAT or APX1 leads to elevated stomatal index and *cat2 cat3* mutants shows dramatically increased translational and transcriptional SPCH expression. Finally, authors show that KIN2

and KIN10 interaction (which has been previously shown to reduce nuclear localization of KIN10 by Ramon et al.(2019)) is repressed by H₂O₂. This data shows that meristemoid specific high H₂O₂ levels further enhances nuclear localization of KIN10 and thus promote stomatal development.

This work contains novel and interesting findings. Authors show that ROS plays a role in stomatal development and refine the regulatory scheme they previously identified. Major weakness of this manuscript is variable description of experimental conditions. It would have been especially important since this work is continuation of the previous work (Han et al., 2020) where authors showed that in the nucleus, KIN10 phosphorylates SPCH and this interaction leads to stabilization of SPCH and promotion of stomatal development only upon carbon starvation. In the current version of the manuscript, information on used experimental conditions is limited and in addition, several methods are poorly described and therefore it is hard to follow what has been done. It is also problematic, that it is not discussed whether the proposed model operated only upon carbon starvation. I will specify this aspect and my other concerns below.

Response: Sorry for this confusing description. In the revised manuscript, we have added the information about the plant growth conditions in the figure legends and Material and Methods section. In addition, we have analyzed the H₂O₂ distribution patterns and KIN10 subcellular localization patterns in the leaf epidermal cells of plants grown under normal growth conditions and carbon starvation conditions, such as low light intensities or short photoperiods. The results showed that the high accumulation of H₂O₂ in meristemoid cells and the specific nuclear localization of KIN10 in meristemoid cells were generally presented in the cotyledon and rosette leaves of plants, which were grown on the ½ MS solid medium with or without 1% sucrose under different light intensities or different photoperiod conditions. (Supplementary Fig. 1a-f, Supplementary Fig. 10a-f, and Supplementary Fig. 11a-d). These results indicated that H₂O₂ is specifically enriched in the meristemoid cells to promote the nuclear localization of KIN10 in the plants that were grown under all examined conditions.

The nuclear localization might not fully activate KIN10 to promote stomatal development, it requires other regulatory modification, such as carbon starvation-mediated phosphorylation at the conserved threonine in the activation T-loop. Therefore, KIN10 has the most significant promoting effects on stomatal development under carbon starvation conditions. We have added a paragraph discussion to explain these results.

Supplementary Fig.1 The enriched nuclear-localization pattern of KIN10 in meristemoid cells is widespread in the plants grown under different conditions.

a-b, Confocal microscopic analysis of subcellular localization of KIN10 in the cotyledon epidermal cells of plants grown under different light intensity conditions. Seedlings of *pKIN10::KIN10-YFP* were grown on $\frac{1}{2}$ MS solid medium with or without 1% sucrose under 16h light/8h dark photoperiod with different light intensity for 4 days. **c-d**, Confocal microscopic analysis of subcellular localization of KIN10 in the cotyledon epidermal cells of plants grown under different photoperiod conditions. Seedlings of *pKIN10::KIN10-YFP* were grown on $\frac{1}{2}$ MS solid medium with or without 1% sucrose under different photoperiod with 100 $\mu\text{Mol}/\text{m}^2/\text{s}$ for 4 days. **e-f**, The subcellular localization of KIN10-YFP in the rosette leaves. Seedlings of *pKIN10::KIN10-YFP* were grown on $\frac{1}{2}$ MS solid medium containing 1% sucrose under 16h light/8h dark photoperiod with 100 $\mu\text{Mol}/\text{m}^2/\text{s}$ for 12 days.

Fluorescent signals were taken using LSM880 microscope from Zeiss. Nuclear and cytoplasmic KIN10-YFP signal from more than 200 epidermal cells in 10 cotyledons or rosette leaves were analyzed by ImageJ software. Scale bars in confocal images represent 10 μm in (a,c) and in (e) represent 20 μm . Different letters above the bars indicated statistically significant differences between the samples (Two-way ANOVA analysis followed by Uncorrected Fisher's LSD multiple comparisons test, $p < 0.05$). Asterisk between the bars indicated statistically significant differences between the samples (Student's t test, **** $p < 0.0001$).

Supplementary Fig. 10 H₂O₂ specifically accumulates in the stomatal lineage cells of plants grown under different conditions.

a-b, H₂DCFDA staining in the cotyledon epidermal cells of plants grown under different light quantity conditions. Seedlings of Col-0 were grown on ½ MS solid medium with or without 1% sucrose under 16h light/8h dark photoperiod with different light intensities for 4 days. **c-d**, H₂DCFDA staining for H₂O₂ in the cotyledon epidermal cells of plants grown under different photoperiod conditions. Seedlings of Col-0 were grown on ½ MS solid medium with or without 1% sucrose under different photoperiods with 100 $\mu\text{Mol}/\text{m}^2/\text{s}$ for 4 days. **e-f**, H₂DCFDA staining in the rosette leaves. Seedlings of Col-0 were grown on ½ MS solid medium containing 1% sucrose 16h light/8h dark photoperiod with 100 $\mu\text{Mol}/\text{m}^2/\text{s}$ for 12 days.

Fluorescent signals were taken using LSM700 microscope from Zeiss. signal from more than 200 epidermal cells in 10 cotyledons or rosette leaves were analyzed by ImageJ software. Scale bars in confocal images represent 10 μm in (a,c) and in (e) represent 20 μm . Different letters above the bars indicated statistically significant differences between the samples (Two-way ANOVA analysis

followed by Uncorrected Fisher's LSD multiple comparisons test, $p < 0.05$). Asterisk between the bars indicated statistically significant differences between the samples (Student's t test, **** $p < 0.0001$).

Major comments

1. Growth conditions and data interpretation

Authors showed previously that kin10 mutant shows consistent response to sucrose only when grown in darkness or liquid growth media, however, response becomes inconsistent when grown on solid media or different light conditions (Han et al., 2020; Figure 2c and Supplementary Figure 5). Therefore, it is unfortunate that authors do not specify clearly in their current work on KIN10 did they use solid or liquid growth media and what were the other growth conditions. Indication of the use of sucrose seems random, which is unfortunate, since sucrose modifies molecular interaction in the studied system. In few cases it has been stated that solid media was used, however, it is not explained why authors choose to use in these particular experiments different conditions. It is difficult to define what are the used standard conditions.

Response: Sorry for this confusing description, we have added the information about the plant growth conditions in the figure legends.

Specific comments to figures:

- Figure 1: What is "medium" – liquid or solid? Is some information missing or were these experiment done with different conditions? For example in Figure 1 b-d: 1% sucrose was used; Figure 1 e-g: sucrose is not mentioned – is this correct? Also, day length is mentioned only for Figure 1 h-l – is used day length same in all the experiments?

Response: Thank you for pointing this. In the figure 1b-l, seedlings were all grown on the $\frac{1}{2}$ MS solid medium containing 1% sucrose under 16 h light/8h dark photoperiod with $100 \mu\text{Mol/m}^2/\text{s}$ for 4 days, and then treated with or without indicated chemicals for different time.

- Figure 4: No growth conditions are mentioned for Figure 4a & 4b, however, Figure 4c label indicates that ½ MS solid medium has been used but no sucrose mentioned. Is this correct information? If it is, what was then the agar concentration used? Please indicate why did you choose to change growth conditions only for this particular experiment where you show that *kin10* does not respond to H₂O₂?

Response: Thank you for pointing this. In the figure 4a and 4b, seedlings of Col-0 and indicated mutants were grown on the ½ MS solid medium with 0.7% Agar containing 1% sucrose under 16 h light/8h dark photoperiod with 100 µMol/m²/s for 5 days. In the figure 4c, seedlings of Col-0 and *kin10* mutants were grown on the ½ MS solid medium with 0.7% Agar containing 1% sucrose and different concentrations of H₂O₂ under 16 h light/8h dark photoperiod with 100 µMol/m²/s for 8 days. In the figure 4d-g, seedlings of *pSPCH::nucGFP*, *pSPCH::nucGFP/cat2cat3*, *pSPCH::SPCH-GFP*, and *pSPCH::SPCH-GFP/cat2cat3* were grown on the ½ MS solid medium with 0.7% Agar containing 1% sucrose and different concentrations of H₂O₂ under 16 h light/8h dark photoperiod with 100 µMol/m²/s for 3 days.

- Figure 5: c-d no sucrose mentioned - Is this correct information?

Response: Thank you for pointing this. In the figure 5c and 5d, seedlings of *pKIN10::KIN10-Myc* and *pKIN10::KIN10-Myc/pKINβ2::KINβ2-YFP* were grown on ½ MS solid medium containing 1% sucrose under 16 h light/8h dark photoperiod with 100 µMol/m²/s for 10 days, and then treated with H₂O (mock) or 2 mM H₂O₂ for 3 h.

- Supplementary fig 1: no sucrose mentioned - correct?

Response: Thank you for pointing this. In the supplementary figure 1 (Supplementary Fig. 2 in the revised manuscript), seedlings of Col-0 were grown on ½ MS solid medium containing 1% sucrose under 16 h light/8h dark photoperiod with 100 µMol/m²/s for 5 days, then treated with 50 µM DCMU and/or 1 mM KI for 12 hours.

- Supplementary fig 2 c-d: no sucrose mentioned - correct?

Response: Thank you for pointing this. In the supplementary figure 2c-d (Supplementary Fig. 4c-d in the revised manuscript), seedlings of *pKIN10::KIN10-YFP* were all grown on the ½ MS solid medium containing 1% sucrose under 16 h light/8h dark photoperiod with 100 µMol/m²/s for 4 days, and then treated with or without 2 mM H₂O₂ and/or 50 µM CHX for 2 hours.

- Supplementary fig 5: no sucrose mentioned - correct?

Response: Thank you for pointing this. In the supplementary figure 5a-c (Supplementary Fig. 13a-c in the revised manuscript), seedlings of *pAPX1::GFP* were all grown on the ½ MS solid medium containing 1% sucrose under 16 h light/8h dark photoperiod with 100 µMol/m²/s for 4 days.

- Supplementary fig 8: solid medium, no sucrose - correct?

Response: Thank you for pointing this. In the supplementary figure 8 (Supplementary Fig. 17a-h in the revised manuscript), seedlings of Col-0 and various materials were all grown on the ½ MS solid medium containing 1% sucrose and/or indicated chemicals under 16 h light/8h dark photoperiod with 100 µMol/m²/s for different time.

Please describe all standard growth conditions carefully in the Materials and methods and if there are any exceptions, clearly indicate this case by case and explain why different growth conditions were used.

Response: Thank you for pointing this. We have added the standard growth information in the Materials and Methods section.

Authors state in the discussion:

“Our previous study showed that the KIN10, the α-catalytic subunit of SnRK1, specifically locates in the nucleus of meristemoids to phosphorylate and stabilize SPCH, and then promoting stomatal development under carbon starvation conditions. Here, we showed that H₂O₂ is abundantly enriched in meristemoid to promote the nuclear localization of KIN10 by reducing the interaction between KIN10 and its regulatory

subunit KIN β 2.” Please discuss your findings in the context of the previously identified regulatory scheme and clearly state, whether or not the H₂O₂ mediated regulation operates only upon hypoxia causing conditions.

Response: Thank you for pointing this. We have added the discussion as followed:

“SnRK1 is a central metabolic regulator of energy homeostasis and plays an important role in regulating plant growth and development in response to energy status (Broeckx et al., 2016; Crepin and Rolland, 2019). Our previous study showed that the KIN10, the α -catalytic subunit of SnRK1, specifically locates in the nucleus of meristemoids to phosphorylate and stabilize SPCH, and then promoting stomatal development under carbon starvation conditions (Han et al., 2020). However, in the present study, we found that the nuclear-localization of KIN10 in meristemoid cells occurs not only in the leaves of plants grown under carbon starvation conditions such as low light intensity or short-photoperiod in the absence of exogenous sucrose, but also in the leaves of plants grown under energy-sufficient conditions such as high light intensity and long-photoperiod in the presence of 1% sucrose. In addition, the nuclear-localization of KIN10 in meristemoid cells was not only observed in cotyledon, but also in the rosette leaves. These results suggested that the specific subcellular distribution pattern of KIN10 in the epidermal cells is widespread in the different leaves of plants grown under various conditions. It is inconsistent with the promoting effects of KIN10 on stomatal development only under carbon starvation conditions. One possibility for the inconsistency is that the nuclear localization in the meristemoid cell might cannot fully activate KIN10 to promote stomatal development, it requires other regulatory modification, such as carbon starvation-mediated phosphorylation at the conserved threonine in the activation T-loop. The *pKIN10::KIN10^{T175D}-YFP* transgenic plants, which mimic the phosphorylation of KIN10 at the activation T-loop, displayed the similar subcellular localization pattern of KIN10^{T175D} to that of wild type KIN10, but showed the significant increased stomatal index compared to wild type plants. These results indicated that the multiple regulatory mechanism allow for fine-tuning of KIN10 activity by environmental and developmental signals.”

References:

- Broeckx, T., Hulsmans, S., and Rolland, F. (2016). The plant energy sensor: evolutionary conservation and divergence of SnRK1 structure, regulation, and function. *J Exp Bot* **67**, 6215-6252
- Crepin, N., and Rolland, F. (2019). SnRK1 activation, signaling, and networking for energy homeostasis. *Curr Opin Plant Biol* **51**, 29-36.
- Han, C., Liu, Y., Shi, W., Qiao, Y., Wang, L., Tian, Y., Fan, M., Deng, Z., Lau, O.S., De Jaeger, G., and Bai, M.Y. (2020). KIN10 promotes stomatal development through stabilization of the SPEECHLESS transcription factor. *Nat Commun* **11**, 4214.

2. DMCU treatment

Recent paper by Zoulias et al. (*Curr Biol*, 2021) showed that DCMU treatment causes decrease in SPCH levels and reduces stomatal index. Here Shi et al., show that DCMU enhance nuclear localization of KIN10 and that this DCMU effect on KIN10 localization is caused mainly by increased accumulation of H₂O₂. However, plants treated with exogenous H₂O₂ or mutants with elevated H₂O₂ levels show increased stomatal index, which is opposite to DCMU caused phenotype described by Zoulias et al. In addition to the differences observed in the phenotype, Zoulias et al. also come up with completely different regulatory circuits upstream of SPCH. Interestingly, Zoulias et al., use milder DCMU treatment: single spray treatment with 10 μM DCMU whereas Shi et al., used 12 hours treatment with 50 μM DCMU. It is possible that this milder short-term treatment causes different effect at the molecular level and that other differences in the experimental set up may also play a role. 12h 50 μM DCMU treatment seems strong – does it affect to the cell area or leaf size? Although DCMU data is not central to this manuscript, it would be interesting to see, whether milder treatment, such as single spray event with 10 μM DCMU, will cause similar effect on KIN10 localization than 12 hours treatment with 50 μM DCMU. Please also refer to the work by Zoulias et al. and discuss, why DCMU and H₂O₂ treatments in these two studies may cause different effects on stomatal index.

Response: Thank you for pointing this. To determine whether these two DCMU treatment methods have different effects on the nuclear localization of KIN10 and stomatal development, we analyzed the subcellular localization of KIN10 and stomatal development of plants with the spray treatment by 10 μM DCMU or grown on the ½

MS solid medium containing 50 μM DCMU for different times. The results showed that DCMU treatment for 12 hours with these two methods both led to the increased nuclear/cytoplasmic ratio of KIN10 protein, and these promoting effects of DCMU were suppressed by KI cotreatment (Fig. 1b-d and Rebuttal Fig. 1a-b). However, KIN10 protein significantly decreased as the increase of DCMU treatment time beyond 48 hours (Rebuttal Fig. 1c). Consistent with this, the long-time DCMU treatment resulted in the decreased stomatal index (Rebuttal Fig. 1d). These results indicated that short-time DCMU treatment induced H_2O_2 accumulation and promoted the KIN10 nuclear localization, while long-time DCMU treatment resulted in the decreased KIN10 protein, which thereby inhibiting stomatal development.

Fig. 1 H_2O_2 induces the nuclear localization of KIN10.

b-d, KI prevents_DCMU-induced nuclear localization of KIN10 in M and SLGC. Seedlings of *pKIN10::KIN10-YFP* were grown on $\frac{1}{2}$ MS solid medium containing 1% sucrose under 16 h light/8h dark photoperiod with $100 \mu\text{Mol/m}^2/\text{s}$ for 4 days, and then treated with or without 50 μM DCMU and/or 1 mM KI for 12 hours. Nuclear and cytoplasmic KIN10-YFP signal from more than 200 stomatal lineage cells in 10 cotyledons were analyzed by ImageJ software. Scale bars in confocal images represent 10 μm . Different letters above the bars indicated statistically significant differences between the samples (Two-way ANOVA analysis followed by Uncorrected Fisher's LSD multiple comparisons test, $p < 0.05$).

Rebuttal Fig. 1 The effects of DCMU on KIN10 localization, KIN10 protein stability and stomatal development.

a-b, KI prevented DCMU-induced nuclear localization of KIN10. Seedlings of *pKIN10::KIN10-YFP* were grown on ½ MS solid medium containing 1% sucrose under 16 h light/8h dark photoperiod with 100 μMol/m²/s for 4 days, and then sprayed with or without 10 μM DCMU and/or 1 mM KI for 6 hours. **c**, DCMU induced the KIN10 protein degradation in plants. Seedlings of *p35S::KIN10-Myc* were grown on ½ MS solid medium containing 1% sucrose under 16 h light/8h dark photoperiod with 100 μMol/m²/s for 5 days, and then transferred to the ½ MS solid medium containing 1% sucrose with or without 10 μM DCMU for different times. **d**, Long-time DCMU treatment resulted in the decreased stomatal development. Seedlings of Col-0 were grown on ½ MS solid medium containing 1% sucrose and different concentrations of DCMU under 16 h light/8h dark photoperiod with 100 μMol/m²/s for 8 days.

Nuclear and cytoplasmic KIN10-YFP signal from more than 200 stomatal lineage cells in 10 cotyledons were analyzed by ImageJ software. Scale bars in confocal images represent 5 μm. Different letters above the bars indicated statistically significant differences between the samples (Two-way ANOVA [b] or One-way ANOVA [d] analysis followed by Uncorrected Fisher's LSD multiple comparisons test, $p < 0.05$). Error bars indicate standard deviation (S.D.) in panel [d].

3. Effect of H₂O₂ on SPCH expression

Here authors show that DCMU enhanced nuclear localization of KIN10 and accumulation of H₂O₂. Mutants with elevated H₂O₂ levels show increased SI and more transcriptional and translational SPCH signal. Please explain how the effect of high H₂O₂ on SPCH transcription is mediated in their model?

Response: Thank you for pointing this. SPCH has been reported to bind to its promoter to induce itself and stomatal related gene expression, forming a positive feedback loop to initiate stomatal formation(Lau et al., 2014). Here, we showed that H₂O₂ promotes the nuclear-localization of KIN10 in stomatal lineage cells, where KIN10 phosphorylates and stabilizes SPCH, then inducing its own expression and promoting stomatal development.

Reference:

Lau, O.S., Davies, K.A., Chang, J., Adrian, J., Rowe, M.H., Ballenger, C.E., and Bergmann, D.C. (2014). Direct roles of SPEECHLESS in the specification of stomatal self-renewing cells. *Science* **345**, 1605-1609.

4. KIN10- SPCH interaction upon H₂O₂ treatment

Authors have previously established variants of SPCH with altered KIN10 phosphorylation sites (Han et al., 2020). These variants would be excellent backgrounds to test the proposed model. Please test whether or not stomatal index and SPCH protein stability are affected in these lines and wild-type as a response to exogenous H₂O₂ treatment.

Response: Thank you for pointing this. We have analyzed the stomatal index of *pSPCH::SPCH-RFP/spch-4* and *pSPCH::SPCH-4A-RFP/spch-4* transgenic plants in response to H₂O₂ treatment. The results showed that H₂O₂ treatment significantly increased the stomatal index of *pSPCH::SPCH-RFP/spch-4* plants, but had weak effects on *pSPCH::SPCH-4A-RFP/spch-4* (Supplementary Fig. 16a). Consistent with this, H₂O₂ treatment significantly increased the SPCH protein levels, but had no significant effects on SPCH-4A protein (Supplementary Fig. 16b, c).

Supplementary Fig. 16 H₂O₂ fails to promote stomatal development on *pSPCH::SPCH-4A-RFP/spch-4* transgenic plants.

a, Quantification of stomatal index of wild type and indicated plants. Seedlings of Col-0 and *pSPCH::SPCH-RFP/spch-4* (*SPCH-8#*) and *pSPCH::SPCH-4A-RFP/spch-4* (*4A-2#*) were grown on ½ MS solid medium containing 1% sucrose with or without 10 Mm H₂O₂ under 16 h light/8h dark photoperiod with 100 μ Mol/m²/s for 8 days, Different letters above the bars indicated statistically significant differences between the samples (One-way ANOVA analysis followed by Uncorrected Fisher's LSD multiple comparisons test, $p < 0.05$). Error bars indicate standard

deviation (S.D.) (n =12–15). **b-c**, Quantification of the effects of H₂O₂ on SPCH or SPCH-4A protein. Seedlings of *p35S::SPCH-Myc* and *p35S::SPCH-4A-Myc* transgenic plants were grown on ½ MS solid medium containing 1% sucrose under 16 h light/8h dark photoperiod with 100 μMol/m²/s for 7 days, then treated with 1 Mm H₂O₂ for different times. Error bars indicate standard deviation (S.D.). Asterisk between the bars indicated statistically significant differences between the samples (Student's t test, **p < 0.01, *p < 0.05)

5. KIN10 localization in kinβ2

Row 373: “H₂O₂ reduces the interaction between KIN10 and KINβ2, thereby promoting the nuclear localization of KIN10”. Currently this is not shown. It would be informative to see KIN10 localization in the kinβ2 and p35S-kinβ2-YFP backgrounds to support this statement.

Response: Thank you for pointing this. We have analyzed the KIN10 localization in the *pKIN10::KIN10-YFP/p35S::KINβ2-RFP* and *pKIN10::KIN10-YFP/kinβ2* plants, and found that the nuclear-to-cytoplasmic ratio of KIN10 was increased in *pKIN10::KIN10-YFP/kinβ2* plants, but decreased in the *pKIN10::KIN10-YFP/p35S::KINβ2-RFP* plants (Fig. 5g, h). Furthermore, H₂O₂ treatment increased the nuclear-to-cytoplasmic ratio of KIN10 in wild-type background, and such promotion effects of H₂O₂ were slightly enhanced in *pKIN10::KIN10-YFP/kinβ2* plants, and reduced in *pKIN10::KIN10-YFP/p35S::KINβ2-RFP* plants (Fig. 5g, h).

Fig. 5 H₂O₂ reduces the interaction between KIN10 and KINβ2.

g,h H₂O₂ reduces the effects of KINβ2 on the KIN10-cytoplasmic retention in Arabidopsis leaves. Seedlings of *pKIN10::KIN10-YFP*, *pKIN10::KIN10-YFP/kinβ2* and *pKIN10::KIN10-YFP/p35S::KINβ2-RFP* were grown on ½ MS solid medium under 16 h light/8h dark photoperiod with 100 μMol/m²/s for 4 days, and then treated with H₂O (mock) or 2 mM H₂O₂ for 3 h. The Z-stack confocal images were taken using LSM880 microscope from Zeiss to capture *pKIN10::KIN10-YFP* expressing nuclei in the three-dimensional space. Nuclear and cytoplasmic KIN10-YFP signal from more than 200 stomatal lineage cells in 10 cotyledons were analyzed by ImageJ software. Scale bars in confocal images represent 10 μm. Different letters above the bars

indicated statistically significant differences between the samples (Two-way ANOVA analysis followed by Uncorrected Fisher's LSD multiple comparisons test, $p < 0.05$).

6. Discussion

Currently big part of the discussion (rows 399-413 and 429-434) is used to describe additional data. If this data is central to this manuscript, describe the data in the Results section and update the manuscript with all the appropriate information related to these experiments. If this data is not central, please remove the data.

Response: Thank you for pointing this. Considering the data about the regulation of BIN2 activity by H_2O_2 is not closely related to our major statement, we have deleted this part in the revised manuscript. However, the data about the $O_2^{\bullet-}$ regulation of stomatal development is closely related to our work, we moved them to the Result section and performed more experiments to prove that $O_2^{\bullet-}$ evenly distributed in the epidermal cells of leaves, and had no significant effects on KIN10 subcellular localization and stomatal development.

Minor comments

7. Figure 1: It seem that DCMU + KI and CAT-OE repress strongly KIN10 expression – do you see this constantly in your experiments? What may be cause for this?

Response: Thank you for pointing this. We also noticed this phenomenon, and found that DCMU treatment slightly induced the KIN10 protein accumulation, while removal H_2O_2 by KI treatment resulted in the decreased KIN10 protein (Rebuttal Fig. 2). Furthermore, the promoting effects of KI on KIN10 protein degradation were inhibited by the proteasome inhibitor MG132, but not the autophagy inhibitor 3-MA (Rebuttal Fig. 2), suggesting KI induces KIN10 protein degradation through a proteasome-dependent pathway.

Rebuttal Fig. 2 DCMU and KI regulates the KIN10 protein stability.

Seedlings of *p35S::KIN10-Myc* plants were grown on ½ MS solid medium under 16 h light/8h dark photoperiod with 100 $\mu\text{Mol/m}^2/\text{s}$ for 7 days, and then treated with DCMU, KI, MG132 and/or 3-MA for 12 h.

8. Please describe difference between H₂DCFDA and BES-H₂O₂-Ac dyes and what exactly do they indicate? Add also reference where they have been used before.

Response: Thank you for pointing this. Cell-permeant 2', 7'-dichlorodihydrofluorescein diacetate (H₂DCFDA) is a widely used ROS indicator. The reduced non-fluorescent fluorescein H₂DCFDA can be oxidized and converted into fluorescent 2', 7'-dichlorofluorescein (DCF) by intracellular H₂O₂, but these compounds suffer from the major drawback that they are poorly selective toward H₂O₂ (Eljebbawi et al., 2021). BES-H₂O₂-Ac is also a highly selective fluorescent probe for H₂O₂, and exhibits much higher H₂O₂ specificity than H₂DCFDA. BES-H₂O₂-Ac is applicable to clarifying cell response as well as dynamic function of H₂O₂ in living cells (Eljebbawi, et al., 2021).

References

Eljebbawi, A., Guerrero, Y., Dunand, C., and Estevez, J.M.. (2021). Highlighting reactive oxygen species as multitaskers in root development. *iScience* **24**, 101978.

9. Please write the gene names open for all genes when you mention them first time and also refer to the original paper describing the gene for the first time (for example UPB1, CAT, APX1).

Response: Thank you for pointing this. We have added the full name of these genes and cited the original papers about these genes when we mention these gene first time.

10. Row 222: Authors may mean positive relationship (larger the cell the more there is signal), not negative relationship. If this is correct, please modify to be in line with the data.

Response: Sorry for this mistake. We have changed it.

12. Row 256: “These results indicated that SPCH directly bound to the promoters of ROS scavenging genes *CAT2* and *APX1* to repress their expression and promote H₂O₂ accumulation in meristemoids.” The data currently shown does not show that SPCH represses *CAT2* and *APX1*. Please modify if additional support for the direct repression is not presented.

Response: Thank you for pointing this. We have performed the quantitative RT-PCR assay with wild-type plants, *p35S::SPCH-Myc* and *spch-4* mutant. The results showed the expression levels of *CAT2* and *APX1* were significantly increased in *spch-4* mutant, but decreased in *p35S::SPCH-Myc*, suggesting SPCH represses *CAT2* and *APX1* expression.

Rebuttal Fig. 3 Quantitative RT-qPCR analysis of the expression of *CAT2* and *APX1* in wild type, *spch-4* and *p35S::SPCH-Myc* plants.

Seedlings for the wild-type Col-0, *spch-4* and *p35S::SPCH-Myc* plants were grown on ½ MS solid medium under 16 h light/8h dark photoperiod with 100 μMol/m²/s for 6 days. *PP2A* gene was used as an internal control. Error bars indicate standard deviation (S.D.). Different letters above the bars indicated statistically significant differences between the samples (One-way ANOVA analysis followed by Uncorrected Fisher’s LSD multiple comparisons test, $p < 0.05$).

13. Figure 4e: 90% SPCH expressing cells seems incredible high number of cells for any plant and especially considering the mild phenotype of this particular mutant. Is this correct? Also consider whether the number 120% on Y-axis is meaningful.

Response: Thank you for pointing this. We have repeated this experiment and found that mutation of *CAT2* and *CAT3* significantly increased, whereas KI treatment decreased the percentage of cell expressing *SPCH* to total epidermal cells (Fig. 4d-g), suggesting CAT-mediated scavenging of H_2O_2 plays an important role of *SPCH* expression.

Fig. 4 H_2O_2 promotes stomatal development.

d-g, *CAT* mutations altered cell fate in Arabidopsis epidermis. Seedlings of *pSPCH::nucGFP*, *pSPCH::nucGFP/cat2cat3*, *pSPCH::SPCH-GFP*, and *pSPCH::SPCH-GFP/cat2cat3* were grown on $\frac{1}{2}$ MS solid medium containing 1% sucrose with or without 1mM KI under 16 h light/8h dark photoperiod with 100μ Mol/m²/s for 3 days. Error bars indicate standard deviation (S.D.) (n =12–15). Scale bars in confocal images represent 20 μ m. Different letters above the bars indicated statistically significant differences between the samples (One-way ANOVA analysis followed by Uncorrected Fisher's LSD multiple comparisons test, $p < 0.05$).

14. Row 307: H_2O_2 may not alter epidermal fates but increases frequency of stomata producing divisions. Please modify.

Response: Thank you for pointing this, we have changed it.

15. Description of pharmacological treatments, staining methods and measurement methods

Please add and improve Materials and Methods so that all the used methods are included so that experiments can be easily repeated.

Response: Thank you for pointing this, we have added the information about the experiments in the Materials and Methods section.

- Please describe DMCU and all other treatments in detail in the Materials & Methods. Describe also what were the mock treatments. Please add this information for all treatments.

Response: Thank you for pointing this. We have added the information about DCMU treatment in the Materials and Methods section, and also added the information in the figure legends.

- Please describe how DAB staining was done. Also, describe how did you measured and quantified DAB data in Supplementary Figure 1 c-d? (veins show very strong signal compared to the other parts of the leaf – which regions were used?)

Response: Thank you for pointing this. We have added the DAB staining assay in the Materials and Methods section, and also added the measurement detail in the figure and figure legend.

- Please describe the source of HyPer line and imaging set up in Materials and Methods.

Response: Thank you for pointing this. We have added the information about pHyper transgenic plants in the Materials and Methods section.

- Description of the nuclear/cytoplasmic ratio measurement is quite ok (Figure legend 1), however, it would be easier to find it from Materials and Methods. So please consider transferring this information there.

Response: Thank you for pointing this. We have added the nuclear-to-cytoplasmic ratio measurement of KIN10 assay in the Materials and Methods section.

Please also consider how this measuring method fits to the idea that KIN10 is localized to the ER as described by Blanco et al., 2019.

Response: Thank you for pointing this. To distinguish the ER and nuclei, we generated the *pKIN10::KIN10-YFP/p35S::Histone3-RFP* plants, in which *p35S::Histone3-RFP* was used as a nuclear marker in a second channel of fluorescence, and analyzed the H₂O₂ effects on the nuclear localization of KIN10 (Supplementary Fig. 7).

Supplementary Fig. 7 Quantification of H₂O₂ effects on the subcellular localization of KIN10-YFP and Histone3-RFP in the *pKIN10::KIN10-YFP/p35S::H3-RFP* transgenic plants.

a-d, The subcellular localization of KIN10-YFP and Histone3-RFP in response to H₂O₂. Seedlings of *pKIN10::KIN10-YFP/p35S::H3-RFP* were grown on ½ MS solid medium containing 1% sucrose under 16h light/8h dark photoperiod with 100 μMol/m²/s for 4 days, and then treated with or without 2 mM H₂O₂ for 3 hours. Magnifications of the epidermal cells are shown on the middle (b). The fluorescent signals of GFP and RFP were determined along a line drawn on the confocal images using ImageJ software. The arrow labeling with A and B represents the fluorescent signals in meristemoid cells, and the arrow labeling with B and C represents the fluorescent signals in SLGC. **e**, Quantification of nuclear localization of KIN10 in different scale epidermal cells. **f**, Quantification of nuclear localization of Histone 3 in different scale epidermal cells.

Signals were taken using LSM700 microscope from Zeiss. Nuclear and cytoplasmic KIN10-YFP and Histone3-RFP signal from more than 200 epidermal cells in 10 cotyledons were analyzed by

ImageJ software. Scale bars in panel (a) represent 20 μm , and in panel (b) represent 10 μm . Different letters above the bars indicated statistically significant differences between the samples (Two-way ANOVA analysis followed by Uncorrected Fisher's LSD multiple comparisons test, $p < 0.05$).

- Please described how cell are/fluorescent intensity was measured (For example presented in the Supplementary figure 2)

Response: Thank you for pointing this. We have added the information about how to measure the cell area and fluorescent intensity in Materials and Methods section.

Reviewer #2 (Remarks to the Author):

The work performed by Shi and coauthors demonstrated that spatially patterned hydrogen peroxide (H_2O_2) plays an essential role in stomatal development. The study is interesting and addresses important aspects related to the involvement of redox-related signals in the development of stomata in arabidopsis cotyledons. The experiments appear to have been well executed and the results are clear. The manuscript is very well written and the obtained results support the main conclusions of this paper. However, the authors should consider the following points:

1) The authors tested the spatial distribution of H_2O_2 and O_2 in epidermal cells and stomatal development. However, it is not clear from the manuscript whether other redox-related signals would play similar roles in stomatal development using the same mechanism.

Response: Thank you for pointing this. Reactive oxygen species (ROS) include hydrogen peroxide (H_2O_2), superoxide anion (O_2^-), singlet oxygen ($^1\text{O}_2$), and hydroxyl radical ($\cdot\text{OH}$). Among these, H_2O_2 and O_2^- are considered the important redox signaling molecule, given their specific physical and chemical properties, including the remarkable stability within cells, and rapid oxidation of target proteins. Singlet oxygen ($^1\text{O}_2$) and hydroxyl radical ($\cdot\text{OH}$) are unstable and have the strong oxidative ability in nature, which can rapidly oxidize target proteins, DNA or lipid. Therefore, H_2O_2 and O_2^- were generally selected to analyze their functions in plants (Suzuki et al., 2011;

Mittler, 2017). Here, in this study, we showed that H_2O_2 specifically enriched in the meristemoid cells to promote the nuclear localization of KIN10 and induce stomatal development. While O_2^- is evenly distributed in all epidermal cells of leaves, and inhibiting or promoting the formation of O_2^- does not change the subcellular location of KIN10 and has no obvious effects on stomatal development (Supplementary Fig. 5a-d, Supplementary Fig. 12a-f and Supplementary Fig. 17).

Supplementary Fig. 5 MV and DMTU have no significant effects on the nuclear localization of KIN10-YFP.

a-d, Confocal microscopic analysis of subcellular localization of KIN10 in the cotyledon epidermal cells of plants treated with different chemicals. Seedlings of *pKIN10::KIN10-YFP* were grown on $\frac{1}{2}$ MS solid medium with 1% sucrose under 16 h light/8h dark photoperiod with $100 \mu Mol/m^2/s$ for 4 days, and then treated with or without $1 \mu M$ MV and $10mM$ DMTU for 6 hours.

The Z-stack confocal images were taken using LSM880 microscope from Zeiss to capture *pKIN10::KIN10-YFP* expressing nuclei in the three-dimensional space. Nuclear and cytoplasmic KIN10-YFP signal from more than 200 stomatal lineage cells in 10 cotyledons were analyzed by ImageJ software. Scale bars in confocal images represent $10 \mu m$ in (a) and in (c) represent $20 \mu m$. Different letters above the bars indicated statistically significant differences between the samples (Two-way ANOVA analysis followed by Uncorrected Fisher's LSD multiple comparisons test, $p < 0.05$).

Supplementary Fig. 12 DHE staining for O_2^- in the epidermis cell of leaves.

a-f, Measurement of O_2^- in the epidermal cells of wild type cotyledon using DHE staining. Seedlings of Col-0 were grown on ½ MS solid medium containing 1% sucrose under 16h light/8h dark photoperiod with 100 $\mu\text{Mol/m}^2/\text{s}$ for 4 days. The fluorescent signals of DHE were determined along a line drawn on the confocal images using ImageJ software. Magnifications of the meristemoid and SLGC are shown on the middle (b). The white arrows inside the images show the areas used for line scan measurements that yielded plot profiles shown in the lower panels. The arrow labeling with A and B represents the fluorescent signals in SLGC, and the arrow labeling with B and C represents the fluorescent signals in meristemoid cells. Different letters above the bars indicated statistically significant differences between the samples (One-way ANOVA analysis followed by Uncorrected Fisher's LSD multiple comparisons test, $p < 0.05$). Scale bars in confocal images represent 20 μm .

Supplementary Fig. 17 MV and DMTU treatment have no marked effects on the stomatal development.

a, Quantification of the effects of MV on stomatal index. **b**, Quantification of the effects of DMTU on stomatal index. Seedlings of wild type Col-0 were grown in ½ MS solid medium containing different concentrations of MV or DMTU under 16 h light/8h dark photoperiod with 100 $\mu\text{Mol/m}^2/\text{s}$ for 8 days. Error bars indicate standard deviation (S.D.) ($n = 12-15$). Different letters above the bars indicated statistically significant differences between the samples (One-way ANOVA analysis followed by Uncorrected Fisher's LSD multiple comparisons test, $p < 0.05$).

References

Mittler, R. (2017). ROS Are Good. Trends Plant Sci **22**, 11-19.
 Suzuki, N., Miller, G., Morales, J., Shulaev, V., Torres, M.A., and Mittler, R. (2011). Respiratory burst

2) The conclusions of this work are based on experiments carried out in cotyledons of *Arabidopsis* seedlings. The article would benefit from the results of experiments carried out on young leaves. It would be important to indicate whether the proposed mechanism is also valid for the final stomatal density observed in adult plants.

Response: Thank you for this excellent suggestion. We have analyzed the spatial distribution of H_2O_2 and subcellular localization of KIN10 in the rosette leaves. The results revealed that H_2O_2 was specifically enriched in the meristemoid cells of rosette leaves, and the nuclear-to-cytoplasmic ratio of KIN10 was higher in meristemoid cells than in the SLGC cells (Rebuttal Fig. 4a-d). Furthermore, the stomatal index of rosette leaves in *cat2 cat3* mutants was much higher than that in wild-type plants (Rebuttal Fig. 4e). These results suggested that the spatial distribution pattern of H_2O_2 is widespread in the leaves of *Arabidopsis*.

Rebuttal Fig. 4 H_2O_2 enriched in the meristemoid cells to promote the nuclear localization of KIN10 in the rosette leaves.

a-b, The subcellular localization of KIN10-YFP in the epidermal cells of rosette leaves. **c-d**, H_2DCFDA staining in the epidermal cells of rosette leaves. **e**, Quantification of stomatal index in the Col-0 and *cat2 cat3* in the rosette leaves. Error bars indicate standard deviation (S.D.).

Seedlings of *pKIN10::KIN10-YFP*, Col-0 and *cat2 cat3* were grown on $\frac{1}{2}$ MS solid medium containing 1% sucrose under 16 h light/8h dark photoperiod with $100 \mu\text{Mol/m}^2/\text{s}$ for 12 days. Fluorescent signals were taken using LSM700 microscope from Zeiss. signal from more than 200 stomatal lineage cells in 10 cotyledons were analyzed by ImageJ software. Scale bars in confocal images represent $20 \mu\text{m}$. Asterisk between the bars indicated statistically significant differences between the samples (Student's t test, ****p < 0.0001, ***p < 0.001, **p < 0.01, *p < 0.05).

3) The manuscript does not present any discussion about the interactions that the proposed mechanism may have with environmental factors. Specifically, the authors could address the following questions: Would abiotic stress conditions lead to some

change in stomata density through changes in the levels of redox-related compounds in epidermal cells via the proposed mechanism? What is the level of involvement of hormones associated with the development of stomata, such as ABA?

Response: Thank you for this suggestion. We have added the discussion as followed:

“H₂O₂ has been reported to promote stomatal closure through increasing the concentrations of cytosolic free calcium. The plant stress hormone abscisic acid (ABA) and diverse biotic and abiotic stress stimulated H₂O₂ accumulation, and then induced stomatal closure, suggesting H₂O₂ acts as a negative regulator for stomata to inhibit their function on gas exchange between plants and atmosphere. In the present study, we showed that H₂O₂ positively regulate stomatal development. This opposite regulation on stomatal functions by H₂O₂ may be due to the different effects of different H₂O₂ contents on stomatal development and stomatal movement. Our results showed that the content of H₂O₂ in guard cells is approximately 10 times higher than that in meristemoid cells when plants were grown under normal conditions. ABA treatment led to about 5-fold increase in the H₂O₂ content in the guard cells (Supplementary Fig. 19). These results suggested that the concentrations of H₂O₂ in meristemid cells that is required to promote stomatal development is at least 50 times lower than the H₂O₂ concentration in guard cells that is responsible for promoting stomatal closure. Consistent with this, our results showed exogenous H₂O₂ treatment promoted stomatal development, but such promoting effects decreased as the increased concentrations H₂O₂. Furthermore, a recent study showed that H₂O₂ promoted stomatal opening in the intact leaves at the low concentrations. The *rbohD rbohF* mutant and *CATALASE2* overexpression transgenic plants displayed the less H₂O₂ in guard cells and failed to normally open stomata. These results indicated that H₂O₂ promotes the stomatal development and stomatal opening at low concentrations, thereby inducing the efficiency of CO₂ uptake and the photosynthesis under the normal growth conditions; but H₂O₂ promotes

stomatal closure at high concentrations, then reducing water loss and improving stress resistance of plant under stress conditions.”

Supplementary Fig.19 ABA treatment leads to increase in the H₂O₂ content in the guard cells.

a-b, H₂DCFDA staining in the epidermal cells of leaves in response to ABA. Seedlings of Col-0 were grown on ½ MS solid medium with 1% sucrose under 16h light/8h dark photoperiod with 100 μMol/m²/s for 4 days, then treated with or without 1 μM ABA for 6 h. Fluorescent signals were taken using LSM700 microscope from Zeiss. Signal from more than 100 guard cells in 10 cotyledons were analyzed by ImageJ software. Scale bars in confocal images represent 20 μm. Different letters above the bars indicated statistically significant differences between the samples (One-way ANOVA analysis followed by Uncorrected Fisher’s LSD multiple comparisons test, p < 0.05).

4) The authors did not explain the source/sites of production of H₂O₂ in the meristemoid cells and why they performed most of the analyses in 4-day-old seedlings. A previous study demonstrated that “Following germination, Arabidopsis seeds rely on storage oil breakdown to supply carbon skeletons and energy for early seedling growth, and massive amounts of H₂O₂ are generated within the peroxisome as a by-product of fatty acid β-oxidation” (Eastmond et al 2007). Accordingly, previous studies show that much of the generation of ROS is formed as byproducts of the fatty acid β-oxidation, albeit additional sources of ROS do exist during early stages of development, such as the NADPH oxidases. Data from Fulda et al. 2004, Bernhardt et al. 2012 and Feitosa-Araujo et al 2020, among others, provided evidence that β-oxidation is strong required in 4-day-old seedlings. The authors should contemplate these points in the discussion.

Response: Thank you for this excellent suggestion. We have added the discussion as followed:

“The spatiotemporal distribution of H₂O₂ in meristems is essential for the stem cell differentiation in plants(Owusu-Ansah and Banerjee, 2009; Tsukagoshi et al., 2010; Morimoto et al., 2013; Zeng et al., 2017). Here, we showed that H₂O₂ is abundantly enriched in meristemoid cells to promote stomatal development. The Arabidopsis seeds contain a lot of triacylglycerols, which are hydrolyzed during seed germination to provide carbon skeletons and energy for early seedling growth, while a large amount of H₂O₂ is generated as a by-product of fatty acid β -oxidation(Fulda et al., 2004; Eastmond, 2007; Bernhardt et al., 2012; Feitosa-Araujo et al., 2020). In addition, the plant shoot apical meristem is embedded in a low-oxygen nich and new leaves are produced under the hypoxic condition. When the new leaves emerge from the stem, they encounter an aerobic environment, thereby producing massive amounts of H₂O₂. Catalases and ascorbate peroxidase are key H₂O₂ scavenging enzymes in plants, and play important roles in plant anti oxidative and detoxification processes that are closely correlated with ROS generation during plant growth and stress responses. Our results showed that *CAT2* and *APX1* are widely expressed in most cells of leaf epidermis except meristemoid cells. SPCH directly binds to the promoters of *CAT2* and *APX1* to repress their expression in meristemoid cells, resulting in the reduced H₂O₂ content in meristemoid cells. UPB1 was previously identified as the key transcription factor to repress a set of peroxidase expression in root. The accumulated H₂O₂ in meristemoids triggers the nuclear localization of KIN10, which phosphorylates and stabilizes SPCH to promote stomatal development, thus forming a positive feedback loop to control stomatal development.”

References:

- Owusu-Ansah, E., and Banerjee, U. (2009). Reactive oxygen species prime *Drosophila* haematopoietic progenitors for differentiation. *Nature* **461**, 537-541.
- Tsukagoshi, H., Busch, W., and Benfey, P.N. (2010). Transcriptional regulation of ROS controls transition from proliferation to differentiation in the root. *Cell* **143**, 606-616.
- Morimoto, H., Iwata, K., Ogonuki, N., Inoue, K., Atsuo, O., Kanatsu-Shinohara, M., Morimoto, T., Yabe-Nishimura, C., and Shinohara, T. (2013). ROS are required for mouse spermatogonial stem cell self-renewal. *Cell Stem Cell* **12**, 774-786.

- Zeng, J., Dong, Z., Wu, H., Tian, Z., and Zhao, Z. (2017). Redox regulation of plant stem cell fate. *EMBO J* **36**, 2844-2855.
- Fulda, M., Schnurr, J., Abbadi, A., Heinz, E., and Browse, J. (2004). Peroxisomal Acyl-CoA synthetase activity is essential for seedling development in *Arabidopsis thaliana*. *Plant Cell* **16**, 394-405.
- Eastmond, P.J. (2007). MONODEHYDROASCORBATE REDUCTASE4 is required for seed storage oil hydrolysis and postgerminative growth in *Arabidopsis*. *Plant Cell* **19**, 1376-1387.
- Bernhardt, K., Wilkinson, S., Weber, A.P., and Linka, N. (2012). A peroxisomal carrier delivers NAD(+) and contributes to optimal fatty acid degradation during storage oil mobilization. *Plant J* **69**, 1-13.
- Feitosa-Araujo, E., da Fonseca-Pereira, P., Pena, M.M., Medeiros, D.B., Perez de Souza, L., Yoshida, T., Weber, A.P.M., Araujo, W.L., Fernie, A.R., Schwarzlander, M., and Nunes-Nesi, A. (2020). Changes in intracellular NAD status affect stomatal development in an abscisic acid-dependent manner. *Plant J* **104**, 1149-1168.

5) Why did the authors not consider SnRK2? According to a recent study published at Nature Plants by Belda-Palazón et al 2020, SnRK2 kinases have a dual role in the regulation of SnRK1 and plant growth.

Reference: Belda-Palazón, B., Adamo, M., Valerio, C. et al. A dual function of SnRK2 kinases in the regulation of SnRK1 and plant growth. *Nat. Plants* 6, 1345–1353 (2020). <https://doi.org/10.1038/s41477-020-00778-w>

Response: Thank you for this suggestion. Numerous studies have indicated the cooperation between SnRK1 and ABA signaling in plant stress response, growth and development (Rodrigues et al., 2013; Carianopol et al., 2020). Under favorable conditions, type 2C phosphatases PP2Cs not only dephosphorylate sub ground III SnRK2 kinases, but also SnRK1, to repress their activity. SnRK2 also binds to SnRK1 to further sequester the SnRK1 activity, allowing TOR to be active and promote plant growth. In response to ABA, these SnRK2 and PP2C-containing SnRK1 repressor complexes dissociate, releasing SnRK2 and SnRK1, which inhibit TOR and growth. ABA treatment has been reported to promote the translocation of the kinase subunit of SnRK1, KIN10 from nucleus to cytoplasm, thereby inhibiting TOR activity and plant growth. A recent study showed that ABA treatment inhibited stomatal development (Tanaka et al., 2013). These results indicated that ABA may inhibit stomatal development through promoting the cytoplasmic localization of KIN10.

References

- Carianopol, C.S., Chan, A.L., Dong, S., Provar, N.J., Lumba, S., and Gazzarrini, S. (2020). An abscisic acid-responsive protein interaction network for sucrose non-fermenting related kinase1 in abiotic stress response. *Commun Biol* **3**, 145.
- Rodrigues, A., Adamo, M., Crozet, P., Margalha, L., Confraria, A., Martinho, C., Elias, A., Rabissi, A., Lumberras, V., Gonzalez-Guzman, M., Antoni, R., Rodriguez, P.L., and Baena-Gonzalez, E. (2013). ABI1 and PP2CA phosphatases are negative regulators of Snf1-related protein kinase1 signaling in Arabidopsis. *Plant Cell* **25**, 3871-3884.
- Tanaka, Y., Nose, T., Jikumaru, Y., and Kamiya, Y. (2013). ABA inhibits entry into stomatal-lineage development in Arabidopsis leaves. *Plant J* **74**, 448-457.

6) The authors mentioned in the description of the results that “catalase and UPB1 play important roles in H₂O₂ distribution pattern in epidermal cells of leaves”. However, at no time UPB1 was mentioned in the discussion. Specifically, the authors could provide a deeper and more integrated mechanism to be included in the discussion and they also could explain how UPB would participate in the observed responses.

Response: Thank you for this suggestion. We have added the discussion to explain the roles of UPB1 on the ROS hemostasis in the epidermal cells of leaves.

7) The authors cited reference 29 to say that “We found that the expression levels of several ROS-scavenging genes, such as CAT2 and APX1, were lower in the meristemoid cells than those in other types of cells”. Where are the protein products of CAT2 and APX1 located? Did the authors or reference 29 demonstrate a change in the expression of other isoforms of catalase and ascorbate peroxidase? Which other ROS-scavenging genes showed a lower expression in meristemoid cells? The authors could provide in a supplemental table the identification (including predicted subcellular location) of the main ROS-scavenging genes (in addition to CAT2 and APX1) whose expression are reduced in meristemoid cells (as provided by reference 29). They should further justify why did they specifically choose CAT2 and APX among these genes in the list to test through the generation of transgenic plants.

Response: Thank you for pointing this. We have added the expression patterns of ROS-related genes in the Supplementary dataset 2.

8) Lines 384-392 of the discussion repeats in part some of the information already mentioned in the previous paragraph of the discussion. The authors could rework these sentences to avoid repetition.

Response: Thank you for pointing this. We have changed it.

Reviewer #3 (Remarks to the Author):

Reviewer 's Remarks to Authors:

Shi, Wang and coworkers have studied the link between the distribution of H₂O₂ in leaves of seedlings and the stomatal development in Arabidopsis. The authors found that the concentration of H₂O₂ is higher in meristemoids cells, which are the precursor cells of stomatal guard cells, than in pavement cells. This finding was in line with reduced activity of H₂O₂-scavenging enzymes like CATALASE2 and ASCORBATE PEROXIDASE1. Moreover, through fluorescence microscopy techniques and COIP, they demonstrated that SPEECHLESS (SPCH), a transcription factor associated with stomatal development, is the regulator of these enzymes and, then, responsive to H₂O₂ concentration pattern. Finally, and as a follow-up of a previous publication of the team, SnRK1.1, aka KIN10; is indicated as a key player of this developmental programme by stabilizing SPCH. The authors conclude that H₂O₂ pattern in epidermal cells is central to orchestrate the development of stomatal cells. This study pursues to shed light on the stomatal development process by identifying 2 of its key players, KIN10 and SPCH and conferring a pivotal role to the levels of H₂O₂ in their coordinated function. The article presents interesting results; for instance, revealing SPCH is able to bind to the CAT2 and APX1 promoters or the effect of mutations in KIN β subunits on the Stomatal index. As mentioned above, the rationale behind this work is to delve into the mechanistic aspects of the link between the energy gauge mediated by SnRK1.1 and modulation of stomatal development in response to changing environments previously informed by Han et al in this Journal last year (NATURE COMMUNICATIONS | (2020) 11:4214 | <https://doi.org/10.1038/s41467-020-18048-w>). This reviewer thinks

that the article is correctly presented and the topic is highly relevant by its potential to understand adaptive mechanisms involving the tuning of gas exchange in plants to external conditions. The deepening in the comprehension of KIN10 and SPCH interaction provides knowledge to prime the design of new approaches to re-adapt photosynthesis in a current adverse climate scenario.

Response: Thank you very much for your summarizing our work. We appreciate your comments and suggestions, which are very helpful for us to improve our manuscript. The revisions following your suggestions in current version are highlighted by blue color, and the point-to-point responds to your comments are followed.

At the same time, some concerns detailed below point to two caveats of the work: first, inaccurate assumptions regarding the pharmacological approaches and, second, the chosen quantification approach to analyse the observed phenomena is not robust enough to draw the presented conclusions.

The first issue is regarding the identification of H₂O₂ as the secondary messenger, whose levels are related to the enrichment of KIN10 in the nucleus. The authors performed a 12 hours-DCMU treatment in ½ MS medium on 4 days seedling, and determined that a) KIN10 presents a mostly nuclear localization in meristemoid cells and Stomatal lineage ground cells, while in pavement cells is mainly localized in the cytoplasm, b) DCMU triggers an increase in the nuclear fraction of KIN10 in all epidermal cell types (the stomatal lineage cells, pavement cells and guard cells) and c) DCMU increases H₂O₂ levels in roots (H₂DCFDA) and leaves (H₂DCFDA, DAB and BES-H₂O₂-AC staining). The identification of H₂O₂ as the trigger of nuclear localization of KIN10 upon DCMU usage might be biased by the temporal frame of the experiment (Mune Bosch 2021 November 05, 2021 DOI:<https://doi.org/10.1016/j.tplants.2021.10.005>). DCMU is a herbicide, aka DIURON that binds to the QB site of PSII, avoiding electron transfer to Plastoquinone (Mackay and O'Malley ; Z Naturforsch C J Biosci . Mar-Apr 1993;48(3-4):291-8). Its main effect in PSII is the blockage of photochemical reactions downstream of QA and DCMU produces a concomitant increase in the lifetime of the primary radical pair

(3P680+Pheo-) (Krieger-Liszkay J Exp. Bot. 56 (411), Jan 2005, 337–346). This scenario provokes a boost in the Singlet oxygen production in photosynthesis, one of the most studied signaling ROS- molecules (Flors et al. J. Exp. Bot., 57(8) May 2006, 1725–1734; Dogra and Kim Front. Plant Sci., 08 January 2020). The kinetics of DCMU in illuminated adult leaves, at concentrations similar to the treatment used in the draft, demonstrate that, as early as 30 minutes, there is an induction in the production of singlet oxygen (tracked by SOSG fluorescence, Flors et al 2006). After 90 min, the whole leaf area is producing singlet oxygen, which leads to i) a profound change in gene expression (multiple studies, e.g. op den Camp et al. (2003) Plant Cell 15, 2320–2332) and ii) a wide range of other (with longer lifespan) ROS species including H₂O₂ but also volatile signaling molecule β -cyclocitral (β -CC) and products of lipid peroxidation. Then, addressing H₂O₂ as the ROS molecule involved in relocalization of KIN10 after 12hrs of treatment with DCMU is not completely correct. This is even more complex if we consider seedling of 4 days whose photosynthetic machinery is far from being fully assembled and are more prone to suffer photoinhibition at PSII and photobleaching. The opinion of this reviewer is that the chosen pharmacological approach to evaluate KIN10 intracellular redistribution (full blocking of photosynthesis for 12 h in seedlings in the middle of the transition to autotrophy) is closer to a harsh stress treatment than a developmental reprogramming signaling leading to an increased gas exchange by higher stomatal index. As suggested option, the usage of controlled or brief treatment with methyl viologen (inducing superoxide) in combination with mutants in catalase and/or peroxidase can be more appropriate to mimic an increase in H₂O₂ concentrations.

Response: Thank you very much for your excellent suggesting and the alternative research strategies. We agree with you that DCMU has the profound effects on the ROS production, gene expression and plant growth. To rule out the side effects of DCMU, we carried out two sets of experiments as followed. First, we have analyzed the effects of DCMU on H₂O₂ accumulation and KIN10 nuclear localization through spraying DCMU for different times. The results showed that treatment with DCMU for 30 mins induced the H₂O₂ accumulation and resulted in the enriched nuclear localization of

KIN10, and the promoting effects of DCMU were enhanced with the increasing treatment time (Supplementary Fig. 3a-d). Second, we analyzed the KIN10 subcellular localization and stomatal development in the presence or absence of MV and DMTU, which are the specific superoxide anion ($O_2^{\cdot-}$)-generating reagent and the scavenger of free radicals, respectively (Hassan and Fridovich, 1977; Kim et al., 2002). The results showed that MV and DMTU had no dramatic effects on the subcellular localization of KIN10 and the stomatal development (Supplementary Fig. 5a-d and Supplementary Fig. 17). These results suggested that H_2O_2 , but not $O_2^{\cdot-}$ is specifically enriched in meristemoid to promote the nuclear localization of KIN10, resulting the increased stomatal index.

Supplementary Fig. 3 DCMU promotes the H_2O_2 accumulation and induces the nuclear localization of KIN10.

a-b, H_2DCFDA staining in the epidermal cells of leaves in response to DCMU. Seedlings of Col-0 were grown on $\frac{1}{2}$ MS solid medium with 1% sucrose under 16h light/8h dark photoperiod with $100 \mu\text{Mol/m}^2/\text{s}$ for 4 days, then sprayed with or without $50 \mu\text{M}$ DCMU for different times. **c-d**, DCMU induced the nuclear localization of KIN10. Seedlings of *pKIN10::KIN10-YFP* were grown on $\frac{1}{2}$ MS solid medium containing 1% sucrose under 16h light/8h dark photoperiod with $100 \mu\text{Mol/m}^2/\text{s}$ for 4 days, and then sprayed with or without $50 \mu\text{M}$ DCMU for different times. Fluorescent signals were taken using LSM700 microscope from Zeiss. signal from more than 200 stomatal lineage cells in 10 cotyledons were analyzed by ImageJ software. Scale bars in confocal images represent $10 \mu\text{m}$. Different letters above the bars indicated statistically significant differences between the samples (Two-way ANOVA analysis followed by Uncorrected Fisher's LSD multiple comparisons test, $p < 0.05$).

Supplementary Fig. 5 MV and DMTU have no significant effects on the nuclear localization of KIN10-YFP.

a-d, Confocal microscopic analysis of subcellular localization of KIN10 in the cotyledon epidermal cells of plants treated with different chemicals. Seedlings of *pKIN10::KIN10-YFP* were grown on ½ MS solid medium with 1% sucrose under 16 h light/8h dark photoperiod with 100 μMol/m²/s for 4 days, and then treated with or without 1 μM MV and 10mM DMTU for 6 hours.

Fluorescent signals were taken using LSM880 microscope from Zeiss. Nuclear and cytoplasmic KIN10-YFP signal from more than 200 epidermal cells in 10 cotyledons were analyzed by ImageJ software. Scale bars in confocal images represent 10 μm in (a) and in (c) represent 20 μm. Different letters above the bars indicated statistically significant differences between the samples (Two-way ANOVA analysis followed by Uncorrected Fisher’s LSD multiple comparisons test, $p < 0.05$).

Supplementary Fig. 17 MV and DMTU treatment have no marked effects on the stomatal development.

a, Quantification of the effects of MV on stomatal index. **b**, Quantification of the effects of DMTU on stomatal index. Seedlings of wild type Col-0 were grown in ½ MS solid medium containing different concentrations of MV or DMTU under 16 h light/8h dark photoperiod with 100 μMol/m²/s for 8 days. Error bars indicate standard deviation (S.D.) (n =12–15). Different letters above the bars indicated statistically significant differences between the samples (One-way ANOVA analysis followed by Uncorrected Fisher’s LSD multiple comparisons test, $p < 0.05$).

References:

- Hassan, H.M., and Fridovich, I. (1977). Regulation of the synthesis of superoxide dismutase in *Escherichia coli*. Induction by methyl viologen. *J Biol Chem* **252**, 7667-7672.
- Kim, Y.K., Lee, S.K., Ha, M.S., Woo, J.S., and Jung, J.S. (2002). Differential role of reactive oxygen species in chemical hypoxia-induced cell injury in opossum kidney cells and rabbit renal cortical slices. *Exp Nephrol* **10**, 275-284.

The second main remark is centred on two methodological aspects which are central in the construction of the working hypothesis of the draft: a) evaluation of KIN10 intracellular change and b) quantification of H₂O₂ concentration.

Regarding a) most of the published evidence relies on at least 2 techniques (e.g. fluorescent microscopy and fractionation/Western blot). It is fully understandable that the complexity of the system complicates any biochemical enrichment of stomatal lineage cells. At the same time, a better imaging via fluorescence microscopy is needed to confirm the conclusions proposed by the authors. Basically, the usage of LSCM leads to better resolution than normal fluorescence microscopy but requires a more detailed imaging along Z axis, and the usage of Bright field Images to support the conclusions obtained from fluorescence images (as a reference paper: for reference see. Putarjuna et al., 2019 *NATURE PLANTS* | VOL 5 | JULY 2019 | 742–754 | www.nature.com/natureplants742). Usually the focal plane of two-neighbour-cells' nuclei do not always coincide, being observed in 2 different positions in the Z axis. Then, if the analysis of their fluorescence is performed in one single image, an intrinsic underestimation of the signal of one of the cells takes place. In other words, the analysis compared one nucleus in focus with maximum fluorescence and one nucleus out of focus. This situation is even more pronounced in comparisons between guard cells/meristemoid cells and pavement cells. In the case of the last ones, the vacuole occupancy of the majority of cell space “push” the nucleus to the abaxial position of the cell. At the same time, guard cells are the epidermal cells with the more adaxial position in Z axis. Then, it is really complicated with only one single image to determine that 2 cells have different KIN10-GFP distribution. For example, the “presumed/assumed” enriched nuclear-localization of KIN10 in meristemoid and

stomatal lineage ground cells, to promote stomatal development, compared to mainly cytoplasmic localization in pavement cells (Fig. 1a-d and Supplemental Fig. 2a, b, also in Han, C. et al. Nat Commun 11, 4214 (2020) might be considered more carefully. Similar assumptions based on “one-focal plane image” can be also found In Han et al 2020 (for instance, Fig 4e show a central pavement cell imaged in a focal plane in Z axis in a more adaxial position with respect to the nucleus position). This limitation in the analysis based on “single fluorescence microscopy image” can be partially corrected by using BF images and/or an organelle marker in a second channel of fluorescence to localize unambiguously the position of the nucleus in each cell. Either a nuclear marker or an ER marker, with the associated perinuclear ring structure, allow to clearly identify nucleus position, and are options to conduct these experiments. Despite this approach, the recommended analysis should be performed using different ROIs for each cell over a set of Z stack images along the Z axis and FIJI associated-plugins (e.g. Costes et al., Biophys J. 2004 Jun;86(6):3993-4003) to ensure unbiased, highly stringent analysis of the fluorescence intensities during the quantification process.

Response: Many thanks for this excellent suggestion. We have reanalyzed the fluorescent signals of KIN10 in the epidermal cells of leaves using the z-stack projection images (Fig. 1b-i, Fig. 5g-h, Supplementary Fig. 1a-f, Supplementary Fig. 4a-d, Supplementary Fig. 5a-b, and Supplementary Fig. 6a-b). The serial z-stack confocal images covering the entire meristemoid cells and SLGC cells were subjected to surface rendering the YFP channel to capture *pKIN10::KIN10-YFP*-expressing nuclei in the three-dimensional space. The results showed that the nuclear-localized KIN10 proteins were highly enriched in the stomatal lineage cells under all examined conditions, which is consistent with our previous results. Besides, we also generated the *pKIN10::KIN10-YFP/p35S::Histone3-RFP* plants to analyzed the location of KIN10 and the H₂O₂ effects on the nuclear localization of KIN10, in which *p35S::Histone3-RFP* was used as a nuclei marker. The results showed that the nuclear localized KIN10 proteins were highly enriched in the stomatal lineage cells(Supplementary Fig. 7a-f).

Supplementary Fig. 7 Quantification of H_2O_2 effects on the subcellular localization of KIN10-YFP and Histone3-RFP in the *pKIN10::KIN10-YFP /p35S::H3-RFP* transgenic plants.

a-d, The subcellular localization of KIN10-YFP and Histone3-RFP in response to H_2O_2 . Seedlings of *pKIN10::KIN10-YFP/p35S::H3-RFP* were grown on $\frac{1}{2}$ MS solid medium containing 1% sucrose under 16h light/8h dark photoperiod with $100 \mu\text{Mol}/\text{m}^2/\text{s}$ for 4 days, and then treated with or without 2 mM H_2O_2 for 3 hours. Magnifications of the epidermal cells are shown on the middle (b). The fluorescent signals of GFP and RFP were determined along a line drawn on the confocal images using ImageJ software. The arrow labeling with A and B represents the fluorescent signals in meristemoid cells, and the arrow labeling with B and C represents the fluorescent signals in SLGC.

e, Quantification of nuclear localization of KIN10 in different scale epidermal cells. **f**, Quantification of nuclear localization of Histone 3 in different scale epidermal cells.

Signals were taken using LSM700 microscope from Zeiss. Nuclear and cytoplasmic KIN10-YFP and Histone3-RFP signal from more than 200 epidermal cells in 10 cotyledons were analyzed by ImageJ software. Scale bars in panel (a) represent $20 \mu\text{m}$, and in panel (b) represent $10 \mu\text{m}$. Different letters above the bars indicated statistically significant differences between the samples (Two-way ANOVA analysis followed by Uncorrected Fisher's LSD multiple comparisons test, $p < 0.05$).

Regarding b) quantification of H_2O_2 concentration, the usage of different in situ ROS detection methods is adequate, though some aspects should be considered. Together with the above-mentioned heed regarding fluorescence microscopy imaging, the

specificity of the detection assays is crucial to infer correct conclusions (Noctor et al., 2016 *Plant, Cell and Environment* (2016) 39, 1140–1160; Smirnov and Arnaud, 2019; *New Phytologist* (2019) 221: 1197–1214). Among the different chosen methods in the draft to detect H₂O₂ (DAB; H₂DCFDA, BES-H₂O₂-Ac and pHYPER), pHyper and BES-H₂O₂-Ac can be considered the most exact approach due to their high specificity to H₂O₂ (Smirnov and Arnaud, 2019; *New Phytologist* (2019) 221: 1197–1214, Maeda et al., 2008 Volume 1130, Issue 1 *Fluorescence Methods and Applications: Spectroscopy, Imaging, and Probes*). On the other hand, Fluorescein probes like H₂DCFDA react with a wide range of radical-based reactive species, including peroxynitrite and reactive sulfur species. DAB is not suitable for live tissues and requires peroxidase activity in the tissue in study to provide fast detection assay. Then, Results obtained by using H₂DCFDA staining methods should be considered as an indicative of an increase in total ROS-levels (Supplemental Fig. 1c, d, Fig 2 a-d), which in part is due to H₂O₂. In this context, a scenario of oxidative stress in the whole leaf tissue is a plausible alternative scenario to the proposed specific induction of H₂O₂ in meristemoid cells. Unfortunately, figure 2 only indicates differences between SLGC and meristemoid cells. With this info, this reviewer thinks that the mechanistic link between H₂O₂ and KIN10 specific redistribution in meristemoid cells is not sufficiently demonstrated.

Response: Thank you for pointing this. We agree with your opinion that BES-H₂O₂-Ac and *pHyPer* exhibit much higher H₂O₂ specificity than H₂DCFDA. In the revised manuscript, we have performed measurements of H₂O₂ with BES-H₂O₂-Ac and *pHyPer* for the rosette leaf epidermal cells of plants. Consistently, the results showed that H₂O₂ is highly enriched in the stomatal lineage cells and guard cells of plants, indicating that this spatial distribution pattern of H₂O₂ is widespread in the epidermal cells of *Arabidopsis* leaves.

Supplementary Fig. 11 H₂O₂ specifically accumulates in the stomatal lineage cells of rosette leaves.

a-b, BES-H₂O₂-Ac staining for H₂O₂ in the epidermal cells of rosette leaves. The BES-H₂O₂-Ac fluorescent signals were in green, PI-marked cell outlines were in purple. Seedlings of Col-0 were grown on ½ MS solid medium containing 1% sucrose under 16h light/8h dark photoperiod with 100 μMol/m²/s for 12 days. **c-d**, Quantification of *pHyPer* fluorescent signals in rosette leaves epidermal cells. Ratio imaging of Arabidopsis epidermal cells expressing *HyPer*. Seedlings of *pHyPer* transgenic plants were grown on ½ MS solid medium containing 1% sucrose under 16h light/8h dark photoperiod with 100 μMol/m²/s for 12 days.

Fluorescent signals were taken using LSM700 microscope from Zeiss. signal from more than 200 stomatal lineage cells in 10 rosette leaves were analyzed by ImageJ software. Scale bars in confocal images represent 20 μm. Asterisk between the bars indicated statistically significant differences between the samples (Student's t test, ****p < 0.0001).

Reviewer #4 (Remarks to the Author):

This manuscript aims to show that SPEECHLESS, the transcription factor that drives stomata development, represses the expression of antioxidant enzyme genes (CAT2, APX), leading to the accumulation of ROS specifically in the cells where SPCH is expressed (meristemoids and SLGC). Increased ROS levels in turn lead to the nuclear translocation of the KIN10 kinase, which in previous work was shown to stabilize SPCH.

The subject of this manuscript is really interesting and the model presented intriguing. Unfortunately, the study lacks robust evidence to support many of the statements.

Main concerns:

- The authors show compelling evidence that ROS accumulates to higher levels in

meristemoid cells. However, all other aspects are extremely difficult to evaluate, as there is not sufficient information provided about the materials and conditions, and when it is provided it changes from one experiment to another. Some of the factors that seem to vary amongst experiments are: presence/absence of sucrose in the media, use of solid vs. liquid media, age of seedlings (see more details below).

Response: Sorry for the confusing description. We have added the information on the growth conditions in the figure legends and Materials and Methods section.

- I believe the statistics used in most of the analyses should be revisited, as there is no correction for multiple comparisons.

Response: Thank you for pointing this. We have changed it.

- The authors try to cover an enormous amount of aspects (some of them only described in the Discussion) but, because of that, most of them remain very superficial and lack robustness.

Response: Thank you for pointing this. Considering the data about the regulation of BIN2 activity by H₂O₂ is not closely related to our major statement, we have deleted this part in the revised manuscript. However, the data about the superoxide anion (O₂^{•-}) regulation of stomatal development is closely related to our work, we moved them to the Result section and performed more experiments to prove that O₂^{•-} evenly distributed in the epidermal cells of leaves, and had no significant effects on KIN10 subcellular localization and stomatal development.

- In many instances authors use a single transgenic line or KO mutant to address a specific aspect rather than the two independent lines that is usually desired to confidently associate a phenotype with a particular modification. In addition, a basic characterization of these lines is missing (to show e.g. that they are indeed overexpressors or knockout mutants, etc).

Response: Thank you for pointing this. We have analyzed the stomatal phenotypes and KIN10 localization pattern using two independent transgenic lines. Furthermore, we have analyzed the expression levels of corresponding genes in these materials (Rebuttal Fig. 5).

Rebuttal Fig. 5 The information of related mutants

a, Schematic diagram showing the T-DNA insertion sites in the genomic region.

b, RT-PCR analysis of the expression of the related genes in Col-0 and the related seedlings. Seedlings of Col-0 and the related seedlings were grown in ½ MS solid medium containing 1% sucrose under 16h light/8h dark photoperiod with 100 μMol/m²/s in the presence of for 7 days. PP2A was used as the internal control.

- Regarding the impact of DCMU/H₂O₂ on KIN10 localization I have the concern that this is rather an effect on protein accumulation (either specific to KIN10 or more general one).

Response: Thank you for pointing this. H₂O₂ treatment indeed induced the KIN10 protein accumulation. To determine whether H₂O₂ promotes the nuclear localization of KIN10, we analyzed the subcellular localization of KIN10 in the presence H₂O₂ and/or protein translation inhibitor CHX. The results showed that CHX treatment had no significant effects on the H₂O₂-induced nuclear localization of KIN10, suggesting the H₂O₂-promoted nuclear localization of KIN10 is independent on the new protein biosynthesis of KIN10 (Fig. 1e-g). To further determine the specific effects of H₂O₂ on the subcellular localization of KIN10, we analyzed the H₂O₂ effects on the nuclear localization of p35S::GFP that was used as control. The results showed that H₂O₂ had no significant effects on the subcellular localization of GFP (Rebuttal Fig. 6a-c).

Rebuttal Fig. 6 H₂O₂ had no significant effects on the subcellular localization of *p35S::GFP* in plants.

a-c, Seedlings of *p35S::GFP* were grown on ½ MS solid medium containing 1% sucrose under 16 h light/8h dark photoperiod with 100 μMol/m²/s for 4 days, and then treated with or without 2 mM H₂O₂ for 3 hours. Fluorescent signals were taken using LSM700 microscope from Zeiss. Nuclear and cytoplasmic *p35S::GFP* signal from more than 200 stomatal lineage cells in 10 cotyledons were analyzed by ImageJ software. Different letters above the bars indicated statistically significant differences between the samples (Two-way ANOVA analysis followed by Uncorrected Fisher’s LSD multiple comparisons test, *p* < 0.05). Scale bars in confocal images represent 20 μm.

- Regarding the interaction of KIN10 with the subunits, the activity of KIN10 should be shown to demonstrate that the peroxide treatment is not simply denaturing the proteins and that is why the interaction is lost.

Response: Thank you for pointing this. To determine the effects of H₂O₂ on the KIN10 activity, we analyzed the kinase activity of KIN10 in the presence of H₂O₂. The results showed that H₂O₂ had no significant effects on KIN10 kinase activity, suggested that H₂O₂ inhibited the interaction between KIN10 and KINβ2 not due to the denature of KIN10 (Rebuttal Fig. 7).

Rebuttal Fig. 7 H₂O₂ had no significant effects on the KIN10 kinase activity.

Quantification of the effects of H₂O₂ on KIN10 kinase activity. Seedlings of *p35S::KIN10-Myc* transgenic plants were grown on ½ MS solid medium containing 1% sucrose under 16h light/8h dark photoperiod with 100 μMol/m²/s for 7 days, then treated with or without 2 mM H₂O₂ for different times.

What is the activity of KIN10 when treated or not with this H₂O₂ concentration (alone and in combination with the beta subunit)?

Response: Thank you for pointing this. We have analyzed the H₂O₂ effects on the subcellular localization of KIN10-GFP using the transient expression system and found that H₂O₂ significantly increased the nuclear/cytoplasmic ratio of KIN10 (Fig. 5e-f), which was consistent with the results using the stable *pKIN10:KIN10-YFP* transgenic plants with or without H₂O₂ treatment in the figure 1e-g.

Fig5. H₂O₂ reduces the effects of KINβ2 on the KIN10-cytoplasmic retention in tobacco leaves.

e-f, H₂O₂ reduced the effects of KINβ2 on the KIN10-cytoplasmic retention in tobacco leaves. The tobacco leaves were transformed with KIN10-GFP or co-transformed with KIN10-GFP and KINβ2-RFP constructs. After 2 days, the leaves were treated with or without 2 mM H₂O₂ for 6 hours. The fluorescent signals of GFP (KIN10) were determined using ImageJ software. (n =30 images). Scale bars in confocal images represent 20 μm. (One-way ANOVA analysis followed by Uncorrected Fisher's LSD multiple comparisons test, p < 0.05).

In more detail:

Figure 1

- Is the effect of DCMU and H₂O₂ really an effect on KIN10 localization or an effect on KIN10 accumulation in general? I think these treatments may be triggering KIN10 accumulation in other tissues/cell types. Also, is the effect shown for KIN10 a general one or specific to KIN10? Could this treatment be inhibitory to the proteasome and in that way reflect ALSO on KIN10? Authors could use as a control an NLS-GFP or RFP line to show that this is indeed KIN10-specific.

Response: Thank you for pointing this. H₂O₂ treatment indeed induced the KIN10 protein accumulation. To determine whether H₂O₂ promotes the nuclear localization of KIN10 through increasing new KIN10 protein synthesis, we analyzed the subcellular localization of KIN10 in the presence H₂O₂ and/or protein translation inhibitor CHX. The results showed that CHX treatment had no significant effects on the H₂O₂-induced nuclear localization of KIN10, suggesting the H₂O₂-promoted nuclear localization of KIN10 is independent on the new protein biosynthesis of KIN10 (Fig. 1e-g). To further determine the specific effects of H₂O₂ on the subcellular localization of KIN10, we analyzed the H₂O₂ effects on the nuclear localization of *p35S::GFP* that was used as control. The results showed that H₂O₂ had no significant effects on the subcellular localization of GFP (Rebuttal Fig. 6a-c).

Rebuttal Fig6. H₂O₂ did not change the nuclear localization of *p35S::GFP* in plants.

a-c, Seedlings of *p35S::GFP* were grown on ½ MS solid medium containing 1% sucrose under 16 h light/8h dark photoperiod with 100 μMol/m²/s for 4 days, and then treated with or without 2 mM H₂O₂ for 3 hours. Fluorescent signals were taken using LSM700 microscope from Zeiss. Nuclear and cytoplasmic *p35S::GFP* signal from more than 200 stomatal lineage cells in 10 cotyledons were analyzed by ImageJ software. Different letters above the bars indicated statistically significant differences between the samples (Two-way ANOVA analysis followed by Uncorrected Fisher's LSD multiple comparisons test, p < 0.05). Scale bars in confocal images represent 20 μm.

- How do the 35S:CAT2-MYC plants look like? These plants were generated in this study but a basic characterization of these is totally lacking. Do the authors have a second independent line that behaves similarly? Can KIN10 accumulation in the nucleus indeed be rescued in these plants by applying H₂O₂ exogenously?

Response: Thank you for pointing this. We have analyzed the expression levels of *CAT2* in two independent lines of *p35S::CAT2-Myc* transgenic plants, and found that *CAT2* was approximately 4-8 fold up-regulated in these plants (Supplementary Fig. 6c). The

ratio of nuclear/cytoplasmic KIN10 was significantly reduced in the *p35S::CAT2-Myc* plants. Consistent with this, the stomatal index of *p35S::CAT2-Myc* plants was lower than that of wild-type plants. H_2O_2 treatment suppressed the decreased-nuclear localization of KIN10 and reduced-stomatal index phenotype of *p35S::CAT2-Myc*, suggesting CAT2 scavenges H_2O_2 to reduce KIN10 nuclear localization, thereby inhibiting stomatal development (Supplementary Fig. 6a, b and (Rebuttal Fig. 8).

Supplementary Fig. 6 H_2O_2 induces the nuclear localization of KIN10.

a-b, The subcellular localization of KIN10-YFP in *pKIN10::KIN10-YFP* or *pKIN10::KIN10-YFP/p35S::CAT2-Myc* plants that were grown on $\frac{1}{2}$ MS solid medium containing 1% sucrose under 16 h light/8h dark photoperiod with $100 \mu\text{Mol/m}^2/\text{s}$ for 4 days, and then treated with or without 2 mM H_2O_2 for 3 hours. The Z-stack confocal images were taken using LSM880 microscope from Zeiss to capture *pKIN10::KIN10-YFP* expressing nuclei in the three-dimensional space. Nuclear and cytoplasmic KIN10-YFP signal from more than 200 stomatal lineage cells in 10 cotyledons were analyzed by ImageJ software. Scale bars represent $10 \mu\text{m}$. Different letters above the bars indicated statistically significant differences between the samples (Two-way ANOVA analysis followed by Uncorrected Fisher's LSD multiple comparisons test, $p < 0.05$). **c**, RT-qPCR analysis of the expression of *CAT2* in wild type and *p35S::CAT2-Myc* mutant. Seedlings for the wild-type Col-0 and *p35S::CAT2-Myc* were grown on $\frac{1}{2}$ MS solid medium containing 1% sucrose under 16 h light/8h dark photoperiod with $100 \mu\text{Mol/m}^2/\text{s}$ for 5 days. *PP2A* gene was used as an internal control. Error bars indicate standard deviation (S.D.). Different letters above the bars indicated statistically significant differences between the samples (One-way ANOVA analysis followed by Uncorrected Fisher's LSD multiple comparisons test, $p < 0.05$).

Supplementary Fig. 15 H₂O₂ treatment suppresses the decreased stomatal phenotypes of *p35S::CAT2-Myc*.

Seedlings of wild type, *p35S::CAT2-Myc* transgenic plants were grown on ½ MS solid medium containing 1% sucrose with or without 10 μM H₂O₂ under 16 h light/8h dark photoperiod with 100 μMol/m²/s for 8 days. Error bars indicate standard deviation (S.D.) (n =12–15). Different letters above the bars indicated statistically significant differences between the samples (Two-way ANOVA analysis followed by Uncorrected Fisher's LSD multiple comparisons test, p < 0.05).

- In the legend of this and other figures it is said: error bars indicate standard deviation, but these seem to be box plots and in that case the bars would represent quartiles.

Response: Thank you for pointing this. We have changed them.

- What was the material used in Suppl Fig. S1 to check that DCMU induces H₂O₂ accumulation and the counteracting effect of KI? The leaves shown in the picture look like leaves of soil-grown plants, and not of 5d-old seedlings. The effect of DCMU and DCMU+KI (Fig. S1) should be shown with the same material and under the same conditions that are used for assaying KIN10 localization and stomata development (Fig. 1).

Response: Thank you for pointing this. We have repeated these experiments using the 5-day-old plants.

- Indeed, many assays are done in MS medium + 1% sucrose. But in others, including those of Suppl Fig. 1 (also Fig. 1e-g, Fig. 4a-c, Fig. 5c-d, Suppl. Fig. 2c-d, Suppl. Fig. 8), no sucrose was used. Why? This is really important, as the previous work of the

authors showed that KIN10 promotes stomata development only in seedlings grown in 1% sucrose. Also, the presence of sugar is likely to influence ROS levels and the energy conditions have been shown to change the localization of SnRK1 (Ramon et al. 2019).
Response: Sorry for this confusing description. In the previous submitted manuscript, seedlings of wild type and indicated materials were grown on ½ MS solid medium containing 1% sucrose under 16 h light/8h dark photoperiod with 100 μM/m²/s for different days. In the revised manuscript, we have added the data about the spatial distribution of H₂O₂ and the subcellular localization of KIN10 of plants that were grown on ½ MS solid medium with or without 1% sucrose under different photoperiod or different light intensity conditions. The results showed that the phenomenon of the specific accumulation of H₂O₂ and nuclear localization of KIN10 in meristemoid cells were appeared regardless of plant growth conditions which we examined.

Figure 2:

- Please describe what BES-H₂O₂-Ac and H₂DCFDA are and what are their differences.

Response: Thank you for pointing this. 2', 7'-dichlorodihydrofluorescein diacetate (H₂DCFDA) is a widely used ROS indicator. The reduced non-fluorescent fluorescein H₂DCFDA can be oxidized and converted into fluorescent 2', 7'-dichlorofluorescein (DCF) by intracellular H₂O₂, but these compounds suffer from the major drawback that they are poorly selective toward H₂O₂ (Eljebbawi, et al., 2021). BES-H₂O₂-Ac is also a highly selective fluorescent probe for H₂O₂, and exhibits much higher H₂O₂ specificity than H₂DCFDA. BES-H₂O₂-Ac is applicable to clarifying cell response as well as dynamic function of H₂O₂ in living cells (Eljebbawi, et al., 2021).

References

Eljebbawi, A., Guerrero, Y., Dunand, C., and Estevez, J.M. (2021). Highlighting reactive oxygen species as multitaskers in root development. *iScience* **24**, 101978.

- Also, what are pHyPer plants?

Response: Thank you for pointing this. HyPer is a genetically encoded yellow fluorescent protein (YFP)-based H₂O₂ sensor, which is highly sensitive to H₂O₂ and not to other ROS in bacteria and animal cells (Belousov et al., 2006). To investigate H₂O₂ metabolism in plants, we amplify the HyPer from the pHyper-cyto plasmid (Evrogen, FP941) and cloned into pENTRTM/SD/D-TOPOTM vectors, and then recombined with destination vector pEARLY 100 (*p35S::X*). This construct was introduced into *Agrobacterium tumefaciens* (strain GV3101), and transformed into Col-0 plants to generate *p35S::HyPer* transgenic plants. Other laboratories have generated the *pHyPer* transgenic plants using the similar method and showed the H₂O₂ metabolism of plants grown under oxidative stress (Costa et al., 2010) or aluminum stress (Hernandez-Barrera et al., 2015).

References

- Belousov, V.V., Fradkov, A.F., Lukyanov, K.A., Staroverov, D.B., Shakhbazov, K.S., Terskikh, A.V., and Lukyanov, S. (2006). Genetically encoded fluorescent indicator for intracellular hydrogen peroxide. *Nat Methods* **3**, 281-286.
- Costa, A., Drago, I., Behera, S., Zottini, M., Pizzo, P., Schroeder, J.I., Pozzan, T., and Lo Schiavo, F. (2010). H₂O₂ in plant peroxisomes: an in vivo analysis uncovers a Ca(2+)-dependent scavenging system. *Plant J* **62**, 760-772.
- Hernandez-Barrera, A., Velarde-Buendia, A., Zepeda, I., Sanchez, F., Quinto, C., Sanchez-Lopez, R., Cheung, A.Y., Wu, H.M., and Cardenas, L. (2015). Hyper, a hydrogen peroxide sensor, indicates the sensitivity of the Arabidopsis root elongation zone to aluminum treatment. *Sensors (Basel)* **15**, 855-867.

• Legend of Fig. 2: “H₂O₂ specifically accumulated in meristemoid cells”. The authors say themselves in the text that it is also in guard cells (lines 165-167), so please correct.

Response: Thank you for pointing this. We have changed it.

• Information on growth conditions used for these assays is totally missing.

Response: Sorry for this missing. We have added the information on growth conditions in the figure legends.

Figure 3

• There is no information nor basic characterization about the pCAT2:GFP or

pAPX:GFP lines. How was the promoter of these lines determined? How can we know that indeed these reporters reflect the expression of the corresponding genes?

Response: Thank you for pointing this. According to the gene structure given by TAIR, we selected the 2632 kb and 2000 kb as the promoters of *CAT2* and *APX1*, respectively. The published transcriptomic data showed that *CAT2* and *APX1* were highly expressed in the pavement cells, but lower in the stomatal lineage cells. Consistent with this, the *pCAT2::GFP* and *pAPX1::GFP* displayed the weak fluorescent signals in the stomatal lineage cells and high signals in the pavement cell, suggesting the promoter fragments of *CAT2* and *APX1* really represent the activity of native promoters of *CAT2* and *APX1*.

- In the chromatin-IP experiment of Fig. 3g (and Fig. S5) it is hard to conclude much without having controls. A TF will bind DNA as compared to a non-TF protein like YFP. The key here is to show that this binding is specific for that particular sequence. For example, showing that SPCH binds to the APX promoter as opposed to another promoter or as opposed to the coding region of APX. Related to the same figures, which regions of the promoters were used? Are these the regions that were shown to be bound by SPCH in the reference #30? Also, it is not clear from the supplementary table and methods description whether the same regions were amplified also for the DNA-protein pulldowns.

Response: Thank you for pointing this. We performed the Chromatin-IP experiments using the *p35S::SPCH-YFP* and *p35S::YFP* transgenic plants. The fragments of *CAT2* and *APX1* promoter that have been reported to associate with SPCH in the ChIP-Seq data of SPCH were selected for quantitative ChIP-PCR analysis in the previous manuscript. In the revised manuscript, we have analyzed the enrichment of SPCH on the other fragments of *CAT2* and *APX1* promoters, the results showed that SPCH bound to the fragment C and D of *CAT2* promoter (Fig. 3h), and the fragment A and C of *APX1* promoter (Supplementary Fig. 13e). To further verify the association of SPCH with the promoters of *CAT2* and *APX1*, we carried out the DNA-protein pull down assays. The results showed that SPCH directly bound to the D of *CAT2* promoter, and the fragment C of *APX1* promoter, but not the promoter of *PP2A* (Fig. 3i and Supplementary Fig.

13d). These results indicated that SPCH directly binds to the promoters of *CAT2* and *APX1*.

Fig. SPCH directly binds the promoters of *CAT2* and *APX1*.

a, Quantitative ChIP-PCR showed that SPCH binds to *CAT2* promoter. Seedlings of *p35S::YFP* and *p35S::SPCH-YFP* were used to performed ChIP assays. The levels of SPCH binding were calculated as the ratio between *p35S::SPCH-YFP* and *p35S::YFP*, and then normalized to that of control gene *PP2A*. Error bars indicate standard deviation (S.D.). * $P < 0.05$, as determined by a Student's t-test.

b, Quantitative ChIP-PCR showed that SPCH binds to *APX1* promoter. Seedlings of *p35S::YFP* and *p35S::SPCH-YFP* were used to performed ChIP assays. The levels of SPCH binding were calculated as the ratio between *p35S::SPCH-YFP* and *p35S::YFP*, and then normalized to that of control gene *PP2A*. Error bars indicate standard deviation (S.D.). * $P < 0.05$, as determined by a Student's t-test.

Figure 4

- Even though I understand the point of the authors in introducing new loss-of-function *cat* mutants to the study, it would be important to have here the *35S::CAT2* line used in Fig. 1. Since that line was used for assessing the effect of ROS on KIN10 localization it would be important to know how it behaves regarding stomata development. Do these plants have reduced stomata numbers?

Response: Thank you for pointing this. We have analyzed the stomatal phenotypes of wild-type plants, *p35S::CAT2-Myc* and *pSPCH::CAT2-RFP* transgenic plants, and found the stomatal index of *p35S::CAT2-Myc* was lower than that of wild-type plants, but higher than in the *pSPCH::CAT2-RFP* transgenic plants (Supplementary Fig. 15b). These results showed that overexpression of *CAT2* inhibited stomatal development, and specifically increased *CAT2* expression in stomatal lineage cells enhanced the inhibiting effects of *CAT2* on stomatal development, suggesting that the specific

reduction of H₂O₂ in meristemoid cells has more dramatic effects on stomatal development.

Supplementary Fig. 15 H₂O₂ treatment suppressed the decreased stomatal phenotypes of *p35S::CAT2-Myc* and *pSPCH::CAT2-RFP* plants.

Seedlings of wild type, *p35S::CAT2-Myc* and *pSPCH::CAT2-RFP* transgenic plants were grown on ½ MS solid medium containing 1% sucrose with or without 10 μM H₂O₂ under 16 h light/8h dark photoperiod with 100 μMol/m²/s for 8 days. Error bars indicate standard deviation (S.D.) (n =12–15). Different letters above the bars indicated statistically significant differences between the samples (Two-way ANOVA analysis followed by Uncorrected Fisher’s LSD multiple comparisons test, p < 0.05).

- The conditions are also different in this figure, as no sucrose seems to have been used.

Response: Sorry for this misleading description. In the figure 4a, seedlings of wild-type plants and indicated materials were grown on the ½ solid MS medium containing 1% sucrose under 16 h light/8h dark photoperiod with 100 μM/m²/s for 5 days.

- The authors say in the text (lines 261-263) “The amount of meristemoid cells and GMC in the abaxial epidermis at 5days after germination (DAG) were significantly increased in *cat2*, *cat3*, *cat2 cat3*, *cat1 cat2 cat3* mutants compared to that in the wild type (Fig. 4a, b)”. However, there are no statistical analysis being performed to back up that statement of significance.

Response: Thank you for pointing this. We have performed the statistical analysis and found that the numbers of meristemoid cells and GMC were significantly increased in

cat2, *cat3*, *cat2 cat3*, and *cat1 cat2 cat3* mutants, but decreased in *p35S::CAT2-Myc* and *pSPCH::CAT2-RFP* transgenic plants compared to that in wild type (Fig. 4a, b).

Fig.4 H₂O₂ promotes stomatal development.

a, Quantification of epidermal cell types of Col-0 and indicated plants, expressed as percentage of total cells. GMC, guard mother cell; M, meristemoid (n =12–15). Seedlings of wild-type plants and indicated materials were grown on the ½ MS solid medium containing 1% sucrose under 16 h light/8h dark photoperiod with 100 μM/m²/s for 5 days. Scale bars in confocal images represent 20 μm.

- In Fig. 4c they use solid medium and the plants were grown for 8d as opposed to the 4-5 days of other experiments.

Response: Thank you for pointing this. For the cotyledon development at 4-5 days after germination, there are various cell types such as meristemoid cell, SLGC cell, GMC cell, guard cell and pavement cell, and the ratios of these cells to the total cell number of cotyledons were relatively high. However, at 8 days after germination, the main cell types of cotyledons are guard cell and pavement cell. In the figure 4a-b, we analyzed the effects of CATs on the cell fate determination in the cotyledon, and we analyzed the H₂O₂ effects on the stomatal index of wild type and *kin10* mutant in the figure 4c, so we used the cotyledon of plants grown for different days in these two experiments.

Do these H₂O₂ treatments lead to different ROS concentrations that can indeed be correlated with the observed stomata indexes? Also, is H₂O₂ stable in plates for so many days in the light? Showing ROS levels would support that H₂O₂ is stable and is having the desired effect.

Response: Thank you for pointing this. To verify the effects of exogenous H₂O₂ on stomatal development, we analyzed the H₂O₂ content of leaves of wild type and *kin10* grown on the medium containing different concentrations of H₂O₂. The results showed that exogenous H₂O₂ treatment indeed increased the H₂O₂ contents in plants (Rebuttal Fig.8a, b)

Rebuttal Fig.8 Quantification of H₂O₂ in wild-type plants and *kin10* mutant in the presence of different concentrations of H₂O₂.

a,b, Seedlings of wild-type plants and *kin10* mutant were grown on ½ MS solid medium containing 1% sucrose with indicated H₂O₂ under 16 h light/8h dark photoperiod with 100 μMol/m²/s for 8 days. Fluorescent signals were taken using LSM700 microscope from Zeiss. The fluorescent signal from more than 100 guard cells in 10 cotyledons were analyzed by ImageJ software. Scale bars in confocal images represent 20 μm. Different letters above the bars indicated statistically significant differences between the samples (Two-way ANOVA analysis followed by Uncorrected Fisher’s LSD multiple comparisons test, p < 0.05).

- In Fig. 4d-g the seedlings are 3d old. It is very difficult to compare results from 3d old seedlings with those from 8d old seedlings, in particular when this involves the transition from heterotrophy to autotrophy.

Response: Thank you for pointing this. Consistent with the above reasons, there are more SPCH-expressed meristemoid and SLGC cells in the 3-day-old seedlings than that of 8-day old seedlings, and it is easy to determine the effects of CATs on the cell fate determination of stomatal lineage cells.

- Related to Fig. 4d-g the text claims in lines 304-308 “We observed that mutation of CAT2 and CAT3 dramatically increased the number of cells marked by pSPCH::nucGFP and pSPCH::SPCH-GFP compared to those in a wild type background (Fig. 4d-g). These results indicated H₂O₂ alters the epidermal cell fates in

the Arabidopsis leaves”. This is quite an overstatement. Lack of CAT2/CAT3 can have numerous consequences beyond increased ROS levels and those could be causing the changes in SPCH. To have such a strong statement the authors should demonstrate that indeed this is due to increased ROS and could perhaps use a ROS scavenger like KI to revert the effect of the CAT mutations.

Response: Thank you for pointing this. We have analyzed the expression patterns of *pSPCH::nucGFP*, *pSPCH::nucGFP/cat2cat3*, *pSPCH::SPCH-GFP*, and *pSPCH::SPCH-GFP/cat2cat3* in the presence or absence of the ROS scavenger KI. The results showed that mutation of *CAT2* and *CAT3* dramatically increased the number of cells marked by *pSPCH::nucGFP* and *pSPCH::SPCH-GFP* compared with those in wild-type background, and KI treatment suppressed such promoting effect (Fig. 4d-g). These results indicated H_2O_2 reduces the expression and protein levels of SPCH, thus resulting in the changes of epidermal cell fates in the Arabidopsis leaves.

Fig.4 H_2O_2 promotes stomatal development.

d-g. *CAT* mutations altered cell fate in Arabidopsis epidermis. Seedlings of *pSPCH::nucGFP*, *pSPCH::nucGFP/cat2 cat3*, *pSPCH::SPCH-GFP*, and *pSPCH::SPCH-GFP/cat2 cat3* were grown on 1/2 MS medium containing 1% sucrose with or without 1mM KI under 16 h light/8h dark photoperiod with $100 \mu M/m^2/s$ for 3 days. Error bars indicate standard deviation (S.D.) (n =12–15). Scale bars in confocal images represent 20 μm . Different letters above the bars indicated statistically significant differences between the samples (One-way ANOVA analysis followed by Uncorrected Fisher’s LSD multiple comparisons test, $p < 0.05$).

Figure 5

• The authors employ in these assays a high concentration of peroxide that may denature the proteins and as consequence of that the interaction is lost. Is KIN10 still active after such treatment? Are other protein interactions not broken under such conditions?

Response: Response: Thank you for pointing this. To determine the effects of H₂O₂ on the KIN10 activity, we analyzed the kinase activity of KIN10 in the presence of H₂O₂. The results showed that H₂O₂ had no significant effects on KIN10 kinase activity, suggested that H₂O₂ inhibited the interaction between KIN10 and KINβ2 not due to the denature of KIN10 (Rebuttal Fig. 9a). To determine whether H₂O₂ denatures the proteins and as consequence of the interaction is lost, we analyzed the interaction between BZR1 and PIF4, which is also regulated by H₂O₂. The results showed that H₂O₂ induced the interaction of BZR1 with PIF4. These results indicated H₂O₂ has different regulatory effects on different proteins (Rebuttal Fig. 9b).

Rebuttal Fig. 9 H₂O₂ had no significant effects on the KIN10 kinase activity.

a, Quantification of the effects of H₂O₂ on KIN10 kinase activity. Seedlings of *p35S::KIN10-Myc* transgenic plants were grown on ½ MS solid medium containing 1% sucrose under 16h light/8h dark photoperiod with 100 μMol/m²/s for 7 days, then treated with or without 2 mM H₂O₂ for different times. **b**, H₂O₂ induced the interaction between BZR1 and PIF4 in vitro. MBP and MBP-BZR1 were incubated with GST-PIF4 bound to glutathione agarose beads with or without 2 mM H₂O₂ for 1 h, and then eluted and analyzed by immunoblotting using anti-MBP antibody.

• Does the H₂O₂ treatment lead to nuclear localization of KIN10 when it is expressed alone as KIN10-GFP? What was the H₂O₂ concentration used in this assay?

Response: Thank you for pointing this. We have analyzed the H₂O₂ effects on the subcellular localization of KIN10-GFP using the transient expression system and found that H₂O₂ significantly increased the nuclear/cytoplasmic ratio of KIN10 (Fig. 5e, f), which was consistent with the results using the stable *pKIN10:KIN10-YFP* transgenic plants with or without H₂O₂ treatment in the figure 1e-g and Supplementary figure 4c-d. The concentration of H₂O₂ in this assay was 2 mM.

Fig.5 H₂O₂ reduces the effects of KINβ2 on the KIN10-cytoplasmic retention in tobacco leaves.

e-f, H₂O₂ reduced the effects of KINβ2 on the KIN10-cytoplasmic retention in tobacco leaves. The tobacco leaves were transformed with KIN10-GFP or co-transformed with KIN10-GFP and KINβ2-RFP constructs. After 2 days, the leaves were treated with or without 2 mM H₂O₂ for 6 hours. The fluorescent signals of KIN10-GFP were determined using ImageJ software. (n =30 images). (One-way ANOVA analysis followed by Uncorrected Fisher's LSD multiple comparisons test, p < 0.05). Scale bars in confocal images represent 20 μm.

- To the best of my knowledge these KINbeta lines have not been previously described so a proper characterization is needed to show that these are bonafide loss-of-function mutants.

Response: Thank you for pointing this. We have added the details of these mutants in the Materials and Methods section and analyzed the expression levels of *KINβ1* and *KINβ2* in the *kinβ1 kinβ2* mutant. The results showed that the expression of *KINβ1* was

undetectable, and the expression of *KINβ2* was significantly decreased in the *kinβ1 kinβ2* double mutant.

Supplementary Fig. 20 The information of *KINβ1* and *KINβ2* T-DNA mutants.

a, Schematic diagram showing the T-DNA insertion sites in the genomic region *KINβ1* and *KINβ2*, respectively.

b, RT-PCR analysis of the expression of *KINβ1* and *KINβ2* in Col-0 and *kinβ1 kinβ2* mutants. Seedlings of Col-0 and *kinβ1 kinβ2* mutants were grown in ½ MS solid medium containing 1% sucrose under 16h light/8h dark photoperiod with 100 μMol/m²/s in the presence of for 7 days. *PP2A* was used as the internal control.

OTHERS

- The paper cited in line 86 has no data regarding nuclear translocation of KIN10

Response: Thank you for pointing this. We have changed it.

- Line 508. Correct chromosome to chromatin

Response: Thank you for pointing this. We have changed it.

- English language should in general be revised

Response: Thank you for pointing this. We have edited our manuscript by the professional English editor.

- The manuscript could be better structured, as it starts with KIN10, in the middle shows the effects of ROS on stomata, and then goes back to KIN10 through the interaction with the beta subunits

Response: Thank you for pointing this. The main focus of this manuscript is that the spatiotemporal pattern of H₂O₂ orchestrates stomatal development. The logic of this manuscript is that we found the spatiotemporal pattern of H₂O₂ in the epidermal cells

of leaves through analyzing the subcellular localization of KIN10 in epidermis, then we investigate how this pattern of H₂O₂ is formed in leaves and what is the effects of this pattern on stomatal development, and finally we investigate how H₂O₂ regulates KIN10 activity to modulate stomatal development.

REVIEWER COMMENTS

Reviewer #1 (Remarks to the Author):

Authors have made an impressive effort to revise the manuscript. Most of my comments have been addressed in detail and I do not have any major concerns. However, there are still few minor comments I would like authors to consider.

1. Growth conditions and data interpretation

- In previous work (Han et al., 2020), KIN10 was shown to be nuclear-localized, therefore stabilizing the stomatal development regulator SPCH, just in carbon starvation and low light conditions. In this paper Shi et al. observed that KIN10 mobilization in the nucleus takes place in normal light conditions and without carbon starvation. It would be interesting to hear authors thoughts on the possible reasons for these different results.

- Materials and Methods are described in detail and in addition, authors have also added method description in each figure legend and text. In order to avoid redundancy authors are encouraged to write all the experimental procedures in the Materials and Methods section, while in figure legends/text only methods differing from the standard procedure.

2. DMCU treatment

The authors have provided evidence DCMU treatment induces H₂O₂ accumulation in meristemoid cells and that both long and short treatment increase KIN10 nuclear localization. Moreover, they proved that KI, a quencher of H₂O₂, can suppress this effect. However, these results are still in contrast with the recent publication by Zoulias et al. (Curr Biol, 2021). Zoulias et al. reported no accumulation of ROS after DCMU treatment (Fig. S3F-G) and observed a reduced expression of the regulators of stomatal development SPCH and MUTE (Fig. 1H, Fig. 2), together with a reduced stomatal index and density (Fig. 1F-G). 10 μ M DCMU was sprayed once on seedlings at 3 or 7 DPG, and results observed from 2 to 72 hours after treatment are opposite compared to the ones shown in the paper and in Rebuttal Figure 1. Authors are encouraged to discuss what might have caused such a difference in the Discussion section.

3. Supplemental Figure 19 This image is new and is quite loosely connected to the main story. Please consider whether it is essential to add that data to the otherwise complete story.

4. Figure 4 d-g: Effect of KI on pSPCH::nucGFP and pSPCH::SPCH-GFP is only shown on cat2 cat3 background. If authors already have such a data, it would be interesting to see effect of KI on pSPCH::nucGFP and pSPCH::SPCH-GFP also in WT background.

5. Figure 5 h and text: "Consistent with this, the ratio of nuclear-to-cytoplasmic KIN10 protein was significantly decreased in p35S::KIN β 2-RFP transgenic plants, but increased in kin β 2 mutant, and H₂O₂ treatment significantly increased the nuclear-to-cytoplasmic ratio of KIN10 in the wild-type background, and such promotion effects of H₂O₂ were enhanced in pKIN10::KIN10-YFP/kin β 2 plants, but reduced in pKIN10::KIN10-YFP/p35S::KIN β 2-RFP plants (Fig. 5g, h)."

Authors describe the data (ratio of nuclear-to-cytoplasmic KIN10 protein) as a relative increase in mutant compared to WT when treated with H₂O₂. It would be also good to show the data as a relative numbers compared to WT, so it would be easier to see the magnitude of change.

Reviewer #2 (Remarks to the Author):

As I mentioned in my previous review, the study performed by Shi and coauthors demonstrated that spatially patterned hydrogen peroxide (H₂O₂) plays an essential role in stomatal development. I found the study highly interesting and addresses important aspects related to the involvement of redox-related signals in the development of stomata in *Arabidopsis* cotyledons.

The authors performed new experiments and analyses, and the results obtained satisfactorily answered all the questions I had asked about the previous version. The changes made to the text were appropriate. Thus, I believe that the manuscript has been significantly improved after this revision.

Reviewer #3 (Remarks to the Author):

Many thanks to the authors for the meticulous work and effort that they put to address the suggestions of this reviewer. The opinion of this reviewer is that there is an improvement in the new draft respect to the original version.

In order to be ready to be published and, overall, to make clear the message of the authors and avoid misinterpretation, there are a few things that require reformulation.

Line 86. The distribution of KIN10 is determined in a spatial manner, without a temporal frame. It should be mentioned "specific spatial pattern" instead of "specific spatiotemporal pattern in plants".

Line 90. The authors conclude that KIN10 is highly enriched in the stomatal lineage cells and guard cells. When linked to the previous sentence: "KIN10 exhibited a specific spatiotemporal pattern and was enriched in the nucleus of the stomatal lineage cells and guard cells as well as in the cytoplasm of the pavement cells in the epidermal cells of *Arabidopsis* cotyledons", it sounds that stomatal lineage cells and guard cells are the main location of KIN10 in epidermal tissue. Moreover, altogether these sentences can be misunderstood as in true leaves, KIN10 is mostly expressed in stomatal lineage cells. Clearly, the draft images show the stomatal localization of KIN10, though the statement is likely to mislead the readers to think that there is no expression in other pavement cells. Please rephrase it, to avoid to mislead the readers to conclude that KIN10 is exclusively or highly enriched only in SLC and GC in cotyledons and true leaves. KIN10 has been extensively found and identified in cells not belonging to stomatal lineage cells in epidermal tissue (Williams et al., 2014; Nukarinen et al, 2016, Jamsheer et al., 2018, Blanco et al., 2019), also in Fig 5 of this draft..

Line 93 & 97, same comment as Line 86.

Line 113 please change "were grown on 1/2 MS solid medium with or without DCMU and/or KI" by "were grown on 1/2 MS solid medium and sprayed with or without DCMU and/or KI".

Line 118. "Accumulation of ROS, including H₂O₂" instead of "accumulation of H₂O₂". Please, in the discussion it should be addressed the previous work which have identified the singlet oxygen as the main ROS produce by DCMU (it can be mentioned Nawrocki et al 2021).

Supplemental Fig 3 axis legend, please correct spelling of meristemoid

Line 143: the focal plane of the image is on the surface of the cell. It is not correct to state that "The results showed that the fluorescent signals of H2DCFDA and BES-H2O2-Ac were specifically enriched in the meristemoid cells and guard cells, but less distributed in pavement cells (Fig. 2a-h and Supplemental Fig. 8a-d)". Same observation as for Supp Fig1. The authors only show the adaxial side of pavement cell. Please rephrase the sentence.

Line 164: same comment as for Line 143, avoid including comparison with other pavement cells when they are not included in the figure as matter of comparison.

Line 181-182: "H2DCFDA fluorescent signals were observed only in the guard cells, but very weak in other cells". As before, the focal plane is on the surface of the tissue and is not enough to confirm the statement about the other cells.

Line 211. The colocalization analysis has no plugging that indeed indicates the degree of colocalization. Fig 3 should include at least the Bright field and the red and green channel in the same field. This schema should apply for pCAT2::GFP and IP (RFP) y and for pCAT2::GFP pSPCH::SPCH-RFP.

Line 251 Spelling "significant"

Line 252. Kin11 is redundant in function with Kin10..Any comment? What about to use a higher H2O2 concentration?. While in figure 2a and 2b, especially 2a; there is clear saturation of the signal, the 1mM H2O2 treatment to confirm the involvement of Kin10 in the process is far too low in terms of concentration (details about sensitivity Kristiansen et al., 2009 doi: 10.1111/j.1399-3054.2009.01243.x).

Reviewer #4 (Remarks to the Author):

I appreciate the efforts of the authors in addressing the questions raised on the previous version (inclusion of many new plant lines, description of experimental settings, addressig the potential denaturation of KIN10 protein in the in vitro assays, etc).

However, I still have several major concerns regarding this manuscript:

1) Statistics. It looks to me that in many of the graphs where ANOVA has been used, the variances of the samples are statistically significant (hence violating the assumption of the ANOVA test). Have the authors performed a test (e.g. Brown-Forsythe) to see whether this is indeed the case? If so, it would not be correct to use ANOVA for those analyses. This may explain why in some cases, what looks to be non-significant may reach statistical significance

2) Is the effect of H2O2 (ad DCMU) indeed a relocalization of KIN10 or a stabilization of the protein under those conditions?

- Authors have introduced new experiments in this version that address whether what appears to be a relocalization could rather be increased protein translation (CHX containing samples); however, they did not check whether the protein is stabilized under these conditions, as previously suggested. Proteasomal degradation of KIN10 has been shown to play a regulatory role in several instances (e.g. Ananieva et al., Crozet et al., Carvalho et al). Hence it would be important to ensure that this is indeed a protein relocalization and not protein stabilization

3) Is the effect of H2O2 on Kin10 localization specific to meristemoids or not? The manuscript has conflicting data on this.

- In Fig. 2l and 2m authors show that H2O2 accumulation is specific to meristemoids

- However, in other figures (e.g. Fig. 1) there seems to be a nuclear localization of KIN10 also in

SLGCs.

- Related to this, Suppl Fig. 7 shows that KIN10 does not relocalize to the nucleus in SLGCs in response to H₂O₂, whilst it does so in Fig. 1....?
- Also, the interaction between Kin10 and the Beta subunit that is shown in Fig. 5i not to be specific to meristemoids, is claimed to mediate the effect of H₂O₂ on KIN10 (in the meristemoids)
- In summary, there seems to be no clear correlation between H₂O₂ and KIN10 localization

4) Disconnect between different MS parts. There is no connection between what is shown in the beginning of the MS about how sucrose and increased light availability lead to increased nuclear Kin10 localization and the subsequent part on the effect of H₂O₂ on this. Although the authors discuss this in the discussion, the explanations are not very convincing.

- Does light/sugar availability lead to increased H₂O₂ production?
- What would be the physiological explanation for having similar consequences on KIN10 and stomatal development under carbon rich and carbon starvation conditions? If Kin10 nuclear localization happens in response to stresses like DCMU to promote stomatal development, why would the same be happening in response to higher light/sucrose availability?

REVIEWER COMMENTS

Reviewer #1 (Remarks to the Author):

Authors have made an impressive effort to revise the manuscript. Most of my comments have been addressed in detail and I do not have any major concerns.

Response: Thank you very much for the positive comment.

However, there are still few minor comments I would like authors to consider.

1. Growth conditions and data interpretation

- In previous work (Han et al., 2020), KIN10 was shown to be nuclear-localized, therefore stabilizing the stomatal development regulator SPCH, just in carbon starvation and low light conditions. In this paper Shi et al. observed that KIN10 mobilization in the nucleus takes place in normal light conditions and without carbon starvation. It would be interesting to hear authors thoughts on the possible reasons for these different results.

Response: Thank you for pointing this out. Our previous work showed that when plants were grown under mild energy starvation conditions, such as short-day photoperiod or liquid cultures, sucrose supply induced the protein accumulation of KIN10, which phosphorylated and stabilized SPCH to promote stomatal development (Han et al., 2020). Such promoting effects of sucrose and KIN10 on stomatal development were significantly reduced when plants were grown under high light irradiance or long-term photoperiod conditions compared to plants grown under low light irradiance or short-term photoperiod conditions, suggesting the low light irradiance or short-term photoperiod specifically activate KIN10 through an unknown mechanism. In a recent study, we showed that sucrose dynamically regulates stomatal development through fine-tuning KIN10 activity. Sucrose induced the total KIN10 protein accumulation at all tested concentrations, but only induced the phosphorylated KIN10 protein accumulation at the low concentrations. Under high sucrose concentrations, sucrose supply triggered the accumulation of trehalose-6-phosphate, which represses KIN10 activity by reducing the interaction between KIN10 and its

upstream kinase SnAK1/2. Consistent with this, sucrose promoted stomatal development at low concentrations, but such promoting activity decreased with the increasing sucrose concentrations, suggesting phosphorylation is an important regulatory mechanism for KIN10-mediated stomatal development (Han et al., 2022). Here, in this study, we showed that H₂O₂ induced the nuclear localization of KIN10 by reducing the interaction between KIN10 and KINβ2. In the epidermal cells of Arabidopsis leaves, H₂O₂ specifically accumulated in the meristemoid cells, which induced the nuclear localization of KIN10. Disruption of the specific spatial distribution of H₂O₂ in the epidermal cells using *SPCH* promoter to drive *CAT2* expression significantly inhibited stomatal development, suggesting H₂O₂-mediated KIN10 nuclear localization plays a critical roles for KIN10-mediated stomatal development. Together, sucrose-mediated KIN10 protein accumulation, H₂O₂-mediated KIN10 nuclear localization and KIN10 phosphorylation, these three regulator mechanisms finely regulate KIN10 activity to optimize stomatal development in response to diverse environmental and developmental signals.

References:

- Han, C., Qiao, Y., Yao, L., Hao, W., Liu, Y., Shi, W., Fan, M., and Bai, M.Y. (2022). TOR and SnRK1 fine tune SPEECHLESS transcription and protein stability to optimize stomatal development in response to exogenously supplied sugar. *New Phytol* **234**, 107-121.
- Han, C., Liu, Y., Shi, W., Qiao, Y., Wang, L., Tian, Y., Fan, M., Deng, Z., Lau, O.S., De Jaeger, G., and Bai, M.Y. (2020). KIN10 promotes stomatal development through stabilization of the SPEECHLESS transcription factor. *Nat Commun* **11**, 4214.

- Materials and Methods are described in detail and in addition, authors have also added method description in each figure legend and text. In order to avoid redundancy authors are encouraged to write all the experimental procedures in the Materials and Methods section, while in figure legends/text only methods differing from the standard procedure.

Response: Thank you for pointing this out. We have moved some descriptions about the detailed information of experiments in the figure legends to the Materials and Methods section.

2. DCMU treatment

The authors have provided evidence DCMU treatment induces H₂O₂ accumulation in meristemoid cells and that both long and short treatment increase KIN10 nuclear localization. Moreover, they proved that KI, a quencher of H₂O₂, can suppress this effect. However, these results are still in contrast with the recent publication by Zoulias et al. (Curr Biol, 2021). Zoulias et al. reported no accumulation of ROS after DCMU treatment (Fig. S3F-G) and observed a reduced expression of the regulators of stomatal development SPCH and MUTE (Fig. 1H, Fig. 2), together with a reduced stomatal index and density (Fig. 1F-G). 10µM DCMU was sprayed once on seedlings at 3 or 7 DPG, and results observed from 2 to 72 hours after treatment are opposite compared to the ones shown in the paper and in Rebuttal Figure 1. Authors are encouraged to discuss what might have caused such a difference in the Discussion section.

Response: Thank you for pointing this out. We have added a paragraph discussion as followed:

“A recent study reported that plastoquinone oxidation inhibits stomatal development by negatively regulating *SPCH* and *MUTE* expression (Zoulias et al., 2021). DCMU is a photosynthesis inhibitor, and can induce the oxidation of plastoquinone. DCMU treatment quickly induced the phosphorylation of MPK6 to activate MPK6. Activated MPK6 interacted and phosphorylated the epidermal specific expressed HD-ZIP transcription factor HDG2, and then inhibited the transcriptional activity of HDG2 and reduced the expression of *SPCH* and *MUTE*, thereby inhibiting stomatal development. In the present study, we showed that short-term DCMU treatment induced the nuclear localization of KIN10, while the long-term DCMU treatment led to KIN10 protein degradation. DCMU treatment induced the accumulation of H₂O₂ that reduced the interaction between KIN10 and KINβ2 and promoted KIN10 nuclear localization. Cotreatment of KI and DCMU counteracted the promoting effects of DCMU on KIN10 nuclear localization, suggesting H₂O₂ is required for DCMU-induced nuclear localization of KIN10. However, DCMU also has been reported to have no significant effects on H₂O₂

accumulation (Wang et al., 2010; Lokdarshi et al., 2020; Zoulias et al., 2021). One possibility for this inconsistency among different studies maybe the different inhibitory effects of DCMU on photosynthesis in different studies. Plastoquinone functions not only as an essential component of photosynthesis to transport electrons, but also acts as a potent antioxidant to regulate the state transitions and gene expression (Havaux, 2020). Treatment with short-term and low-dose DCMU leads to plastoquinone oxidation, while long-term and high-dose DCMU treatment leads to plastoquinone oxidation and H₂O₂ generation, and much high-dose DCMU treatment leads to higher H₂O₂ accumulation and programmed cell death.”

References:

- Havaux, M.** (2020). Plastoquinone In and Beyond Photosynthesis. *Trends Plant Sci* **25**, 1252-1265.
- Lokdarshi, A., Guan, J., Urquidi Camacho, R.A., Cho, S.K., Morgan, P.W., Leonard, M., Shimono, M., Day, B., and von Arnim, A.G.** (2020). Light Activates the Translational Regulatory Kinase GCN2 via Reactive Oxygen Species Emanating from the Chloroplast. *Plant Cell* **32**, 1161-1178.
- Wang, P., Du, Y., Li, Y., Ren, D., and Song, C.P.** (2010). Hydrogen peroxide-mediated activation of MAP kinase 6 modulates nitric oxide biosynthesis and signal transduction in Arabidopsis. *Plant Cell* **22**, 2981-2998.
- Zoulias, N., Rowe, J., Thomson, E.E., Dabrowska, M., Sutherland, H., Degen, G.E., Johnson, M.P., Sedelnikova, S.E., Hulmes, G.E., Hetteima, E.H., and Casson, S.A.** (2021). Inhibition of Arabidopsis stomatal development by plastoquinone oxidation. *Curr Biol* **31**, 5622-5632 e5627.

3. Supplemental Figure 19 This image is new and is quite loosely connected to the main story. Please consider whether it is essential to add that data to the otherwise complete story.

Response: Thank you for pointing this out. We agree your opinion, and have deleted this figure in the revised manuscript.

4. Figure 4 d-g: Effect of KI on pSPCH::nucGFP and pSPCH::SPCH-GFP is only shown on cat2 cat3 background. If authors already have such a data, it would be interesting to see effect of KI on pSPCH::nucGFP and pSPCH::SPCH-GFP also in WT background.

Response: Thank you for pointing this out. We have repeated this experiments and analyzed the effects of KI on *pSPCH::nucGFP* and *pSPCH::SPCH-GFP* in both Col-0 and *cat2 cat3* background (Fig. 4d-g).

Fig. 4 H₂O₂ promotes stomatal development.

d-g, *CAT* mutations altered cell fate in Arabidopsis epidermis. Seedlings of *pSPCH::nucGFP*, *pSPCH::nucGFP/cat2 cat3*, *pSPCH::SPCH-GFP*, and *pSPCH::SPCH-GFP/cat2 cat3* were grown on ½ MS solid medium containing 1% sucrose with or without 1mM KI under 16 h light/8h dark photoperiod with 100 μMol/m²/s for 3 days. Error bars indicate standard deviation (S.D.) (n =12–15). Different letters above the bars indicated statistically significant differences between the samples (Brown-Forsythe ANOVA analysis followed by Dunnett's T3 multiple comparisons test, *p* < 0.05). Scale bars in confocal images represent 20 μm.

5. Figure 5 h and text: “Consistent with this, the ratio of nuclear-to-cytoplasmic KIN10 protein was significantly decreased in p35S::KINβ2-RFP transgenic plants, but increased in kinβ2 mutant, and H₂O₂ treatment significantly increased the nuclear-to-cytoplasmic ratio of KIN10 in the wild-type background, and such promotion effects of H₂O₂ were enhanced in pKIN10::KIN10-YFP/kinβ2 plants, but reduced in pKIN10::KIN10-YFP/p35S::KINβ2-RFP plants (Fig. 5g, h).”

Authors describe the data (ratio of nuclear-to-cytoplasmic KIN10 protein) as a relative increase in mutant compared to WT when treated with H₂O₂. It would be also good to show the data as a relative number compared to WT, so it would be easier to see the magnitude of change.

Response: Thank you for pointing this out. We have analyzed the magnitude of

nucleocytoplasmic change of *pKIN10::KIN10-YFP* by H₂O₂ in different backgrounds. The results showed that H₂O₂ treatment significantly increased the nuclear-to-cytoplasmic ratio of KIN10 in the wild-type background, and such promotion effects of H₂O₂ were enhanced in *pKIN10::KIN10-YFP/p35S::KINβ2-RFP* plants, but reduced in *pKIN10::KIN10-YFP/kinβ2* plant. This unexpected results may be due to the less nuclear localization of KIN10 in *p35S::KINβ2-RFP* plants, and H₂O₂ treatment reduces the interaction between KINβ2 and KIN10, then promoting KIN10 nuclear localization. However, in the *kinβ2* mutant, most KIN10 protein already localized in the nucleus, and H₂O₂ had weak effects to promote more KIN10 protein to translocate into nucleus (Fig. 5g, h).

Fig. 5 H₂O₂ reduces the effects of KINβ2 on the KIN10-cytoplasmic retention in Arabidopsis leaves

g,h Seedlings of *pKIN10::KIN10-YFP*, *pKIN10::KIN10-YFP/kinβ2* and *pKIN10::KIN10-YFP/p35S::KINβ2-RFP* were grown on ½ MS solid medium under 16 h light/8h dark photoperiod with 100 μMol/m²/s for 4 days, and then treated with H₂O (Mock) or 2 mM H₂O₂ for 3 h. Serial Z-stack projection images were used for quantitative analysis. Numbers between bars indicated the magnitude of nucleo-cytoplasmic change of *pKIN10::KIN10-YFP* by H₂O₂ in different backgrounds. Asterisk indicated statistically significant differences of relative fold changes in *kinβ2* or *p35S::KINβ2-RFP* background compared to in WT background (Student's t test, *****p* < 0.0001, ****p* < 0.001, ***p* < 0.01). Scale bars in confocal images represent 10 μm.

Reviewer #2 (Remarks to the Author):

As I mentioned in my previous review, the study performed by Shi and coauthors demonstrated that spatially patterned hydrogen peroxide (H₂O₂) plays an essential role in stomatal development. I found the study highly interesting and addresses

important aspects related to the involvement of redox-related signals in the development of stomata in *Arabidopsis* cotyledons. The authors performed new experiments and analyses, and the results obtained satisfactorily answered all the questions I had asked about the previous version. The changes made to the text were appropriate. Thus, I believe that the manuscript has been significantly improved after this revision.

Response: Thank you very much for the positive comment.

Reviewer #3 (Remarks to the Author):

Many thanks to the authors for the meticulous work and effort that they put to address the suggestions of this reviewer. The opinion of this reviewer is that there is an improvement in the new draft respect to the original version.

Response: Thank you very much for the positive comment.

In order to be ready to be published and, overall, to make clear the message of the authors and avoid misinterpretation, there are a few things that require reformulation. Line 86. The distribution of KIN10 is determined in a spatial manner, without a temporal frame. It should be mentioned “specific spatial pattern” instead of “specific spatiotemporal pattern in plants”.

Response: Thank you for pointing this out. We have changed it.

Line 90. The authors conclude that KIN10 is highly enriched in the stomatal lineage cells and guard cells. When linked to the previous sentence: “KIN10 exhibited a specific spatiotemporal pattern and was enriched in the nucleus of the stomatal lineage cells and guard cells as well as in the cytoplasm of the pavement cells in the epidermal cells of *Arabidopsis* cotyledons”, it sounds that stomatal lineage cells and guard cells are the main location of KIN10 in epidermal tissue. Moreover, altogether these sentences can be misunderstood as in true leaves, KIN10 is mostly expressed in stomatal lineage cells. Clearly, the draft images show the stomatal localization of

KIN10, though the statement is likely to mislead the readers to think that there is no expression in other pavement cells. Please rephrase it, to avoid to mislead the readers to conclude that KIN10 is exclusively or highly enriched only in SLC and GC in cotyledons and true leaves. KIN10 has been extensively found and identified in cells not belonging to stomatal lineage cells in epidermal tissue (Williams et al., 2014; Nukarinen et al, 2016, Jamsheer et al., 2018, Blanco et al., 2019), also in Fig 5 of this draft..

Response: Thank you for pointing this out. We have changed them.

Line 93 & 97, same comment as Line 86.

Response: Thank you for pointing this out. We have changed it.

Line 113 please change “were grown on ½ MS solid medium with or without DCMU and/or KI” by “were grown on ½ MS solid medium and sprayed with or without DCMU and/or KI”.

Response: Thank you for pointing this out. We have changed it.

Line 118. “Accumulation of ROS, including H₂O₂” instead of “accumulation of H₂O₂”. Please, in the discussion it should be addressed the previous work which have identified the singlet oxygen as the main ROS produce by DCMU (it can be mentioned Nawrocki et al 2021).

Response: Thank you for pointing this out. We have changed it.

Supplemental Fig 3 axis legend, please correct spelling of meristemoid

Response: Thank you for pointing this out. We have changed it.

Line 143: the focal plane of the image is on the surface of the cell. It is not correct to state that “The results showed that the fluorescent signals of H₂DCFDA and BES-H₂O₂-Ac were specifically enriched in the meristemoid cells and guard cells, but less distributed in pavement cells (Fig. 2a-h and Supplemental Fig. 8a-d)”. Same

observation as for Supp Fig1. The authors only show the adaxial side of pavement cell. Please rephrase the sentence.

Response: Thank you for pointing this out. We have changed it.

Line 164: same comment as for Line 143, avoid including comparison with other pavement cells when they are not included in the figure as matter of comparison.

Response: Thank you for pointing this out. We have changed it.

Line 181-182: “H2DCFDA fluorescent signals were observed only in the guard cells, but very weak in other cells”. As before, the focal plane is on the surface of the tissue and is not enough to confirm the statement about the other cells.

Response: Thank you for pointing this out. We have changed it.

Line 211. The colocalization analysis has no plugging that indeed indicates the degree of colocalization. Fig 3 should include at least the Bright field and the red and green channel in the same field. This schema should apply for pCAT2::GFP and IP (RFP) y and for pCAT2::GFP pSPCH::SPCH-RFP.

Response: Thank you for pointing this out. We have added the bright field channel in the revised figure 3.

Fig. 3 SPCH directly represses the expression of CAT2.

a-d, The expression pattern of *CAT2* in the cotyledon epidermal cells. Seedlings of *pCAT2::GFP* were grown on ½ MS solid medium containing 1% sucrose under 16h light/8h dark photoperiod with 100 µMol/m²/s for 4 days. Magnifications of the epidermal cells are shown on the middle (b). The *pCAT2::GFP* fluorescent signals were in green, PI-marked cell outlines were in red. Merged images showed the higher *CAT2* expression levels in the pavement cells and the lower levels in the smaller cells where cell division occurs. Scale bars in confocal images represent 20 µm. **e-g**, Co-localization of *pCAT2::GFP* and *pSPCH::SPCH-RFP* in cotyledon epidermal cells. Seedlings of *pCAT2::GFP/pSPCH::SPCH-RFP* were grown on ½ MS solid medium containing 1% sucrose under 16h light/8h dark photoperiod with 100 µMol/m²/s for 3 days. Scale bars in confocal images represent 20 µm.

Line 251 Spelling “significant”

Response: Thank you for pointing this out. We have changed it.

Line 252. Kin11 is redundant in function with Kin10..Any comment?

Response: Thank you for pointing this out. Our previous study showed that KIN10 and KIN11 redundantly promote stomatal development, mutation of *KIN10* or *KIN11* both displayed the decreased stomatal index.

What about to use a higher H₂O₂ concentration?. While in figure 2a and 2b, especially 2a; there is clear saturation of the signal, the 1mM H₂O₂ treatment to confirm the involvement of Kin10 in the process is far too low in terms of concentration (details about sensitivity Kristiansen et al., 2009 doi: 10.1111/j.1399-3054.2009.01243.x).

Response: Thank you for pointing this out. Our results showed that H₂O₂ promotes stomatal development at low concentrations, and such promoting effects reaches the peak at about 0.1 mM, but decreases at 1 mM H₂O₂.

Reviewer #4 (Remarks to the Author):

I appreciate the efforts of the authors in addressing the questions raised on the previous version (inclusion of many new plant lines, description of experimental settings, addressig the potential denaturation of KIN10 protein in the in vitro assays,

etc).

Response: Thank you very much for the positive comment.

However, I still have several major concerns regarding this manuscript:

1) Statistics. It looks to me that in many of the graphs where ANOVA has been used, the variances of the samples are statistically significant (hence violating the assumption of the ANOVA test). Have the authors performed a test (e.g. Brown-Forsythe) to see whether this is indeed the case? If so, it would not be correct to use ANOVA for those analyses. This may explain why in some cases, what looks to be non-significant may reach statistical significance

Response: Thank you for pointing this out. We have analyzed the variance in all the datasets where ANOVA was used, and changed the statistical analysis method to Brown-Forsythe ANOVA analysis followed by Dunnett's T3 multiple comparisons test for the dataset with unequal variance.

2) Is the effect of H₂O₂ (ad DCMU) indeed a relocalization of KIN10 or a stabilization of the protein under those conditions?

- Authors have introduced new experiments in this version that address whether what appears to be a relocalization could rather be increased protein translation (CHX containing samples); however, they did not check whether the protein is stabilized under these conditions, as previously suggested. Proteasomal degradation of KIN10 has been shown to play a regulatory role in several instances (e.g. Ananieva et al., Crozet et al., Carvalho et al). Hence it would be important to ensure that this is indeed a protein relocalization and not protein stabilization

Response: Thank you for pointing this out. To determine whether DCMU induces the nuclear localization of KIN10 through regulating KIN10 relocalization, we analyzed the effects of DCMU and KI on the subcellular localization of KIN10 with or without proteasome inhibitor MG132. The results showed that DCMU promotes the nuclear localization of KIN10, but such promoting effects were suppressed by KI cotreatment in both presence or absence of MG132, suggesting the DCMU-induced nuclear

localization of KIN10 is independent on the protein stabilization of KIN10 (Rebuttal Fig. 1).

Rebuttal Fig. 1 MG132 had no significant effects on DCMU-induced nuclear localization of KIN10.

a-b, KI prevents DCMU-induced nuclear localization of KIN10 in Meristemoid and SLGC. Seedlings of *pKIN10::KIN10-YFP* were grown on ½ MS solid medium containing 1% sucrose under 16 h light/8h dark photoperiod with 100 µMol/m²/s for 4 days, and then treated with or without 50 µM DCMU and/or 1 mM KI and/or 30 µM MG132 for 12 hours. Serial Z-stack projection images were used for quantitative analysis. Different letters above the bars indicated statistically significant differences between the samples (Brown-Forsythe ANOVA analysis followed by Dunnett's T3 multiple comparisons test, *p* < 0.05). Scale bars in confocal images represent 10 µm.

3) Is the effect of H₂O₂ on Kin10 localization specific to meristemoids or not? The manuscript has conflicting data on this.

- In Fig. 2l and 2m authors show that H₂O₂ accumulation is specific to meristemoids
- However, in other figures (e.g. Fig. 1) there seems to be a nuclear localization of KIN10 also in SLGCs.
- Related to this, Suppl Fig. 7 shows that KIN10 does not relocalize to the nucleus in SLGCs in response to H₂O₂, whilst it does so in Fig. 1....?
- Also, the interaction between Kin10 and the Beta subunit that is shown in Fig. 5i not

to be specific to meristemoids, is claimed to mediate the effect of H₂O₂ on KIN10 (in the meristemoids)

- In summary, there seems to be no clear correlation between H₂O₂ and KIN10 localization

Response: Thank you for pointing this out. Our results showed that exogenous H₂O₂ treatment induced the nuclear localization of KIN10 not only in meristemoid and guard cells, but also in SLGC and pavement cells. In addition, H₂O₂ treatment also induced KIN10 nuclear localization in mesophyll cells, hypocotyl and root (Rebuttal Fig. 2). Consistent with this, exogenous H₂O₂ treatment reduced the interaction between KIN10 and KINβ2 in whole plants. In the intact leaves, H₂O₂ specifically accumulated in the meristemoid cell, thereby promoting the nuclear localization of KIN10 in the nucleus of meristemoid cells. To further verify the specific nuclear localization of KIN10 in the meristemoid cells, we analyzed the subcellular localization of KIN10-YFP and Histone3-RFP using the *pKIN10::KIN10-YFP/p35S::Histone3-RFP* transgenic plants, in which Histone3-RFP was used as an inner control. The results showed that Histone3-RFP was localized in the nucleus of all types of cells and KIN10 displayed the specific nuclear localization in the meristemoid and guard cells. H₂O₂ treatment had no significantly effects on the subcellular localization of Histone3-RFP, but promoted the nuclear localization of KIN10 in all types of cells (Supplementary Fig. 7). These results demonstrated that H₂O₂ plays an important role for KIN10 nuclear localization in plants.

Rebuttal Fig. 2 H₂O₂ induces the nuclear localization of KIN10.

a-f, H₂O₂ induced nuclear localization of KIN10 in mesophyll cells, hypocotyl cells, and root cells. Seedlings of *pKIN10::KIN10-YFP* were grown on ½ MS solid medium containing 1% sucrose under 16 h light/8h dark photoperiod with 100 μMol/m²/s for 4 days, and then treated with or without 2 mM H₂O₂ for 12 hours. Nuclear and cytoplasmic KIN10-YFP signal from more than 50 cells were analyzed by ImageJ software. Scale bars in confocal images represent 10 μm. Asterisk between the bars indicated statistically significant differences between the samples (Student's t test, *****p* < 0.0001).

Supplementary Fig. 7 Quantification of H₂O₂ effects on the subcellular localization of KIN10-YFP and Histone3-RFP in the *pKIN10::KIN10-YFP/p35S::H3-RFP* transgenic plants.

a-d, The subcellular localization of KIN10-YFP and Histone3-RFP in response to H₂O₂. Seedlings of *pKIN10::KIN10-YFP/p35S::H3-RFP* were grown on ½ MS solid medium containing 1% sucrose under 16h light/8h dark photoperiod with 100 μMol/m²/s for 4 days, and then treated with or without 2 mM H₂O₂ for 3 hours. Magnifications of the epidermal cells are shown on the middle (b). The fluorescent signals of GFP and RFP were determined along a line drawn on the confocal images using ImageJ software. The arrow labeling with A and B represents the fluorescent signals in meristemoid cells, and the arrow labeling with B and C represents the fluorescent signals in SLGC. **e**, Quantification of nuclear localization of KIN10 in different scale epidermal cells. **f**, Quantification of nuclear localization of Histone 3 in different scale epidermal cells.

Signals were taken using LSM700 microscope from Zeiss. Nuclear and cytoplasmic KIN10-YFP and Histone3-RFP signal from more than 200 epidermal cells in 10 cotyledons were analyzed by ImageJ software. Scale bars in panel (a) represent 20 μm, and in panel (b) represent 10 μm. Different letters above the bars indicated statistically significant differences between the samples (Brown-Forsythe ANOVA analysis followed by Dunnett's T3 multiple comparisons test, *p* < 0.05).

4) Disconnect between different MS parts. There is no connection between what is shown in the beginning of the MS about how sucrose and increased light availability lead to increased nuclear Kin10 localization and the subsequent part on the effect of

H₂O₂ on this. Although the authors discuss this in the discussion, the explanations are not very convincing.

- Does light/sugar availability lead to increased H₂O₂ production?
- What would be the physiological explanation for having similar consequences on KIN10 and stomatal development under carbon rich and carbon starvation conditions? If Kin10 nuclear localization happens in response to stresses like DCMU to promote stomatal development, why would the same be happening in response to higher light/sucrose availability?

Response: Thank you for pointing this out. Our previous work showed that when plants were grown under mild energy starvation conditions, such as short-day photoperiod or liquid cultures, sucrose supply induced the protein accumulation of KIN10, which phosphorylated and stabilized SPCH to promote stomatal development (Han et al., 2020). Such promoting effects of sucrose and KIN10 on stomatal development were significantly reduced when plants were grown under high light irradiance or long-term photoperiod conditions compared to plants grown under low light irradiance or short-term photoperiod conditions, suggesting the low light irradiance or short-term photoperiod specifically activate KIN10 through an unknown mechanism (Han et al., 2020). In a recent study, we showed that sucrose dynamically regulates stomatal development through fine-tuning KIN10 activity. Sucrose induced the total KIN10 protein accumulation at all tested concentrations, but only induced the phosphorylated KIN10 protein accumulation at the low concentrations. Under high sucrose concentrations, sucrose supply triggered the accumulation of trehalose-6-phosphate, which represses KIN10 activity by reducing the interaction between KIN10 and its upstream kinase SnAK1/2. Consistent with this, sucrose promoted stomatal development at low concentrations, but such promoting activity decreased with the increasing sucrose concentrations, suggesting phosphorylation is an important regulatory mechanism for KIN10-mediated stomatal development (Han et al., 2022). Here, in this study, we showed that H₂O₂ induced the nuclear localization of KIN10 by reducing the interaction between KIN10 and KINβ2. In the epidermal cells of Arabidopsis leaves, H₂O₂ specifically accumulated in the meristemoid cells,

which induced the nuclear localization of KIN10. Disruption of the specific spatial distribution of H₂O₂ in the epidermal cells using *SPCH* promoter to drive *CAT2* expression significantly inhibited stomatal development, suggesting H₂O₂-mediated KIN10 nuclear localization plays critical roles for KIN10-mediated stomatal development. Taken together, sucrose-mediated KIN10 protein accumulation, H₂O₂-mediated KIN10 nuclear localization, and KIN10 phosphorylation, these three regulator mechanisms finely regulate KIN10 activity to optimize stomatal development in response to diverse environmental and developmental signals.

References:

- Han, C., Qiao, Y., Yao, L., Hao, W., Liu, Y., Shi, W., Fan, M., and Bai, M.Y.** (2022). TOR and SnRK1 fine tune SPEECHLESS transcription and protein stability to optimize stomatal development in response to exogenously supplied sugar. *New Phytol* **234**, 107-121.
- Han, C., Liu, Y., Shi, W., Qiao, Y., Wang, L., Tian, Y., Fan, M., Deng, Z., Lau, O.S., De Jaeger, G., and Bai, M.Y.** (2020). KIN10 promotes stomatal development through stabilization of the SPEECHLESS transcription factor. *Nat Commun* **11**, 4214.

REVIEWERS' COMMENTS

Reviewer #1 (Remarks to the Author):

This manuscript has been greatly improved by the revision. I thank authors for addressing all my comments.

Reviewer #3 (Remarks to the Author):

Thanks to the authors for addressing my suggestions and questions. I congratulate them for the nice work and am looking forward to reading about their research in stomatal development in the future.

Reviewer #4 (Remarks to the Author):

I dont have any further comments to this manuscript. Congratulations to the authors.

REVIEWERS' COMMENTS

Reviewer #1 (Remarks to the Author):

This manuscript has been greatly improved by the revision. I thank authors for addressing all my comments.

Response: Thank you very much for the positive comments about our work.

Reviewer #3 (Remarks to the Author):

Thanks to the authors for addressing my suggestions and questions. I congratulate them for the nice work and am looking forward to reading about their research in stomatal development in the future.

Response: Thank you very much for the positive comments about our work.

Reviewer #4 (Remarks to the Author):

I dont have any further comments to this manuscript. Congratulations to the authors.

Response: Thank you very much for the positive comments about our work.